# Diffusion Schrödinger Bridge with Applications to Score-Based Generative Modeling

**Valentin De Bortoli**
Department of Statistics,
University of Oxford, UK

**James Thornton**
Department of Statistics,
University of Oxford, UK

**Jeremy Heng**
ESSEC Business School,
Singapore

**Arnaud Doucet**
Department of Statistics,
University of Oxford, UK

## Abstract

Progressively applying Gaussian noise transforms complex data distributions to approximately Gaussian. Reversing this dynamic defines a generative model. When the forward noising process is given by a Stochastic Differential Equation (SDE), Song et al. (2021) demonstrate how the time inhomogeneous drift of the associated reverse-time SDE may be estimated using score-matching. A limitation of this approach is that the forward-time SDE must be run for a sufficiently long time for the final distribution to be approximately Gaussian while ensuring that the corresponding time-discretization error is controlled. In contrast, solving the Schrödinger Bridge (SB) problem, *i.e.* an entropy-regularized optimal transport problem on path spaces, yields diffusions which generate samples from the data distribution in finite time. We present Diffusion SB (DSB), an original approximation of the Iterative Proportional Fitting (IPF) procedure to solve the SB problem, and provide theoretical analysis along with generative modeling experiments. The first DSB iteration recovers the methodology proposed by Song et al. (2021), with the flexibility of using shorter time intervals, as subsequent DSB iterations reduce the discrepancy between the final-time marginal of the forward (resp. backward) SDE with respect to the Gaussian prior (resp. data) distribution. Beyond generative modeling, DSB offers a computational optimal transport tool as the continuous state-space analogue of the popular Sinkhorn algorithm (Cuturi, 2013).

## 1  Introduction

*Score-Based Generative Modeling* (SGM) is a recently developed approach to probabilistic generative modeling that exhibits state-of-the-art performance on several audio and image synthesis tasks; see *e.g.* Song and Ermon (2019); Cai et al. (2020); Chen et al. (2021a); Kong et al. (2021); Gao et al. (2020); Jolicoeur-Martineau et al. (2021b); Ho et al. (2020); Song and Ermon (2020); Song et al. (2020, 2021); Niu et al. (2020); Durkan and Song (2021); Hoogeboom et al. (2021); Saharia et al. (2021); Luhman and Luhman (2021, 2020); Nichol and Dhariwal (2021); Popov et al. (2021); Dhariwal and Nichol (2021). Existing SGMs generally consist of two parts. Firstly, noise is incrementally added to the data in order to obtain a perturbed data distribution approximating an easy-to-sample *prior* distribution *e.g.* Gaussian. Secondly, a neural network is used to learn the reverse-time denoising dynamics, which when initialized at this prior distribution, defines a generative model (Sohl-Dickstein et al., 2015; Ho et al., 2020; Song and Ermon, 2019; Song et al., 2021). Song et al. (2021) have shown that one could fruitfully view the noising process as a Stochastic Differential Equation (SDE) that progressively perturbs the initial data distribution into an approximately Gaussian one.

35th Conference on Neural Information Processing Systems (NeurIPS 2021).

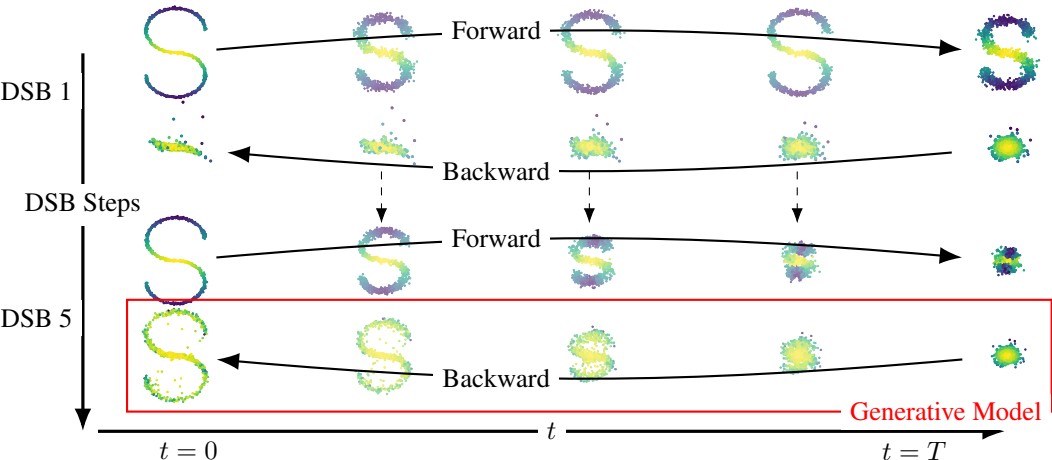

Figure 1: The reference forward diffusion initialized from the 2-dimensional data distribution fails to converge to the Gaussian prior in $T = 0.2$ diffusion-time ($N = 20$ discrete time steps), and the reverse diffusion initialized from the Gaussian prior does not converge to the data distribution. However, convergence does occur after 5 DSB iterations.

The corresponding reverse-time SDE is an inhomogeneous diffusion whose drift depends on the logarithmic gradients of the perturbed data distributions, *i.e.* the scores. In practice, these scores are approximated using neural networks and score-matching techniques (Hyvärinen and Dayan, 2005; Vincent, 2011) while numerical SDE integrators are used for the sampling procedure.

Although SGM provides state-of-the-art results (Dhariwal and Nichol, 2021), sample generation is computationally expensive. In order to learn the reverse-time SDE from the prior, *i.e.* the generative model, the forward noising SDE must be run for a sufficiently long time to converge to the prior and the step size must be sufficiently small to obtain a good numerical approximation of this SDE. By reformulating generative modeling as a Schrödinger bridge (SB) problem, we mitigate this issue and propose a novel algorithm to solve SB problems. Our detailed contributions are as follows.

**Generative modeling as a Schrödinger bridge problem.** The SB problem is a famous entropy-regularized Optimal Transport (OT) problem introduced by Schrödinger (1932); see *e.g.* (Léonard, 2014b; Chen et al., 2021b) for reviews. Given a reference diffusion with finite time horizon $T$, a data distribution and a prior distribution, solving the SB amounts to finding the closest diffusion to the reference (in terms of Kullback–Leibler divergence on path spaces) which admits the data distribution as marginal at time $t = 0$ and the prior at time $t = T$. The reverse-time diffusion solving this SB problem provides a new SGM algorithm which enables approximate sample generation from the data distribution using shorter time intervals compared to the original SGM methods. Our method differs from the entropy-regularized OT formulation in (Genevay et al., 2018), which deals with discrete distributions and relies on a static formulation of SB, as opposed to our dynamical approach for continuous distributions which operates on path spaces. It also differs from (Finlay et al., 2020) which approximates the SB solution by a diffusion whose drift is computed using potentials of the dual formulation of SB. Finally, Wang et al. (2021) have recently proposed to perform generative modeling by solving not one but two SB problems. Contrary to us, they do not formulate generative modeling as computing the SB between the data and prior distributions. D **Solving the Schrödinger bridge problem using score-based diffusions.** The SB problem can be solved using Iterative Proportional Fitting (IPF) (Fortet, 1940; Kullback, 1968; Chen et al., 2021b). We propose Diffusion SB (DSB), a novel implementation of IPF using score-based diffusion techniques. DSB does not require discretizing the state-space (Chen et al., 2016; Reich, 2019), approximating potential functions using regression (Bernton et al., 2019; Dessein et al., 2017; Pavon et al., 2021), nor performing kernel density estimation (Pavon et al., 2021). The first DSB iteration recovers the method proposed by Song et al. (2021), with the flexibility of using shorter time intervals, as additional DSB iterations reduce the discrepancy between the final-time marginal of the forward (resp. backward) SDE w.r.t. the prior (resp. data) distribution; see Figure 1 for an illustration. An algorithm akin to DSB has been proposed concurrently and independently by Vargas et al. (2021); the main difference with our algorithm is that they estimate the drifts of the SDEs using Gaussian processes while we use neural networks and score matching ideas.

**Theoretical results.** We provide the first quantitative convergence results for the methodology of Song et al. (2021). In particular, we show that while we simulate Langevin-type diffusions in potentially extremely high-dimensional spaces, the SGM approach does *not* suffer from poor mixing times. Additionally, we derive novel quantitative convergence results for IPF in continuous state-space which do not rely on classical compactness assumptions (Chen et al., 2016; Ruschendorf et al., 1995) and improve on the recent results of Léger (2020). Finally, we show that DSB may be viewed as the time discretization of a dynamic version of IPF on path spaces based on forward/backward diffusions.

**Experiments.** We validate our methodology by generating image datasets such as MNIST and CelebA. In particular, we show that using multiple steps of DSB always improve the generative model. We also show how DSB can be used to interpolate between two data distributions.

**Notation.** In the continuous-time setting, we set $\mathcal{C} = \mathrm{C}([0, T], \mathbb{R}^d)$ the space of continuous functions from $[0, T]$ to $\mathbb{R}^d$ and $\mathcal{B}(\mathcal{C})$ the Borel sets on $\mathcal{C}$. For any measurable space $(\mathsf{E}, \mathcal{E})$, we denote by $\mathscr{P}(\mathsf{E})$ the space of probability measures on $(\mathsf{E}, \mathcal{E})$. For any $\ell \in \mathbb{N}$, let $\mathscr{P}_\ell = \mathscr{P}((\mathbb{R}^d)^\ell)$. When it is defined, we denote $\mathrm{H}(p) = -\int_{\mathbb{R}^d} p(x) \log p(x) \mathrm{d}x$ as the entropy of $p$ and $\mathrm{KL}(p|q)$ as the Kullback–Leibler divergence between $p$ and $q$. When there is no ambiguity, we use the same notation for distributions and their densities. All proofs are postponed to the supplementary.

## 2 Denoising Diffusion, Score-Matching and Reverse-Time SDEs

### 2.1 Discrete-Time: Markov Chains and Time Reversal

Consider a data distribution with positive density $p_{\text{data}}$[1], a positive prior density $p_{\text{prior}}$ w.r.t. Lebesgue measure both with support on $\mathbb{R}^d$ and a Markov chain with initial density $p_0 = p_{\text{data}}$ on $\mathbb{R}^d$ evolving according to positive transition densities $p_{k+1|k}$ for $k \in \{0, \dots, N-1\}$. Hence for any $x_{0:N} = \{x_k\}_{k=0}^N \in \mathcal{X} = (\mathbb{R}^d)^{N+1}$, the joint density may be expressed as

$$p(x_{0:N}) = p_0(x_0) \prod_{k=0}^{N-1} p_{k+1|k}(x_{k+1}|x_k). \tag{1}$$

This joint density also admits the backward decomposition

$$p(x_{0:N}) = p_N(x_N) \prod_{k=0}^{N-1} p_{k|k+1}(x_k|x_{k+1}), \text{ with } p_{k|k+1}(x_k|x_{k+1}) = \frac{p_k(x_k)p_{k+1|k}(x_{k+1}|x_k)}{p_{k+1}(x_{k+1})}, \tag{2}$$

where $p_k(x_k) = \int p_{k|k-1}(x_k|x_{k-1})p_{k-1}(x_{k-1})\mathrm{d}x_{k-1}$ is the marginal density at step $k \geq 1$. For the purpose of generative modeling, we will choose transition densities such that $p_N(x_N) = \int p(x_{0:N})\mathrm{d}x_{0:N-1} \approx p_{\text{prior}}(x_N)$ for large $N$, where $p_{\text{prior}}$ is an easy-to-sample *prior* density. One may sample approximately from $p_{\text{data}}$ using ancestral sampling with the reverse-time decomposition (2), *i.e.* first sample $X_N \sim p_{\text{prior}}$ followed by $X_k \sim p_{k|k+1}(\cdot|X_{k+1})$ for $k \in \{N-1, \dots, 0\}$. This idea is at the core of all recent SGM methods. The reverse-time transitions in (2) cannot be simulated exactly but may be approximated if we consider a forward transition density of the form

$$p_{k+1|k}(x_{k+1}|x_k) = \mathcal{N}(x_{k+1}; x_k + \gamma_{k+1}f(x_k), 2\gamma_{k+1}\mathbf{I}), \tag{3}$$

with drift $f : \mathbb{R}^d \to \mathbb{R}^d$ and stepsize $\gamma_{k+1} > 0$. We first make the following approximation from (2)

$$
\begin{aligned}
p_{k|k+1}(x_k|x_{k+1}) &= p_{k+1|k}(x_{k+1}|x_k)\exp[\log p_k(x_k) - \log p_{k+1}(x_{k+1})] \\
&\approx \mathcal{N}(x_k; x_{k+1} - \gamma_{k+1}f(x_{k+1}) + 2\gamma_{k+1}\nabla \log p_{k+1}(x_{k+1}), 2\gamma_{k+1}\mathbf{I}),
\end{aligned} \tag{4}
$$

using that $p_k \approx p_{k+1}$, a Taylor expansion of $\log p_{k+1}$ at $x_{k+1}$ and $f(x_k) \approx f(x_{k+1})$. In practice, the approximation holds if $\|x_{k+1} - x_k\|$ is small which is ensured by choosing $\gamma_{k+1}$ small enough. Although $\nabla \log p_{k+1}$ is not available, one may obtain an approximation using denoising score-matching methods (Hyvärinen and Dayan, 2005; Vincent, 2011; Song et al., 2021).

Assume that the conditional density $p_{k+1|0}(x_{k+1}|x_0)$ is available analytically as in (Ho et al., 2020; Song et al., 2021). We have $p_{k+1}(x_{k+1}) = \int p_0(x_0)p_{k+1|0}(x_{k+1}|x_0)\mathrm{d}x_0$ and elementary calculations show that $\nabla \log p_{k+1}(x_{k+1}) = \mathbb{E}_{p_{0|k+1}}[\nabla_{x_{k+1}} \log p_{k+1|0}(x_{k+1}|X_0)]$. We can therefore

---

[1]In this presentation, we assume that all distributions admit a density w.r.t. the Lebesgue measure for simplicity. However, the algorithms presented here only require having access to samples from $p_{\text{data}}$ and $p_{\text{prior}}$.

formulate score estimation as a regression problem and use a flexible class of functions, *e.g.* neural networks, to parametrize an approximation $s_{\theta^\star}(k, x_k) \approx \nabla \log p_k(x_k)$ such that

$$\theta^\star = \arg \min_\theta \sum_{k=1}^N \mathbb{E}_{p_{0,k}}[||s_\theta(k, X_k) - \nabla_{x_k} \log p_{k|0}(X_k|X_0)||^2],$$

where $p_{0,k}(x_0, x_k) = p_0(x_0)p_{k|0}(x_k|x_0)$ is the joint density at steps 0 and $k$. If $p_{k|0}$ is not available, we use $\theta^\star = \arg \min_\theta \sum_{k=1}^N \mathbb{E}_{p_{k-1,k}}[||s_\theta(k, X_k) - \nabla_{x_k} \log p_{k|k-1}(X_k|X_{k-1})||^2]$. In summary, SGM involves first estimating the score function $s_{\theta^\star}$ from noisy data, and then sampling $X_0$ using $X_N \sim p_{\text{prior}}$ and the approximation (4), *i.e.*

$$X_k = X_{k+1} - \gamma_{k+1}f(X_{k+1}) + 2\gamma_{k+1}s_{\theta^\star}(k+1, X_{k+1}) + \sqrt{2\gamma_{k+1}}Z_{k+1}, Z_{k+1} \overset{\text{i.i.d.}}{\sim} \mathcal{N}(0, \mathbf{I}). \quad (5)$$

The random variable $X_0$ is approximately $p_0 = p_{\text{data}}$ distributed if $p_N(x_N) \approx p_{\text{prior}}(x_N)$. In what follows, we let $\{Y_k\}_{k=0}^N = \{X_{N-k}\}_{k=0}^N$ and remark that $\{Y_k\}_{k=0}^N$ satisfies a forward recursion.

## 2.2 Continuous-Time: SDEs, Reverse-Time SDEs and Theoretical results

For appropriate transition densities, Song et al. (2021) showed that the forward and reverse-time Markov chains may be viewed as discretized diffusions. We derive the continuous-time limit of the procedure presented in Section 2.1 and establish convergence results. The Markov chain with kernel (3) corresponds to an Euler–Maruyama discretization of $(\mathbf{X}_t)_{t\in[0,T]}$, solving the following SDE

$$d\mathbf{X}_t = f(\mathbf{X}_t)dt + \sqrt{2}d\mathbf{B}_t, \quad \mathbf{X}_0 \sim p_0 = p_{\text{data}}, \quad (6)$$

where $(\mathbf{B}_t)_{t\in[0,T]}$ is a Brownian motion and $f : \mathbb{R}^d \to \mathbb{R}^d$ is regular enough so that (strong) solutions exist. Under conditions on $f$, it is well-known (see Haussmann and Pardoux (1986); Föllmer (1985); Cattiaux et al. (2021) for instance) that the reverse-time process $(\mathbf{Y}_t)_{t\in[0,T]} = (\mathbf{X}_{T-t})_{t\in[0,T]}$ satisfies

$$d\mathbf{Y}_t = \{-f(\mathbf{Y}_t) + 2\nabla \log p_{T-t}(\mathbf{Y}_t)\} dt + \sqrt{2}d\mathbf{B}_t, \quad (7)$$

with initialization $\mathbf{Y}_0 \sim p_T$, where $p_t$ denotes the marginal density of $\mathbf{X}_t$.

The reverse-time Markov chain $\{Y_k\}_{k=0}^N$ associated with (5) corresponds to an Euler–Maruyama discretization of (7), where the score functions $\nabla \log p_t(x)$ are approximated by $s_{\theta^\star}(t, x)$.

In what follows, we consider $f(x) = -\alpha x$ for $\alpha \geq 0$. This framework includes the one of Song and Ermon (2019) ($\alpha = 0$, $p_{\text{prior}}(x) = \mathcal{N}(x; 0, 2T\mathbf{I})$) for which $(\mathbf{X}_t)_{t\in[0,T]}$ is simply a Brownian motion and Ho et al. (2020) ($\alpha > 0$, $p_{\text{prior}}(x) = \mathcal{N}(x; 0, \mathbf{I}/\alpha)$) for which it is an Ornstein–Uhlenbeck process, see Section S3.3 for more details. Contrary to Song et al. (2021) we consider time homogeneous diffusions. Both approaches approximate (5) using distinct discretizations but our setting leverages the ergodic properties of the Ornstein–Uhlenbeck process to establish Theorem 1.

**Theorem 1.** *Assume that there exists* $\mathtt{M} \geq 0$ *such that for any* $t \in [0, T]$ *and* $x \in \mathbb{R}^d$

$$\|s_{\theta^\star}(t, x) - \nabla \log p_t(x)\| \leq \mathtt{M}, \quad (8)$$

*with* $s_{\theta^\star} \in \mathrm{C}([0,T] \times \mathbb{R}^d, \mathbb{R}^d)$. *Assume that* $p_{\text{data}} \in \mathrm{C}^3(\mathbb{R}^d, (0, +\infty))$ *is bounded and that there exist* $d_1, A_1, A_2, A_3 \geq 0$, $\beta_1, \beta_2, \beta_3 \in \mathbb{N}$ *and* $\mathtt{m}_1 > 0$ *such that for any* $x \in \mathbb{R}^d$ *and* $i \in \{1, 2, 3\}$

$$\|\nabla^i \log p_{\text{data}}(x)\| \leq A_i(1 + \|x\|^{\beta_i}), \quad \langle \nabla \log p_{\text{data}}(x), x \rangle \leq -\mathtt{m}_1 \|x\|^2 + d_1 \|x\|,$$

*with* $\beta_1 = 1$. *Then for any* $\alpha \geq 0$, *there exist* $B_\alpha, C_\alpha, D_\alpha \geq 0$ *such that for any* $N \in \mathbb{N}$ *and* $\{\gamma_k\}_{k=1}^N$ *with* $\gamma_k > 0$ *for any* $k \in \{1, \ldots, N\}$, *the following bounds on the total variation distance hold:*

*(a) if* $\alpha > 0$, *we have* $\|\mathcal{L}(X_0) - p_{\text{data}}\|_{\text{TV}} \leq C_\alpha(\mathtt{M} + \bar{\gamma}^{1/2}) \exp[D_\alpha T] + B_\alpha \exp[-\alpha^{1/2}T]$;

*(b) if* $\alpha = 0$, *we have* $\|\mathcal{L}(X_0) - p_{\text{data}}\|_{\text{TV}} \leq C_0(\mathtt{M} + \bar{\gamma}^{1/2}) \exp[D_0 T] + B_0(T^{-1} + T^{-1/2})$;

*where* $T = \sum_{k=1}^N \gamma_k$, $\bar{\gamma} = \sup_{k\in\{1,\ldots,N\}} \gamma_k$ *and* $\mathcal{L}(X_0)$ *is the distribution of* $X_0$ *given in* (5).

*Proof.* We provide here a sketch of the proof. The whole proof is detailed in Section S3.2. Denote $\mathbb{P} \in \mathscr{P}(\mathcal{C})$ the path measure associated with (6) and $\mathbb{P}^R$ its time-reversal. Denote $Q_N$ the Markov kernel taking us from $Y_0$ to $Y_N$ induced by (5). We have

$$\|p_{\text{prior}}Q_N - p_{\text{data}}\|_{\text{TV}} = \|p_{\text{prior}}Q_N - p_{\text{data}}\mathbb{P}_{T|0}(\mathbb{P}^R)_{T|0}\|_{\text{TV}}$$

$$\leq \|p_{\text{prior}}Q_N - p_{\text{prior}}(\mathbb{P}^R)_{T|0}\|_{\text{TV}} + \|p_{\text{prior}}(\mathbb{P}^R)_{T|0} - p_{\text{data}}\mathbb{P}_{T|0}(\mathbb{P}^R)_{T|0}\|_{\text{TV}}$$
$$\leq \|p_{\text{prior}}Q_N - p_{\text{prior}}(\mathbb{P}^R)_{T|0}\|_{\text{TV}} + \|p_{\text{prior}} - p_T\|_{\text{TV}}.$$

We control the first term by bounding the discretization error of $Q_N$ when compared to $(\mathbb{P}^R)_{T|0}$ via the Girsanov theorem. The second term is controlled using the mixing properties of the forward diffusion process. □

Condition (8) ensures that the neural network approximates the score with a given precision $\mathtt{M} \geq 0$. Under (8) and conditions on $p_{\text{data}}$, Theorem 1 states how the Markov chain defined by (5) approximates $p_{\text{data}}$ in the total variation norm $\| \cdot \|_{\text{TV}}$. The bounds of Theorem 1 show that there is a trade-off between the mixing properties of the forward diffusion which increases with $\alpha$, and the quality of the discrete-time approximation which deteriorates as $\alpha$ and $T$ increase, since $B_\alpha, C_\alpha D_\alpha \to_{\alpha \to +\infty} +\infty$. Indeed increasing $\alpha$ makes the drift steeper and the continuous-time process converges faster but smaller step sizes are required in order to control the error between the discrete and the continuous-time processes. Theorem 1 is the first theoretical result assessing the convergence of SGM methods. Indeed while Block et al. (2020) establish convergence results for a *time-homogeneous* Langevin diffusion targeting a density whose score is approximated by a neural network, all SGM methods used in practice rely on *time-inhomogeneous* processes. Contrary to the time-homogeneous case, this approach does not suffer from poor mixing times as the mixing time dependency in the bounds of Theorem 1 is entirely determined by the mixing time of the *forward* process, given by a simple Brownian motion or an Ornstein–Ulhenbeck process, and is independent of the dimension. Finally, note that (8) is a strong assumption. In practice we expect to obtain such bounds in expectation over $X$ with high probability w.r.t. the data distribution as in (Block et al., 2020, Proposition 9). Our results are also related to (Tzen and Raginsky, 2019, Theorem 3.1) which establishes the expressiveness of related generative models using tools from stochastic control.

## 3    Diffusion Schrödinger Bridge and Generative Modeling

### 3.1    Schrödinger Bridges

The SB problem is a classical problem appearing in applied mathematics, optimal control and probability; see *e.g.* Föllmer (1988); Léonard (2014b); Chen et al. (2021b). In the discrete-time setting, it takes the following (dynamic) form. Consider as *reference* density $p(x_{0:N})$ given by (1), describing the process adding noise to the data. We aim to find $\pi^\star \in \mathscr{P}_{N+1}$ such that

$$\pi^\star = \arg\min \left\{ \text{KL}(\pi|p) \,:\, \pi \in \mathscr{P}_{N+1}, \, \pi_0 = p_{\text{data}}, \, \pi_N = p_{\text{prior}} \right\}. \tag{9}$$

Assuming $\pi^\star$ is available, a generative model can be obtained by sampling $X_N \sim p_{\text{prior}}$, followed by the reverse-time dynamics $X_k \sim \pi^\star_{k|k+1}(\cdot|X_{k+1})$ for $k \in \{N-1, \dots, 0\}$. Before deriving a method to approximate $\pi^\star$ in Section 3.2, we highlight some desirable features of Schrödinger bridges.

**Static Schrödinger bridge problem.**    First, we recall that the dynamic formulation (9) admits a static analogue. Using *e.g.* Léonard (2014a, Theorem 2.4), the following decomposition holds for any $\pi \in \mathscr{P}_{N+1}$, $\text{KL}(\pi|p) = \text{KL}(\pi_{0,N}|p_{0,N}) + \mathbb{E}_{\pi_{0,N}}[\text{KL}(\pi_{|0,N}|p_{|0,N})]$, where for any $\mu \in \mathscr{P}_{N+1}$ we have $\mu = \mu_{0,N}\mu_{|0,N}$ with $\mu_{|0,N}$ the conditional distribution of $X_{1:N-1}$ given $X_0, X_N$[2]. Hence we have $\pi^\star(x_{0:N}) = \pi^{\text{s},\star}(x_0, x_N)p_{|0,N}(x_{1:N-1}|x_0, x_N)$ where $\pi^{\text{s},\star} \in \mathscr{P}_2$ with marginals $\pi^{\text{s},\star}_0$ and $\pi^{\text{s},\star}_N$ is the solution of the static SB problem

$$\pi^{\text{s},\star} = \arg\min \left\{ \text{KL}(\pi^{\text{s}}|p_{0,N}) \,:\, \pi^{\text{s}} \in \mathscr{P}_2, \, \pi^{\text{s}}_0 = p_{\text{data}}, \, \pi^{\text{s}}_N = p_{\text{prior}} \right\}. \tag{10}$$

**Link with optimal transport.** Under mild assumptions, the static SB problem can be seen as an entropy-regularized optimal transport problem since (10) is equivalent to

$$\pi^{\text{s},\star} = \arg\min \left\{ -\mathbb{E}_{\pi^{\text{s}}}[\log p_{N|0}(X_N|X_0)] - \text{H}(\pi^{\text{s}}) \,:\, \pi^{\text{s}} \in \mathscr{P}_2, \, \pi^{\text{s}}_0 = p_{\text{data}}, \, \pi^{\text{s}}_N = p_{\text{prior}} \right\}.$$

If $p_{k+1|k}(x_{k+1}|x_k) = \mathcal{N}(x_{k+1}; x_k, \sigma^2_{k+1})$ as in Song and Ermon (2019), then $p_{N|0}(x_N|x_0) = \mathcal{N}(x_N; x_0, \sigma^2)$ with $\sigma^2 = \sum_{k=1}^N \sigma^2_k$ which induces a quadratic cost and

$$\pi^{\text{s},\star} = \arg\min \left\{ \mathbb{E}_{\pi^{\text{s}}}[\|X_0 - X_N\|^2] - 2\sigma^2 \text{H}(\pi^{\text{s}}) \,:\, \pi^{\text{s}} \in \mathscr{P}_2, \, \pi^{\text{s}}_0 = p_{\text{data}}, \, \pi^{\text{s}}_N = p_{\text{prior}} \right\}.$$

---

[2]See Section S4.1 for a rigorous presentation using the disintegration theorem for probability measures.

Mikami (2004) showed that $\pi^{\mathrm{s},\star} \to \pi^{\star}_{\mathcal{W}}$ weakly and $2\sigma^2 \mathrm{KL}(\pi^{\mathrm{s},\star}|p_{0,N}) \to \mathcal{W}_2^2(p_{\mathrm{data}}, p_{\mathrm{prior}})$ as $\sigma \to 0$, where $\pi^{\star}_{\mathcal{W}}$ is the optimal transport plan between $p_{\mathrm{data}}$ and $p_{\mathrm{prior}}$ and $\mathcal{W}_2$ is the 2-Wasserstein distance. Note that the transport cost $c(x, x') = -\log p_{N|0}(x'|x)$ is not necessarily symmetric.

## 3.2   Iterative Proportional Fitting and Time Reversal

In all but trivial cases, the SB problem does not admit a closed-form solution. However, it can be solved using Iterative Proportional Fitting (IPF) (Fortet, 1940; Kullback, 1968; Ruschendorf et al., 1995) which is defined by the following recursion for $n \in \mathbb{N}$ with initialization $\pi^0 = p$ given in (1):

$$\pi^{2n+1} = \arg\min \left\{ \mathrm{KL}(\pi|\pi^{2n}) \,:\, \pi \in \mathscr{P}_{N+1}, \, \pi_N = p_{\mathrm{prior}} \right\}, \tag{11}$$

$$\pi^{2n+2} = \arg\min \left\{ \mathrm{KL}(\pi|\pi^{2n+1}) \,:\, \pi \in \mathscr{P}_{N+1}, \, \pi_0 = p_{\mathrm{data}} \right\}.$$

This sequence is well-defined if there exists $\tilde{\pi} \in \mathscr{P}_{N+1}$ such that $\tilde{\pi}_0 = p_{\mathrm{data}}$, $\tilde{\pi}_N = p_{\mathrm{prior}}$ and $\mathrm{KL}(\tilde{\pi}|p) < +\infty$. A standard representation of $\pi^n$ is obtained by updating the joint density $p$ using potential functions, see Section S4.2 for details. However, this representation of the IPF iterates is difficult to approximate as it requires approximating the potentials. Our methodology builds upon an alternative representation that is better suited to numerical approximations for generative modeling where one has access to samples of $p_{\mathrm{data}}$ and $p_{\mathrm{prior}}$.

**Proposition 2.** *Assume that* $\mathrm{KL}(p_{\mathrm{data}} \otimes p_{\mathrm{prior}}|p_{0,N}) < +\infty$. *Then for any* $n \in \mathbb{N}$, $\pi^{2n}$ *and* $\pi^{2n+1}$ *admit positive densities w.r.t. the Lebesgue measure denoted as* $p^n$ *resp.* $q^n$ *and for any* $x_{0:N} \in \mathcal{X}$, *we have* $p^0(x_{0:N}) = p(x_{0:N})$ *and*

$$q^n(x_{0:N}) = p_{\mathrm{prior}}(x_N) \prod_{k=0}^{N-1} p^n_{k|k+1}(x_k|x_{k+1}), \, p^{n+1}(x_{0:N}) = p_{\mathrm{data}}(x_0) \prod_{k=0}^{N-1} q^n_{k+1|k}(x_{k+1}|x_k).$$

In practice we have access to $p^n_{k+1|k}$ and $q^n_{k|k+1}$. Hence, to compute $p^n_{k|k+1}$ and $q^n_{k+1|k}$ we use

$$p^n_{k|k+1}(x_k|x_{k+1}) = \frac{p^n_{k+1|k}(x_{k+1}|x_k) p^n_k(x_k)}{p^n_{k+1}(x_{k+1})}, \, q^n_{k+1|k}(x_{k+1}|x_k) = \frac{q^n_{k|k+1}(x_k|x_{k+1}) q^n_{k+1}(x_{k+1})}{q^n_k(x_k)}.$$

To the best of our knowledge, this representation of the IPF iterates has surprisingly neither been presented nor explored in the literature. One may interpret these formulas as follows. At iteration $2n$, we have $\pi^{2n} = p^n$ with $p^0 = p$ given by the noising process (1). This forward process initalized with $p^n_0 = p_{\mathrm{data}}$ defines reverse-time transitions $p^n_{k|k+1}$, which, when combined with an initialization $p_{\mathrm{prior}}$ at step $N$ defines the reverse-time process $\pi^{2n+1} = q^n$. The forward transitions $q^n_{k+1|k}$ associated to $q^n$ are then used to obtain $\pi^{2n+2} = p^{n+1}$. IPF then iterates this procedure.

## 3.3   Diffusion Schrödinger Bridge as Iterative Mean-Matching Proportional Fitting

To approximate the IPF recursion defined in Proposition 2, we use similar approximations to Section 2.1. If at step $n \in \mathbb{N}$ we have $p^n_{k+1|k}(x_{k+1}|x_k) = \mathcal{N}(x_{k+1}; x_k + \gamma_{k+1} f^n_k(x_k), 2\gamma_{k+1}\mathbf{I})$ where $p^0 = p$ and $f^0_k = f$, then we can approximate the reverse-time transitions in Proposition 2 by

$$q^n_{k|k+1}(x_k|x_{k+1}) = p^n_{k+1|k}(x_{k+1}|x_k) \exp[\log p^n_k(x_k) - \log p^n_{k+1}(x_{k+1})]$$

$$\approx \mathcal{N}(x_k; x_{k+1} + \gamma_{k+1} b^n_{k+1}(x_{k+1}), 2\gamma_{k+1}\mathbf{I}),$$

with $b^n_{k+1}(x_{k+1}) = -f^n_k(x_{k+1}) + 2\nabla \log p^n_{k+1}(x_{k+1})$. We can also approximate the forward transitions in Proposition 2 by $p^{n+1}_{k+1|k}(x_{k+1}|x_k) \approx \mathcal{N}(x_{k+1}; x_k + \gamma_{k+1} f^{n+1}_k(x_k), 2\gamma_{k+1}\mathbf{I})$ with $f^{n+1}_k(x_k) = -b^n_{k+1}(x_k) + 2\nabla \log q^n_k(x_k)$. Hence we have $f^{n+1}_k(x_k) = f^n_k(x_k) - 2\nabla \log p^n_{k+1}(x_k) + 2\nabla \log q^n_k(x_k)$. It follows that one could estimate $f^{n+1}_k, b^{n+1}_k$ by using score-matching to approximate $\{\nabla \log p^i_{k+1}(x)\}_{i=0}^n$, $\{\nabla \log q^i_k(x)\}_{i=0}^n$. This approach is prohibitively costly in terms of memory and compute, see Section S5. We follow an alternative approach which avoids these difficulties.

**Proposition 3.** *Assume that for any* $n \in \mathbb{N}$ *and* $k \in \{0, \ldots, N-1\}$,

$$q^n_{k|k+1}(x_k|x_{k+1}) = \mathcal{N}(x_k; B^n_{k+1}(x_{k+1}), 2\gamma_{k+1}\mathbf{I}), \, p^n_{k+1|k}(x_{k+1}|x_k) = \mathcal{N}(x_{k+1}; F^n_k(x_k), 2\gamma_{k+1}\mathbf{I}),$$

*with* $B^n_{k+1}(x) = x + \gamma_{k+1} b^n_{k+1}(x)$, $F^n_k(x) = x + \gamma_{k+1} f^n_k(x)$ *for any* $x \in \mathbb{R}^d$. *Then we have for any* $n \in \mathbb{N}$ *and* $k \in \{0, \ldots, N-1\}$

$$B^n_{k+1} = \arg\min_{B \in L^2(\mathbb{R}^d, \mathbb{R}^d)} \mathbb{E}_{p^n_{k,k+1}}[\|B(X_{k+1}) - (X_{k+1} + F^n_k(X_k) - F^n_k(X_{k+1}))\|^2], \tag{12}$$

$$F^{n+1}_k = \arg\min_{F \in L^2(\mathbb{R}^d, \mathbb{R}^d)} \mathbb{E}_{q^n_{k,k+1}}[\|F(X_k) - (X_k + B^n_{k+1}(X_{k+1}) - B^n_{k+1}(X_k))\|^2]. \tag{13}$$

Proposition 3 shows how one can recursively approximate $B_{k+1}^n$ and $F_k^{n+1}$. In practice, we use neural networks $B_{\beta^n}(k, x) \approx B_k^n(x)$ and $F_{\alpha^n}(k, x) \approx F_k^n(x)$. Note that the networks could also be learned jointly. In this case, at equilibrium, we would obtain a bridge between $p_{\text{data}}$ and $p_{\text{prior}}$ but not necessarily the Schrödinger bridge.

Network parameters $\alpha^n, \beta^n$ are learnt through gradient descent to minimize empirical versions of the sum over $k$ of the loss functions given by (12) and (13) computed using $M$ samples and denoted as $\hat{\ell}_n^b(\beta)$ and $\hat{\ell}_{n+1}^f(\alpha)$. The resulting algorithm approximating $L \in \mathbb{N}$ IPF iterations is called Diffusion Schrödinger Bridge (DSB) and is summarized in Algorithm 1 with $Z_k^j, \tilde{Z}_k^j \overset{\text{i.i.d.}}{\sim} \mathcal{N}(0, \mathbf{I})$, see Figure 1 for an illustration.

---

**Algorithm 1** Diffusion Schrödinger Bridge

1: **for** $n \in \{0, \dots, L\}$ **do**
2:     **while** not converged **do**
3:         Sample $\{X_k^j\}_{k,j=0}^{N,M}$, where $X_0^j \sim p_{\text{data}}$, and
        $X_{k+1}^j = F_{\alpha^n}(k, X_k^j) + \sqrt{2\gamma_{k+1}}Z_{k+1}^j$
4:         Compute $\hat{\ell}_n^b(\beta^n)$ approximating (12)
5:         $\beta^n \leftarrow$ Gradient Step$(\hat{\ell}_n^b(\beta^n))$
6:     **end while**
7:     **while** not converged **do**
8:         Sample $\{X_k^j\}_{k,j=0}^{N,M}$, where $X_N^j \sim p_{\text{prior}}$, and
        $X_{k-1}^j = B_{\beta^n}(k, X_k^j) + \sqrt{2\gamma_k}\tilde{Z}_k^j$
9:         Compute $\hat{\ell}_{n+1}^f(\alpha^{n+1})$ approximating (13)
10:       $\alpha^{n+1} \leftarrow$ Gradient Step$(\hat{\ell}_{n+1}^f(\alpha^{n+1}))$
11:     **end while**
12: **end for**
13: **Output:** $(\alpha^{L+1}, \beta^L)$

---

The DSB algorithm is initialized using the reference dynamics $f_{\alpha^0}(k, x) = f(x)$. Once $\beta^L$ is learnt we can easily approximately sample from $p_{\text{data}}$ by sampling $X_N \sim p_{\text{prior}}$ and then using $X_{k-1} = B_{\beta^L}(k, X_k) + \sqrt{2\gamma_k}Z_k$ with $Z_k \overset{\text{i.i.d.}}{\sim} \mathcal{N}(0, \mathbf{I})$. The resulting samples $X_0$ will be approximately distributed from $p_{\text{data}}$. Although DSB requires learning a sequence of network parameters, $\alpha^n, \beta^n$, fewer diffusion steps are needed compared to standard SGM. In addition, as detailed in Section S9, $\beta^0$ may be trained efficiently in a similar manner to previous SGM methods. Subsequent $\alpha^{n+1}, \beta^{n+1}$ are refinements of $\alpha^n, \beta^n$, hence may be fine-tuned from previous iterations.

### 3.4 Convergence of Iterative Proportional Fitting

In this section, we investigate the theoretical properties of IPF. When the state-space is discrete and finite (Franklin and Lorenz, 1989; Peyré and Cuturi, 2019) or in the case where $p_{\text{data}}$ and $p_{\text{prior}}$ are compactly supported (Chen et al., 2016), IPF converges at a geometric rate w.r.t. the Hilbert-Birkhoff metric, see Lemmens and Nussbaum (2014) for a definition. Other than recent work by Léger (2020), only qualitative results exist in the general case where $p_{\text{data}}$ or $p_{\text{prior}}$ is not compactly supported (Ruschendorf et al., 1995; Rüschendorf and Thomsen, 1993). We establish here quantitative convergence of IPF in this non-compact setting as well as novel monotonicity results. We require only the following mild assumption.

**A1.** $p_N, p_{\text{prior}} > 0$, $|\text{H}(p_{\text{prior}})| < +\infty$, $\int_{\mathbb{R}^d} |\log p_{N|0}(x_N|x_0)| p_{\text{data}}(x_0) p_{\text{prior}}(x_N) \mathrm{d}x_0 \mathrm{d}x_N < +\infty$.

Assumption **A1** is satisfied in all of our experimental settings. We recall that for $\mu, \nu \in \mathscr{P}(\mathsf{E})$ with $(\mathsf{E}, \mathcal{E})$ a measurable space, the Jeffrey's divergence is given by $\text{J}(\mu, \nu) = \text{KL}(\mu|\nu) + \text{KL}(\nu|\mu)$.

**Proposition 4.** *Assume **A1**. Then $(\pi^n)_{n \in \mathbb{N}}$ is well-defined and for any $n \geq 1$ we have*

$$\text{KL}(\pi^{n+1}|\pi^n) \leq \text{KL}(\pi^{n-1}|\pi^n), \qquad \text{KL}(\pi^n|\pi^{n+1}) \leq \text{KL}(\pi^n|\pi^{n-1}).$$

*In addition, $(\|\pi^{n+1} - \pi^n\|_{\text{TV}})_{n \in \mathbb{N}}$ and $(\text{J}(\pi^{n+1}, \pi^n))_{n \in \mathbb{N}}$ are non-increasing. Finally, we have $\lim_{n \to +\infty} n \{\text{KL}(\pi_0^n|p_{\text{data}}) + \text{KL}(\pi_N^n|p_{\text{prior}})\} = 0$.*

A more general result with additional monotonicity properties is given in Section S6. Under similar assumptions, Léger (2020, Corollary 1) established $\text{KL}(\pi_0^n|p_0) \leq C/n$ with $C \geq 0$ using a Bregman divergence gradient descent perspective. In contrast, our proof relies only on tools from information geometry. In addition, we improve the convergence rate and show that $(\pi^n)_{n \in \mathbb{N}}$ converges in total variation towards $\pi^\infty$, *i.e.* we not only obtain convergence of the marginals but also convergence of the joint distribution. Under restrictive conditions on $p_{\text{data}}$ and $p_{\text{prior}}$, Ruschendorf et al. (1995) showed that $\pi^\infty$ is the Schrödinger bridge. In the following proposition, we avoid this assumption using results on automorphisms of measures (Beurling, 1960).

**Proposition 5.** *Assume* **A**1. *Then there exists a solution* $\pi^\star \in \mathscr{P}_{N+1}$ *to the SB problem. Assume that* $\lim_{n\to+\infty} \|\pi^n - \pi^\infty\|_{TV} = 0$ *with* $\pi^\infty \in \mathscr{P}_{N+1}$. *Let* $h = p_{0,N}/(p_0 \otimes p_N)$ *and assume that* $h \in \mathrm{C}((\mathbb{R}^d)^2, (0, +\infty))$ *and that there exist* $\Phi_0, \Phi_N \in \mathrm{C}(\mathbb{R}^d, (0, +\infty))$ *such that*

$$\int_{\mathbb{R}^d \times \mathbb{R}^d} (|\log h(x_0, x_N)| + |\log \Phi_0(x_0)| + |\log \Phi_N(x_N)|) p_{\mathrm{data}}(x_0) p_{\mathrm{prior}}(x_N) \mathrm{d}x_0 \mathrm{d}x_N < +\infty,$$

*with* $h(x_0, x_N) \leq \Phi_0(x_0) \Phi_N(x_N)$. *If* $p$ *is absolutely continuous w.r.t.* $\pi^\infty$ *then* $\pi^\infty = \pi^\star$.

Proposition 5 extends previous IPF convergence results without the assumption that the mapping $h$ is lower bounded, see Ruschendorf et al. (1995); Chen et al. (2016). Our assumption on $h$ can be relaxed and replaced by a tighter condition on $\pi^\infty$, see Section S6.2. Proposition 4 suggests a convergence rate of order $o(n)$ for the IPF in the non-compact setting. However, in some situations, we recover geometric convergence rates with explicit dependency w.r.t. the problem constants, see Section S7. In practice, we do not run IPF for $p_{\mathrm{data}}, p_{\mathrm{prior}}$ but using empirical versions of these distributions. Recent results in Deligiannidis et al. (2021) show that the iterates of IPF based on empirical distributions remain close to the iterates one would obtain using the true distributions, uniformly in time. In particular, the SB computed using the empirical distributions converges to the one computed using the true distributions as the number of samples goes to infinity.

### 3.5 Continuous-time IPF

We describe an IPF algorithm for solving SB problems in continuous-time. We show that DSB proposed in Algorithm 1 can be seen as a discretization of this IPF. Given a reference measure $\mathbb{P} \in \mathscr{P}(\mathcal{C})$, the continuous formulation of the SB involves solving the following problem

$$\Pi^\star = \arg\min \{ \mathrm{KL}(\Pi|\mathbb{P}) \, : \, \Pi \in \mathscr{P}(\mathcal{C}), \, \Pi_0 = p_{\mathrm{data}}, \, \Pi_T = p_{\mathrm{prior}} \}, \quad T = \sum_{k=0}^{N-1} \gamma_{k+1}.$$

Similarly to (11), we define the IPF $(\Pi^n)_{n\in\mathbb{N}}$ with $\Pi^0 = \mathbb{P}$ associated with (6) and for any $n \in \mathbb{N}$

$$\Pi^{2n+1} = \arg\min \{ \mathrm{KL}(\Pi|\Pi^{2n}) \, : \, \Pi \in \mathscr{P}(\mathcal{C}), \, \Pi_T = p_{\mathrm{prior}} \},$$

$$\Pi^{2n+2} = \arg\min \{ \mathrm{KL}(\Pi|\Pi^{2n+1}) \, : \, \Pi \in \mathscr{P}(\mathcal{C}), \, \Pi_0 = p_{\mathrm{data}} \}.$$

One can show that for any $n \in \mathbb{N}$, $\Pi^n = \pi^{\mathrm{s},n} \mathbb{P}_{|0,T}$, with $(\pi^{\mathrm{s},n})_{n\in\mathbb{N}}$ the IPF for the static SB problem. In particular, Proposition 4 and Proposition 5 extend to the continuous IPF framework. In what follows, for any $\mathbb{P} \in \mathscr{P}(\mathcal{C})$, we define $\mathbb{P}^R$ as the reverse-time measure, *i.e.* for any $\mathsf{A} \in \mathcal{B}(\mathcal{C})$ we have $\mathbb{P}^R(\mathsf{A}) = \mathbb{P}(\mathsf{A}^R)$ where $\mathsf{A}^R = \{ t \mapsto \omega(T-t) \, : \, \omega \in \mathsf{A} \}$. The following result is the continuous counterpart of Proposition 2 and states that each IPF iteration is associated with a diffusion, showing that DSB can be seen as a discretization of the continuous IPF.

**Proposition 6.** *Assume* **A**1 *and that there exist* $\mathbb{M} \in \mathscr{P}(\mathcal{C})$, $U \in \mathrm{C}^1(\mathbb{R}^d, \mathbb{R})$, $C \geq 0$ *such that for any* $n \in \mathbb{N}$, $x \in \mathbb{R}^d$, $\mathrm{KL}(\Pi^n|\mathbb{M}) < +\infty$, $\langle x, \nabla U(x) \rangle \geq -C(1 + \|x\|^2)$ *and* $\mathbb{M}$ *is associated with*

$$\mathrm{d}\mathbf{X}_t = -\nabla U(\mathbf{X}_t)\mathrm{d}t + \sqrt{2}\mathrm{d}\mathbf{B}_t, \tag{14}$$

*with* $\mathbf{X}_0$ *distributed according to the invariant distribution of* (14). *Then, for any* $n \in \mathbb{N}$ *we have:*

*(a)* $(\Pi^{2n+1})^R$ *is associated with* $\mathrm{d}\mathbf{Y}_t^{2n+1} = b_{T-t}^n(\mathbf{Y}_t^{2n+1})\mathrm{d}t + \sqrt{2}\mathrm{d}\mathbf{B}_t$ *with* $\mathbf{Y}_0^{2n+1} \sim p_{\mathrm{prior}}$;

*(b)* $\Pi^{2n+2}$ *is associated with* $\mathrm{d}\mathbf{X}_t^{2n+2} = f_t^{n+1}(\mathbf{X}_t^{2n+2})\mathrm{d}t + \sqrt{2}\mathrm{d}\mathbf{B}_t$ *with* $\mathbf{X}_0^{2n+2} \sim p_{\mathrm{data}}$; *where for any* $n \in \mathbb{N}$, $t \in [0, T]$ *and* $x \in \mathbb{R}^d$, $b_t^n(x) = -f_t^n(x) + 2\nabla \log p_t^n(x)$, $f_t^{n+1}(x) = -b_t^n(x) + 2\nabla \log q_t^n(x)$, *with* $f_t^0(x) = f(x)$, *see* (6), *and* $p_t^n$, $q_t^n$ *the densities of* $\Pi_t^{2n}$ *and* $\Pi_t^{2n+1}$.

## 4 Experiments

**Gaussian example.** We first confirm that our algorithm recovers the true SB in a Gaussian setting where the ground truth is available. Let $p_{\mathrm{prior}} = \mathcal{N}(-a, \mathbf{I})$, $p_{\mathrm{data}} = \mathcal{N}(a, \mathbf{I})$ with $a \in \mathbb{R}^d$ and consider a Brownian motion as reference dynamics. The analytic expression for the static SB is $\mathcal{N}((-a, a), \Sigma)$ with $\Sigma \in \mathbb{R}^{2d \times 2d}$ given in Section S7.2. We let $a = 0.1 \times \mathbf{1}$ with $d = 50$ or $d = 5$. In Figure 2, we illustrate the convergence of DSB. We train each DSB with a batch size of 128, $N = 20$ and $\gamma = 1/40$. We compare two network configurations: "small" where the network is given by Figure S2 (30k parameters) whereas "large" corresponds to the same network but with twice as many latent dimensions (240k parameters). The small network recovers the statistics of SB in the low-dimensional setting ($d = 5$) but is unable to recover the variance and covariance for $d = 50$. Increasing the size of the network solves this problem.

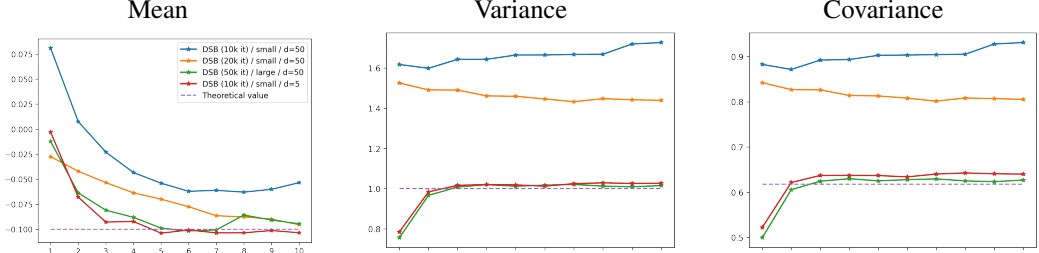

Figure 2: Convergence of DSB to ground-truth. From left to right: estimated mean, variance and covariance (first component) after each DSB iteration. The ground-truth value is given by the dashed line in each scenario.

**Two dimensional toy experiments.** We evaluate the validity of our approach on toy two dimensional examples. Contrary to existing SGM approaches we do *not* require that the number of steps is large enough for $p_N \approx p_{\text{prior}}$ to hold. We use a fully connected network with positional encoding (Vaswani et al., 2017) to approximate $B_k^n$ and $F_k^n$, see Section S10.1 for details about our implementation. Animated plots of the DSB iterations may be found online on our project webpage[3]. In Figure 3, we illustrate the benefits of DSB over classical SGM. We fix $f(x) = -\alpha x$ and choose $p_{\text{prior}} = \mathcal{N}(0, \sigma_{\text{data}}^2 \mathbf{I})$, hence $\alpha = 1/\sigma_{\text{data}}^2$ where $\sigma_{\text{data}}^2$ is the variance of the dataset. We let $N = 20$ and $\gamma_k = 0.01$,

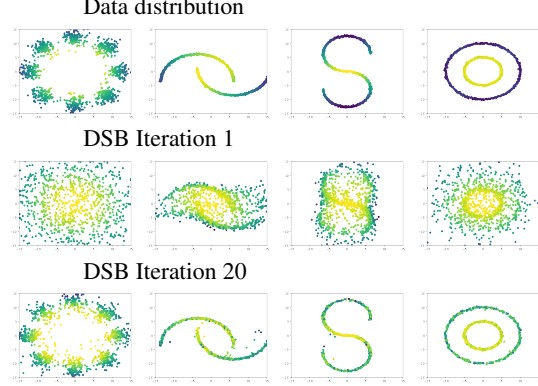

Figure 3: Data distributions $p_{\text{data}}$ vs distribution at $t = 0$ for $T = 0.2$ after 1 and 20 DSB iterations.

*i.e.* $T = 0.2$. Since $T$ is small, we do not have $p_N \approx p_{\text{prior}}$ and the reverse-time process obtained after the first DSB iteration (corresponding to original SGM methods) does not yield a satisfactory generative model. However, multiple iterations of DSB improve the quality of the synthesis.

**Generative modeling.** DSB is the first practical algorithm for approximating the solution to the SB problem in high dimension ($d = 3072$ for CelebA). Whilst our implementation does not yet compete with state-of-the-art methods, we show promising results with fewer diffusion steps compared to initial SGMs (Song and Ermon, 2019) and demonstrate its performance on MNIST (LeCun and Cortes, 2010) and CelebA (Liu et al., 2015).

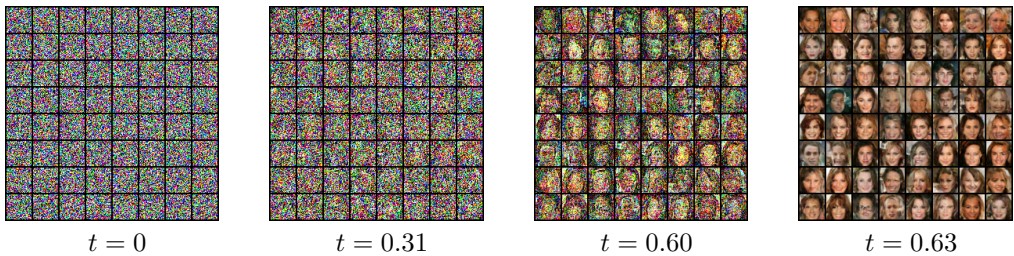

$t = 0$    $t = 0.31$    $t = 0.60$    $t = 0.63$

Figure 4: Generative model for CelebA $32 \times 32$ after 10 DSB iterations with $N = 50$ ($T = 0.63$)

A reduced U-net architecture based on Nichol and Dhariwal (2021) is used to approximate $B_k^n$ and $F_k^n$. Further details are given in Section S10.2. Our method is validated on downscaled CelebA in Figure 4. Figure 5 illustrates qualitative improvement over 8 DSB iterations with as few as $N = 12$ diffusion steps. Note, as shown in Section S10.2, we obtain better results with higher $N$ yet still significantly fewer steps than in the original SGM procedures (Song and Ermon, 2020, 2019) which use $N = 100$. Figure 6 illustrates how the sample quality, measured quantitatively in terms of Fréchet

---

[3]https://vdeborto.github.io/publication/schrodinger_bridge/

Inception Distance (FID) (Heusel et al., 2017), improves with the number of DSB iterations for various numbers of steps $N$.

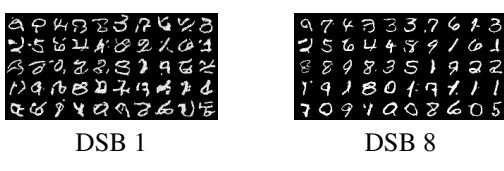

DSB 1          DSB 8

Figure 5: Generated samples ($N = 12$)

Figure 6: FID vs DSB Iterations.

**Dataset interpolation.** Schrödinger bridges not only allow us to reduce the number of steps in SGM methods but also enable flexibility in the choice of the prior density $p_{\text{prior}}$, which is not necessarily Gaussian contrary to previous works on SGM. In particular, our approach is still valid if $p_{\text{prior}}$ is any other data distribution $p'_{\text{data}}$. In this case DSB converges towards a bridge between $p_{\text{data}}$ and $p'_{\text{data}}$, see Figure 7. These experiments pave the way towards high-dimensional optimal transport between arbitrary data distributions.

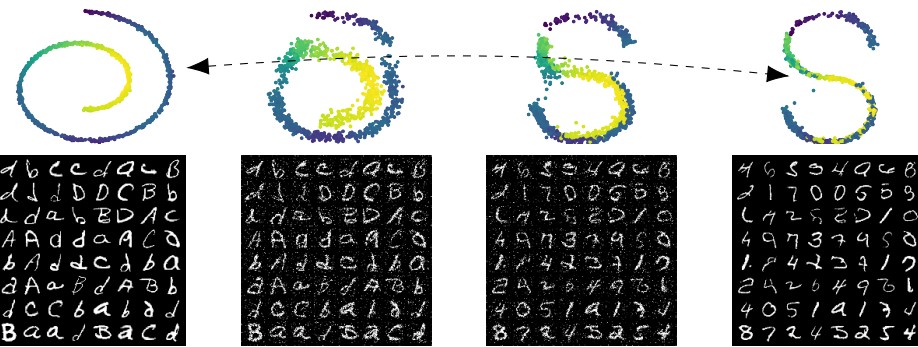

Figure 7: First row: Swiss-roll to S-curve (2D). Iteration 9 of DSB with $T = 1$ ($N = 50$). From left to right: $t = 0, 0.4, 0.6, 1$. Second row: EMNIST (Cohen et al., 2017) to MNIST. Iteration 10 of DSB with $T = 1.5$ ($N = 30$). From left to right: $t = 0, 0.4, 1.25, 1.5$.

## 5 Discussion

Score-based generative modeling (SGM) may be viewed as the first stage of solving a Schrödinger bridge problem. Building on this interpretation, we developed a novel methodology, the Diffusion Schrödinger Bridge (DSB), that extends initial SGM approaches and allows one to perform generative modeling with fewer diffusion steps. DSB complements recent techniques to speed up existing SGM methods that rely on either different noise schedules (Nichol and Dhariwal, 2021; San-Roman et al., 2021; Watson et al., 2021), alternative discretizations (Jolicoeur-Martineau et al., 2021a) or knowledge distillation (Luhman and Luhman, 2021). Additionally, as the solution of the Schrödinger problem is a diffusion, it is possible as in Song et al. (2021, Section 4.3) to obtain an equivalent neural ordinary differential equation that admits the same marginals as the diffusion but enables exact likelihood computation, see Section S8.3. Even though the final time $T > 0$ within DSB can be arbitrarily small, we observed that this has limits as choosing $T$ too close to 0 decreases the quality of the generative models. One reason for this behavior is that if the endpoint of the original forward process is too far from the target distribution $p_{\text{prior}}$, then learning the score around the support of $p_{\text{prior}}$ is challenging even for DSB. From a theoretical point of view, we have provided quantitative convergence results for SGM methods and derived new state-of-the-art convergence bounds for IPF as well as novel monotonicity results. We have demonstrated DSB on generative modeling and data interpolation tasks. Finally, although this work was motivated by generative modeling, DSB is much more widely applicable as it can be thought of as the continuous state-space counterpart of the celebrated Sinkhorn algorithm (Cuturi, 2013; Peyré and Cuturi, 2019). For example, DSB could be used to solve multi-marginal Schrödinger bridges problems (Di Marino and Gerolin, 2020), compute Wasserstein barycenters, find the minimizers of entropy-regularized Gromov–Wasserstein problems (Mémoli, 2011) or perform domain adaptation in continuous state-spaces.

## Acknowledgments and Disclosure of Funding

Valentin De Bortoli and Arnaud Doucet are supported by the EPSRC CoSInES (COmputational Statistical INference for Engineering and Security) grant EP/R034710/1, James Thornton by the OxWaSP CDT through grant EP/L016710/1 and Jeremy Heng by the CY Initiative of Excellence (grant "Investissements d'Avenir" ANR-16-IDEX-0008). Computing resources were provided through the Google Cloud research credits programme. Arnaud Doucet also acknowledges support from the UK Defence Science and Technology Laboratory (DSTL) and EPSRC under grant EP/R013616/1. This is part of the collaboration between US DOD, UK MOD and UK EPSRC under the Multidisciplinary University Research Initiative. We thank Marcel Nutz for pointing out a mistake in an earlier version of the paper.

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
