# Diffusion Schrödinger Bridge with Applications to Score-Based Generative Modeling
## SUPPLEMENTARY DOCUMENT

**Valentin De Bortoli**
Department of Statistics,
University of Oxford, UK

**James Thornton**
Department of Statistics,
University of Oxford, UK

**Jeremy Heng**
ESSEC Business School,
Singapore

**Arnaud Doucet**
Department of Statistics,
University of Oxford, UK

## S1   Organization of the supplementary

The supplementary is organized as follows. We define our notation in Section S2. In Section S3, we prove Theorem 1 and draw links between our approach of SGM and existing works. We recall the classical formulation of IPF, prove Proposition 2 and draw links with autoencoders in Section S4. In Section S5 we present alternative variational formulas for Algorithm 1 and prove Proposition 3. We gather the proofs of our theoretical study of Schrödinger bridges (Proposition 4 and Proposition 5) in Section S6. A quantitative study of IPF with Gaussian targets and reference measure is presented in Section S7. In particular, we show that the convergence rate of IPF is geometric in this case. In Section S8 we study the links between continuous-time and discrete-time IPF and prove Proposition 6. We also provide details on the likelihood computation of generative models obtained with Schrödinger bridges. We detail training techniques to improve training times in Section S9 then present architecture details and additional experiments in Section S10.

## S2   Notation

For ease of reading in this section we recall and detail some of the notation introduced in Section 1. For any measurable space $(\mathsf{E}, \mathcal{E})$, we denote by $\mathscr{P}(\mathsf{E})$ the space of probability measures over $\mathsf{E}$. For any $\ell \in \mathbb{N}$, we also denote $\mathscr{P}_\ell = \mathscr{P}((\mathbb{R}^d)^\ell)$. For any $\pi \in \mathscr{P}(\mathsf{E})$ and Markov kernel $\mathsf{K} : \mathsf{E} \times \mathcal{F} \to [0, 1]$ where $(\mathsf{F}, \mathcal{F})$ is a measurable space, we define $\pi \mathsf{K} \in \mathscr{P}(\mathsf{F})$ such that for any $\mathsf{A} \in \mathcal{F}$ we have $\pi \mathsf{K}(\mathsf{A}) = \int_\mathsf{E} \mathsf{K}(x, \mathsf{A}) \mathrm{d}\pi(x)$. If $\mathsf{E} = \mathcal{C}$ then for any $\mathbb{P} \in \mathscr{P}(\mathsf{E})$ and $s, t \in [0, T]$, we denote by $\mathbb{P}_{s,t}$ the marginals of $\mathbb{P}$ at time $s$ and $t$. In addition, we denote by $\mathbb{P}_{|s,t}$ the disintegration Markov kernel given by the mapping $\omega \mapsto (\omega(s), \omega(t))$, see Section S4.1 for a definition. In particular, we have $\mathbb{P} = \mathbb{P}_{s,t} \mathbb{P}_{|s,t}$. All defined mappings are considered to be measurable unless stated otherwise.

For any $\mathbb{P} \in \mathscr{P}(\mathcal{C})$ we define $\mathbb{P}^R$ the reverse-time measure, *i.e.* for any $\mathsf{A} \in \mathcal{B}(\mathcal{C})$ we have $\mathbb{P}^R(\mathsf{A}) = \mathbb{P}(\mathsf{A}^R)$ where $\mathsf{A}^R = \{t \mapsto \omega(T - t) : \omega \in \mathsf{A}\}$. We say that $\mathbb{P} \in \mathscr{P}(\mathcal{C})$ is *associated with a diffusion* if it solves the corresponding martingale problem. More precisely, $\mathbb{P} \in \mathscr{P}(\mathcal{C})$ is associated with $\mathrm{d}\mathbf{X}_t = b(t, \mathbf{X}_t)\mathrm{d}t + \sqrt{2}\mathrm{d}\mathbf{B}_t$ for $b : [0, T] \times \mathbb{R}^d \to \mathbb{R}^d$ measurable if for any $v \in \mathrm{C}_c^2(\mathbb{R}^d, \mathbb{R})$, $(M_t^v)_{t \in [0, T]}$ is a $\mathbb{P}$-local martingale, where for any $t \in [0, T]$

$$M_t^v = v(\mathbf{X}_t) - \int_0^t \mathcal{A}_s(v)(\mathbf{X}_s)\mathrm{d}s \tag{S1}$$

with for any $v \in \mathrm{C}^2(\mathbb{R}^d, \mathbb{R})$, $t \in [0, t]$ and $x \in \mathbb{R}^d$

$$\mathcal{A}_t(v)(x) = \langle b(t, x), \nabla v(x) \rangle + \Delta v(x).$$

35th Conference on Neural Information Processing Systems (NeurIPS 2021).

We refer to Revuz and Yor (1999) for a rigorous treatment of local martingales. Note that (S1) uniquely defines $\mathbb{P}_{t|s}$ for any $s, t \in [0, T]$ with $t \geq s$. Hence $\mathbb{P}$ is uniquely defined up to $\mathbb{P}_0$.

In some cases, we say that $\mathbb{P} \in \mathscr{P}(\mathcal{C})$ is *associated with a diffusion* if it solves the corresponding martingale problem with initial condition. More precisely, $\mathbb{P} \in \mathscr{P}(\mathcal{C})$ is associated with $\mathrm{d}\mathbf{X}_t = b(t, \mathbf{X}_t)\mathrm{d}t + \sqrt{2}\mathrm{d}\mathbf{B}_t$ and $\mathbf{X}_0 \sim \mu_0 \in \mathscr{P}(\mathbb{R}^d)$ if it solves the martingale problem and $\mathbb{P}_0 = \mu_0$. Note that in this case $\mathbb{P}$ is uniquely defined.

Finally, for any measurable space $(\mathsf{E}, \mathcal{E})$ and $\mu, \nu \in \mathscr{P}(\mathsf{E})$ we recall that the Jeffrey's divergence is given by $\mathrm{J}(\mu, \nu) = \mathrm{KL}(\mu|\nu) + \mathrm{KL}(\nu|\mu)$.

## S3 Time-reversal and existing work

Before giving the proof of Theorem 1 we start by deriving estimates on the logarithmic derivatives of the density of the Ornstein-Ulhenbeck process given growth conditions on the initial density in Section S3.1. Note that our estimates are uniform w.r.t. the time variable. We give the proof of Theorem 1 in Section S3.2. Finally, we draw links with existing works in Section S3.3.

### S3.1 Estimates for logarithmic derivatives

We start by recalling the following multivariate Faa di Bruno's formula and a useful technical lemma. Then in Section S3.1.1 we derive bounds for the logarithmic derivatives which are non-vacuous for small times. In Section S3.1.2 we derive bounds for the logarithmic derivatives which are non-vacuous for large times. We combine them in Section S3.1.3.

For any $\alpha \in \mathbb{N}^d$ we denote $|\alpha| = \sum_{i=1}^d \alpha_i$ and $\alpha! = \prod_{i=1}^d \alpha_i!$. If $f : \mathbb{R}^d \to \mathbb{R}$ is $m$-differentiable with $m \in \mathbb{N}$, then for any $\lambda \in \mathbb{N}^d$ with $|\lambda| \leq m$ we denote for any $x \in \mathbb{R}^d$, $\partial_\lambda f(x) = \partial_1^{\lambda_1} \ldots \partial_d^{\lambda_d} f(x)$. Similarly to Constantine and Savits (1996), we define $\prec$ the order on $\mathbb{N}^d$ such that for any $\lambda^1, \lambda^2 \in \mathbb{N}^d$, $\lambda^1 \prec \lambda^2$ if $|\lambda^1| < |\lambda^2|$ or $|\lambda^1| = |\lambda^2|$ and there exists $j \in \{1, \ldots, d\}$ such that $\lambda_j^1 < \lambda_j^2$ and for any $i \in \{1, \ldots, j\}, \lambda_i^1 = \lambda_i^2$.

**Proposition S1.** *Let* $\mathsf{U} \subset \mathbb{R}$ *open,* $N \in \mathbb{N}$, $f \in \mathrm{C}^N(\mathsf{U}, \mathbb{R})$, $g \in \mathrm{C}^N(\mathbb{R}^d, \mathsf{U})$ *and* $h = f \circ g$. *Then for any* $\lambda \in \mathbb{N}^d$ *with* $|\lambda| \leq N$ *and* $x \in \mathbb{R}^d$ *we have*

$$\partial_\lambda h(x) = \sum_{k,s=1}^{|\lambda|} \sum_{p_s(\lambda,k)} f^{(k)}(g(x))\lambda! \prod_{j=1}^s \partial_{\ell_j} g(x)^{m_j}/(m_j!\ell_j!^{m_j}),$$

*with*

$$p_s(\lambda, k) = \{\{\ell_i\}_{i=1}^s \in (\mathbb{N}^d)^s, \ \{m_i\}_{i=1}^s \in \mathbb{N}^s : \ \ell_1 \prec \cdots \prec \ell_s, \ \sum_{i=1}^s m_i = k, \ \sum_{i=1}^s m_i \ell_i = \lambda\}.$$

*Proof.* The proposition is a direct application of Constantine and Savits (1996). $\qquad\square$

From this multivariate Faa di Bruno formula we derive the following lemma drawing links between exponential and logarithmic derivatives.

**Lemma S2.** *Let* $N \in \mathbb{N}$, $g_1 \in \mathrm{C}^N(\mathbb{R}^d, \mathbb{R})$, $g_2 \in \mathrm{C}^N(\mathbb{R}^d, (0, +\infty))$, $h_1 = \exp[g_1]$ *and* $h_2 = \log(g_2)$. *Then for any* $\lambda \in \mathbb{N}^d$ *with* $|\lambda| \leq N$ *let* $c_{d,\lambda} = \sum_{k=1}^{|\lambda|} d^k$ *and the following hold:*

(a) *There exists* $\mathrm{P}_{\lambda,\exp}$ *a real polynomial with* $c_{d,\lambda}$ *variables such that for any* $x \in \mathbb{R}^d$

$$\partial_\lambda h_1(x) = \mathrm{P}_{\lambda,\exp}((\partial_\ell g_1(x))_{|\ell| \leq |\lambda|})h_1(x).$$

(b) *There exists* $\mathrm{P}_{\lambda,\log}$ *a real polynomial with* $c_{d,\lambda}$ *variables such that for any* $x \in \mathbb{R}^d$

$$\partial_\lambda h_2(x) = \mathrm{P}_{\lambda,\log}((\partial_\ell g_2(x)/g_2(x))_{|\ell| \leq |\lambda|}).$$

*Proof.* The proof of (a) is a direct application of Proposition S1 upon noting that for any $k \in \mathbb{N}$, $f^{(k)} = \exp$ if $f = \exp$. Similarly, the proof of (b) is a direct application of Proposition S1 upon noting that, in the case where $f = \log$, for any $k \in \mathbb{N}$ and $x > 0$, $f^{(k)}(x) = (-1)^{k-1}(k-1)!x^{-k}$ and that for any $s \in \{1, \ldots, |\lambda|\}$ and $(\ell_1, \ldots, \ell_s, m_1, \ldots, m_s) \in p_s(\lambda, k)$ we have $\sum_{i=1}^s m_i = k$. $\qquad\square$

We will also make use of the following technical lemma.

**Lemma S3.** *Let $p \in \mathbb{N}$. Then for any $a \geq 0$, $b > 0$ and $x \in \mathbb{R}^d$ we have*

$$- b\|x\|^{2p} + a\|x\|^{2p-1} \leq -(b/2)\|x\|^{2p} + a(2a/b)^{2p-1}, \tag{S2}$$

$$- b\|x\|^{2p} + a\|x\|^{2p-2} \leq -(b/2)\|x\|^{2p} + a(2a/b)^{p-1}. \tag{S3}$$

*In addition for any $a \geq 0$, $b > 0$ and $x \in \mathbb{R}^d$ we have*

$$-b\|x\|^{2p} + a\|x\|^{2p-1} \leq (2p-1)^{2p-1}(2p)^{-2p}a^{2p}b^{1-2p}.$$

*Proof.* For the first part of the proof, we only prove (S2). The proof of (S3) is similar. Let $a \geq 0$, $b > 0$. For any $x \in \mathbb{R}^d$ with $\|x\| \leq (b/2a)^{-1}$ we have $a\|x\|^{2p-1} \leq a(b/2a)^{-2p+1}$. For any $x \in \mathbb{R}^d$ with $\|x\| \geq (b/2a)^{-1}$ we have $a\|x\|^{2p-1} \leq (b/2)\|x\|^{2p}$. Hence, we get that for any $x \in \mathbb{R}^d$ we have

$$a\|x\|^{2p-1} - b\|x\|^{2p} \leq a(b/2a)^{-2p+1} - (b/2)\|x\|^{2p},$$

which concludes the first part of the proof. For the second part of the proof, remark that the maximum of h : $t \mapsto -bt^{2p} + at^{2p-1}$ is attained for $t^\star = (2p-1)/(2p)(a/b)$. We conclude upon noting that $h(t^\star) = (2p-1)^{2p-1}(2p)^{-2p}a^{2p}b^{1-2p}$. $\qquad\square$

### S3.1.1 Small times estimates

Lemma S2 is key in the following proposition which establishes upper bounds on the logarithmic derivatives of the density of the Ornstein-Ulhenbeck process. In what follows, we define $(p_t)_{t \in [0,T]}$ the density w.r.t. the Lebesgue measure of $\mathbf{X}_t$ satisfying

$$\mathrm{d}\mathbf{X}_t = -\alpha\mathbf{X}_t\mathrm{d}t + \sqrt{2}\mathrm{d}\mathbf{B}_t, \qquad \mathbf{X}_0 \sim p_{\mathrm{data}},$$

with $\alpha \geq 0$. In the rest of this section, $\alpha$ is fixed.

**Proposition S4.** *Let $N \in \mathbb{N}$. Assume that $p_{\mathrm{data}} \in \mathrm{C}^N(\mathbb{R}^d, (0, +\infty))$ is bounded and that for any $\ell \in \{1, \ldots, N\}$ there exist $A_\ell \geq 0$ and $\alpha_\ell \in \mathbb{N}$ such that for any $x \in \mathbb{R}^d$*

$$\|\nabla^\ell \log p_{\mathrm{data}}(x)\| \leq A_\ell(1 + \|x\|^{\alpha_\ell}). \tag{S4}$$

*Then for any $t \geq 0$, $p_t \in \mathrm{C}^N(\mathbb{R}^d, (0, +\infty))$ and for any $\ell \in \{1, \ldots, N\}$, there exist $B_\ell \geq 0$ and $\beta_\ell \in \mathbb{N}$ such that for any $t \geq 0$*

$$\|\nabla^\ell \log p_t(x)\| \leq c_t^{-2\beta_\ell}B_\ell(1 + \int_{\mathbb{R}^d} \|x_0\|^{\beta_\ell} p_{0|t}(x_0|x_t)\mathrm{d}x_0),$$

*with $c_t^2 = \exp[-2\alpha t]$.*

*Proof.* First note that for any $t \geq 0$ and $x_t \in \mathbb{R}^d$ we have

$$p_t(x_t) = \int_{\mathbb{R}^d} p_{\mathrm{data}}(x_0)\mathrm{g}(x_t - c_t x_0)\mathrm{d}x_0, \tag{S5}$$

with for any $\tilde{x} \in \mathbb{R}^d$

$$c_t = \exp[-\alpha t], \qquad \mathrm{g}(\tilde{x}) = (2\pi\sigma_t^2)^{-d/2}\exp[-\|\tilde{x}\|^2/(2\sigma_t^2)], \qquad \sigma_t^2 = (1 - \exp[-2\alpha t])/\alpha.$$

Let $t \geq 0$. We have that $p_t \in \mathrm{C}^N(\mathbb{R}^d, (0, +\infty))$ upon combining the fact that $p_{\mathrm{data}}$ is bounded, (S5) and the dominated convergence theorem. Let $\ell \in \{1, \ldots, N\}$ and $\lambda \in \mathbb{N}^d$ such that $|\lambda| \leq \ell$. Using Lemma S2-(b) we have for any $x_t \in \mathbb{R}^d$

$$\partial_\lambda \log p_t(x_t) = \mathrm{P}_{\lambda,\log}((\partial_m p_t(x_t)/p_t(x_t))_{|m| \leq |\lambda|}). \tag{S6}$$

Using (S5) and the change of variable $z = x_t - c_t x_0$, we have for any $x_t \in \mathbb{R}^d$

$$p_t(x_t) = c_t^{-1} \int_{\mathbb{R}^d} p_{\mathrm{data}}((x_t - z)/c_t)\mathrm{g}(z)\mathrm{d}z.$$

Hence, combining this result, the dominated convergence theorem and Lemma S2-(a) we get that for any $x_t \in \mathbb{R}^d$ and $m \in \mathbb{N}^d$ with $|m| \leq \ell$

$$\partial_m p_t(x_t) = c_t^{-|m|} \int_{\mathbb{R}^d} \partial_m p_{\mathrm{data}}(x_0)\mathrm{g}(x_t - c_t x_0)\mathrm{d}x_0$$

$$= c_t^{-|m|} \int_{\mathbb{R}^d} \mathrm{P}_{m,\exp}((\partial_j \log p_{\mathrm{data}}(x_0))_{|j| \leq |m|})p_{\mathrm{data}}(x_0)\mathrm{g}(x_t - c_t x_0)\mathrm{d}x_0.$$

We conclude the proof upon combining this result, (S4), (S6) and the fact that $c_t \leq 1$. $\qquad\square$

For any $t \geq 0$ and $x_t \in \mathbb{R}^d$ we introduce the infinitesimal generator $\mathcal{A}_{t,x_t} : \mathrm{C}_2(\mathbb{R}^d, \mathbb{R}) \to \mathrm{C}_2(\mathbb{R}^d, \mathbb{R})$ given for any $\varphi \in \mathrm{C}^2(\mathbb{R}^d, \mathbb{R})$ and $x_0 \in \mathbb{R}^d$ by

$$\mathcal{A}_{t,x_t}(\varphi)(x_0) = \langle \nabla_{x_0} \log p_{0|t}(x_0|x_t), \nabla \varphi(x_0) \rangle + \Delta \varphi(x_0) \tag{S7}$$
$$= \langle \nabla \log p_{\mathrm{data}}(x_0), \nabla \varphi(x_0) \rangle + (c_t/\sigma_t^2)\langle x_t - c_t x_0, \nabla \varphi(x_0) \rangle + \Delta \varphi(x_0).$$

Establishing Foster-Lyapunov drift condition for this infinitesimal generator will allow us to derive moment bounds for $x_0 \mapsto p_{0|t}(x_0|x_t)$. We now introduce the Lyapunov functional which will allow us to control these moments. For any $p \in \mathbb{N}$, $t > 0$ and $x_t \in \mathbb{R}^d$, let $V_{p,t,x_t} : \mathbb{R}^d \to [1, +\infty)$ given for any $x_0 \in \mathbb{R}^d$ by

$$V_{p,t,x_t}(x_0) = 1 + \|x_0 - x_t/c_t\|^{2p}, \qquad c_t = \exp[-\alpha t].$$

**Proposition S5.** *Assume $p_{\mathrm{data}} \in \mathrm{C}^1(\mathbb{R}^d, \mathbb{R})$ and that there exist $\mathtt{m}_0 > 0$, $d_0, C_0 \geq 0$ such that for any $x_0 \in \mathbb{R}^d$ we have*

$$\langle x_0, \nabla \log p_{\mathrm{data}}(x_0) \rangle \leq -\mathtt{m}_0 \|x_0\|^2 + d_0 \|x_0\|, \quad \|\nabla \log p_{\mathrm{data}}(x_0)\| \leq C_0(1 + \|x_0\|). \tag{S8}$$

*Then for any $t > 0$, $x_t \in \mathbb{R}^d$ and $p \in \mathbb{N}$ there exist $\beta_p \in \mathbb{N}$, $a_p > 0$ and $b_p \geq 0$ (independent of $t$ and $x_t$) such that for any $x_0 \in \mathbb{R}^d$ we have*

$$\mathcal{A}_{t,x_t}(V_{p,t,x_t})(x_0) \leq -a_p V_{p,t,x_t}(x_0) + b_p(1 + \|x_t/c_t\|^{\beta_p}),$$

*with $\beta_p = 2p$.*

*Proof.* Let $t \geq 0$, $x_0, x_t \in \mathbb{R}^d$ and $p \in \mathbb{N}$. First, we have for any $x_0 \in \mathbb{R}^d$

$$V_{p,t,x_t}(x_0) = \|x_0 - x_t/c_t\|^{2p}, \quad \nabla V_{p,t,x_t}(x_0) = 2p(x_0 - x_t/c_t)\|x_0 - x_t/c_t\|^{2(p-1)}, \tag{S9}$$
$$\Delta V_{p,t,x_t}(x_0) = 2p(2p-1)\|x_0 - x_t/c_t\|^{2(p-1)}.$$

Second, using Lemma S3, the Cauchy-Schwarz inequality and (S8), we have for any $x_0 \in \mathbb{R}^d$

$$\langle \nabla \log p_{\mathrm{data}}(x_0), x_0 - x_t/c_t \rangle \leq -\mathtt{m}_0 \|x_0\|^2 + d_0 \|x_0\| + \|\nabla \log p_{\mathrm{data}}(x_0)\| \|x_t/c_t\|$$
$$\leq -\mathtt{m}_0 \|x_0 - x_t/c_t\|^2 + 2\mathtt{m}_0 \|x_0\| \|x_t\|/c_t + C_0(1 + \|x_0\|)\|x_t\|/c_t$$
$$+ d_0 \|x_0 - x_t/c_t\| + d_0 \|x_t\|/c_t + \mathtt{m}_0 \|x_t\|^2/c_t^2$$
$$\leq -\mathtt{m}_0 \|x_0 - x_t/c_t\|^2 + \{(2\mathtt{m}_0 + C_0)\|x_t\|/c_t + d_0\}\|x_0 - x_t/c_t\|$$
$$+ (3\mathtt{m}_0 + C_0)\|x_t\|^2/c_t^2 + (C_0 + d_0)\|x_t\|/c_t.$$

Combining this result and (S9), we have for any $x_0 \in \mathbb{R}^d$

$$\langle \nabla \log p_{\mathrm{data}}(x_0), \nabla V_{p,t,x_t}(x_0) \rangle$$
$$\leq -2p\mathtt{m}_0 \|x_0 - x_t/c_t\|^{2p} + 2p\{(2\mathtt{m}_0 + C_0)\|x_t\|/c_t + d_0\}\|x_0 - x_t/c_t\|^{2p-1}$$
$$+ 2p((3\mathtt{m}_0 + C_0)\|x_t\|^2/c_t^2 + (C_0 + d_0)\|x_t\|/c_t)\|x_0 - x_t/c_t\|^{2p-2}.$$

Combining this result with (S7) and the fact that for any $x_0 \in \mathbb{R}^d$, $(c_t/\sigma_t^2)\langle x_t - c_t x_0, \nabla V_{p,t,x_t}(x_0) \rangle \leq 0$, we get that for any $x_0 \in \mathbb{R}^d$

$$\mathcal{A}_{t,x_t}(V_{p,t,x_t})(x_0) \leq -2p\mathtt{m}_0 \|x_0 - x_t/c_t\|^{2p} + 2p\{(2\mathtt{m}_0 + C_0)\|x_t\|/c_t + d_0\}\|x_0 - x_t/c_t\|^{2p-1}$$
$$+ 2p((3\mathtt{m}_0 + C_0)\|x_t\|^2/c_t^2 + (C_0 + d_0)\|x_t\|/c_t)\|x_0 - x_t/c_t\|^{2p-2}.$$

Using Lemma S3 there exist $\beta_p \in \mathbb{N}$, $a_p > 0$ and $b_p \geq 0$ (independent of $x_t$ and $t$) such that for any $x_0 \in \mathbb{R}^d$ we have

$$\mathcal{A}_{t,x_t}(V_{p,t,x_t})(x_0) \leq -a_p V_{p,t,x_t}(x_0) + b_p(1 + (\|x_t\|/c_t)^{\beta_p}),$$

which concludes the proof. $\qquad \square$

Using this Foster-Lyapunov drift we are now ready to bound the moments of $x_0 \mapsto p_{0|t}(x_0|x_t)$.

**Proposition S6.** *Assume that $p_{\mathrm{data}} \in \mathrm{C}^2(\mathbb{R}^d, \mathbb{R})$ and that there exist $\mathtt{m}_0 > 0$, $d_0, C_0 \geq 0$ such that for any $x_0 \in \mathbb{R}^d$ we have*

$$\langle x_0, \nabla \log p_{\mathrm{data}}(x_0) \rangle \leq -\mathtt{m}_0 \|x_0\|^2 + d_0\|x_0\|, \quad \|\nabla \log p_{\mathrm{data}}(x_0)\| \leq C_0(1 + \|x_0\|).$$

*Then, for any $p \in \mathbb{N}$ there exist $C_p \geq 0$ and $\beta_p \in \mathbb{N}$ such that for any $t \geq 0$ and $x_t \in \mathbb{R}^d$*

$$\int_{\mathbb{R}^d} \|x_0\|^p \, p(x_0|x_t) \mathrm{d}x_0 \leq C_p c_t^{-2\beta_p}(1 + \|x_t\|^{\beta_p}), \tag{S10}$$

*with $c_t^2 = \exp[-2\alpha t]$ and $\beta_p = p$.*

*Proof.* Let $t \geq 0$ and $x_t \in \mathbb{R}^d$. Using (Ikeda and Watanabe, 1989, Theorem 2.3, Theorem 3.1), Proposition S5 and (Meyn and Tweedie, 1993, Theorem 2.1) for any $x \in \mathbb{R}^d$, there exists a unique strong solution $(\mathbf{X}_u^x)_{u \geq 0}$ such that $\mathbf{X}_0^x \sim \delta_x$ and

$$\mathrm{d}\mathbf{X}_u^x = \nabla \log p_{0|t}(\mathbf{X}_u^x|x_t)\mathrm{d}u + \sqrt{2}\mathrm{d}\mathbf{B}_u.$$

Using (Leha and Ritter, 1984, Theorem 5.19) we get that $\{(\mathbf{X}_u^x)_{u \geq 0} \, : \, x \in \mathbb{R}^d\}$ is associated with a Feller semi-group. In addition, we have that for any $f \in \mathrm{C}_c^2(\mathbb{R}^d)$, $\int_{\mathbb{R}^d} \mathcal{A}_{t,x_t}(f)(x_0)p_{0|t}(x_0|x_t)\mathrm{d}x_0 = 0$. Therefore, using (Revuz and Yor, 1999, Proposition 1.5) and (Ethier and Kurtz, 1986, Theorem 9.17) we get that the probability distribution with density $x_0 \mapsto p_{0|t}(x_0|x_t)$ is an invariant distribution for the semi-group associated with $\{(\mathbf{X}_u^x)_{u \geq 0} \, : \, x \in \mathbb{R}^d\}$. Therefore, using Proposition S5 and (Meyn and Tweedie, 1993, Theorem 4.6) we get that for any $p \in \mathbb{N}$

$$\int_{\mathbb{R}^d}(1 + \|x_0 - c_t^{-1}x_t\|^{2p})p_{0|t}(x_0|x_t)\mathrm{d}x_0 \leq b_p(1 + \|x_t/c_t\|^{\beta_p})/a_p$$

which concludes the proof upon using that $c_t \leq 1$ and Jensen's inequality. $\qquad\square$

### S3.1.2  Large times estimates

In Proposition S6, the bound in (S10) goes to $+\infty$ as $t \to +\infty$ since $\lim_{t \to +\infty} c_t^{-1} = +\infty$ (if $\alpha > 0$). This does not yield any degeneracy in our setting since we consider a fixed time horizon $T > 0$. However, we can improve the result by deriving another bound which is bounded at $t \to +\infty$ but explodes as $t \to 0$. In this section we assume that $h : \, u \mapsto (\exp[u] - 1)/u$ is extended to 0 by continuity with $h(0) = 1$.

The following proposition is the equivalent of Proposition S4 with a bound which explodes for $t \to 0$ instead of $t \to +\infty$. Note that contrary to Proposition S4 we do not require any differentiability condition the initial distribution $p_{\mathrm{data}}$.

**Proposition S7.** *Let $N \in \mathbb{N}$. Assume that $p_{\mathrm{data}} \in \mathrm{C}^0(\mathbb{R}^d, (0, +\infty))$ is bounded. Then for any $t \geq 0$, $p_t \in \mathrm{C}^N(\mathbb{R}^d, (0, +\infty))$ and for any $\ell \in \{1, \ldots, N\}$, there exist $B_\ell \geq 0$ and $\beta_\ell \in \mathbb{N}$ such that for any $t \geq 0$*

$$\|\nabla^\ell \log p_t(x)\| \leq \sigma_t^{-\beta_\ell} B_\ell(1 + \int_{\mathbb{R}^d} \|x_t - c_t x_0\|^{\beta_\ell} \, p_{0|t}(x_0|x_t)\mathrm{d}x_0)$$
$$\leq \sigma_t^{-\beta_\ell} B_\ell(1 + \int_{\mathbb{R}^d} \|x_t - x_0\|^{\beta_\ell} \, q_{0|t}(x_0|x_t)\mathrm{d}x_0).$$

*with $\sigma_t^2 = (1 - \exp[-2\alpha t])/\alpha$ and for any $\tilde{x} \in \mathbb{R}^d$*

$$q_{0|t}(x_0|x_t) = p_{\mathrm{data}}(x_0/c_t)\mathrm{g}(x_t - x_0)/\int_{\mathbb{R}^d} p_{\mathrm{data}}(x_0/c_t)\mathrm{g}(x_t - x_0)\mathrm{d}x_0,$$
$$\mathrm{g}(\tilde{x}) = (2\pi\sigma_t^2)\exp[-\|\tilde{x}\|^2/(2\sigma_t^2)].$$

*Proof.* First note that for any $t \geq 0$ and $x_t \in \mathbb{R}^d$ we have

$$p_t(x_t) = \int_{\mathbb{R}^d} p_{\mathrm{data}}(x_0)\mathrm{g}(x_t - c_t x_0)\mathrm{d}x_0, \tag{S11}$$

with

$$c_t = \exp[-\alpha t], \qquad \mathrm{g}(\tilde{x}) = (2\pi\sigma_t^2)^{-d/2}\exp[-\|\tilde{x}\|^2/(2\sigma_t^2)], \qquad \sigma_t^2 = (1 - \exp[-2\alpha t])/\alpha.$$

Let $t \geq 0$. We have $p_t \in \mathrm{C}^N(\mathbb{R}^d, (0, +\infty))$ upon combining the fact that $p_{\mathrm{data}}$ is bounded, (S11) and the dominated convergence theorem. Let $\ell \in \{0, \ldots, N\}$ and $\lambda \in \mathbb{N}^d$ such that $|\lambda| \leq \ell$. Using Lemma S2-(b) we have for any $x_t \in \mathbb{R}^d$

$$\partial_\lambda \log p_t(x_t) = \mathrm{P}_{\lambda,\log}((\partial_m p_t(x_t)/p_t(x_t))_{|m| \leq |\lambda|}).$$

For any $m \in \mathbb{N}^d$ with $|m| \leq |\lambda|$, using the dominated convergence theorem, there exist $C_m \geq 0$ and $\beta_m \in \mathbb{N}$ such that for any $x_t \in \mathbb{R}^d$ we have

$$|\partial_m p_t(x_t)| \leq C_m \sigma_t^{-2\beta_m} \int_{\mathbb{R}^d} (1 + \|x_t - c_t x_0\|^{\beta_m}) p_{\text{data}}(x_0) \mathrm{g}(x_t - c_t x_0) \mathrm{d}x_0,$$

which concludes the proof. $\qquad\square$

For any $t \geq 0$ and $x_t \in \mathbb{R}^d$ we introduce the infinitesimal generator $\tilde{\mathcal{A}}_{t,x_t} : \mathrm{C}_2(\mathbb{R}^d, \mathbb{R}) \to \mathrm{C}_2(\mathbb{R}^d, \mathbb{R})$ given for any $\varphi \in \mathrm{C}^2(\mathbb{R}^d, \mathbb{R})$ and $x_0 \in \mathbb{R}^d$ by

$$\tilde{\mathcal{A}}_{t,x_t}(f)(x_0) = \langle \nabla \log q_{0|t}(x_0|x_t), \nabla \varphi(x_0) \rangle + \Delta \varphi(x_0)$$
$$= c_t^{-1} \langle \nabla \log p_{\text{data}}(x_0/c_t), \nabla \varphi(x_0) \rangle + \sigma_t^{-2} \langle x_t - x_0, \nabla \varphi(x_0) \rangle + \Delta \varphi(x_0).$$

For any $p \in \mathbb{N}$, let $V_p : \mathbb{R}^d \to [1, +\infty)$ given for any $x_0 \in \mathbb{R}^d$ by

$$V_p(x_0) = 1 + \|x_0\|^{2p}.$$

The following proposition is the counterpart to Proposition S5.

**Proposition S8.** *Assume that $p_{\text{data}} \in \mathrm{C}^1(\mathbb{R}^d, \mathbb{R})$ and that there exist $\mathtt{m}_0 > 0$, $d_0 \geq 0$ such that for any $x_0 \in \mathbb{R}^d$ we have*

$$\langle x_0, \nabla \log p_{\text{data}}(x_0) \rangle \leq -\mathtt{m}_0 \|x_0\|^2 + d_0 \|x_0\|. \tag{S12}$$

*Then for any $t > 0$, $x_t \in \mathbb{R}^d$ and $p \in \mathbb{N}$ there exist $\beta_p \in \mathbb{N}$, $a_p > 0$ and $b_p \geq 0$ (independent of $t$ and $x_t$) such that for any $x_0 \in \mathbb{R}^d$ we have*

$$\tilde{\mathcal{A}}_{t,x_t}(V_p)(x_0) \leq -a_p \sigma_t^{-2} V_p(x_0) + b_p(1 + \|x_t/\sigma_t^2\|^{\beta_p}),$$

*with $\beta_p = 2p$.*

*Proof.* Let $t \geq 0$, $x_0, x_t \in \mathbb{R}^d$ and $p \in \mathbb{N}$. First, we have for any $x_0 \in \mathbb{R}^d$

$$V_p(x_0) = 1 + \|x_0\|^{2p}, \qquad \nabla V_p(x_0) = 2p \|x_0\|^{2(p-1)} x_0, \qquad \Delta V_p(x_0) = 2p(2p-1) \|x_0\|^{2(p-1)}.$$

Using this result, (S12) and Lemma S3, we get that for any $x_0 \in \mathbb{R}^d$

$$2p \langle \nabla \log p_{\text{data}}(x_0/c_t), x_0/c_t \rangle \|x_0\|^{2(p-1)} \leq 2p c_t^{-1}(-\mathtt{m}_0 \|x_0\|^{2p}/c_t + d_0 \|x_0\|^{2p-1})$$
$$\leq c_t^{-1}(2p-1)^{2p-1}(2p)^{1-2p}(\mathtt{m}_0/c_t)^{1-2p} d_0^{2p}.$$

Combining this result and the fact that $c_t \leq 1$, there exists $d_p \geq 0$ (independent from $t$ and $x_t$) such that for any $x_0 \in \mathbb{R}^d$

$$2p \langle \nabla \log p_{\text{data}}(x_0/c_t), x_0/c_t \rangle \|x_0\|^{2(p-1)} \leq d_p. \tag{S13}$$

In addition, we have for any $x_0 \in \mathbb{R}^d$

$$(2p/\sigma_t^2) \langle x_0, x_t - x_0 \rangle \|x_0\|^{2(p-1)} + 2p(2p-1) \|x_0\|^{2(p-1)}$$
$$\leq -(2p/\sigma_t^2) \|x_0\|^{2p} + (2p/\sigma_t^2) \|x_0\|^{2p-1} \|x_t\| + 2p(2p-1) \|x_0\|^{2p-1} + 2p(2p-1).$$

Combining this result and (S13) we have for any $x_0 \in \mathbb{R}^d$

$$\tilde{\mathcal{A}}_{t,x_t}(V_p)(x_0)$$
$$\leq -(2p/\sigma_t^2) \|x_0\|^{2p} + (2p/\sigma_t^2) \|x_0\|^{2p-1} \|x_t\| + 2p(2p-1) \|x_0\|^{2p-1} + 2p(2p-1) + d_p.$$

We conclude upon using Lemma S3. $\qquad\square$

The next proposition is the counterpart of Proposition S6.

**Proposition S9.** *Assume that $p_{\text{data}} \in \mathrm{C}^2(\mathbb{R}^d, \mathbb{R})$ and that there exist $\mathtt{m}_0 > 0$, $d_0 \geq 0$ such that for any $x_0 \in \mathbb{R}^d$ we have*

$$\langle x_0, \nabla \log p_{\text{data}}(x_0) \rangle \leq -\mathtt{m}_0 \|x_0\|^2 + d_0 \|x_0\|.$$

*Then, for any $p \in \mathbb{N}$ there exist $C_p \geq 0$ and $\beta_p \in \mathbb{N}$ such that for any $t \in \geq 0$ and $x_t \in \mathbb{R}^d$*

$$\int_{\mathbb{R}^d} \|x_t - x_0\|^p q_{0|t}(x_0|x_t) \mathrm{d}x_0 \leq C_p \sigma_t^{-2\beta_p}(1 + \|x_t\|^{\beta_p}),$$

*with $\sigma_t^2 = (1 - \exp[-2\alpha t])/\alpha$ and $\beta_p = p$.*

*Proof.* The proof is similar to the one of Proposition S6. $\qquad\square$

### S3.1.3 Uniform in time logarithmic derivatives estimates

In this section we combine the results of Section S3.1.2 and Section S3.1.1 to establish uniform in time estimates for the logarithmic derivatives of the density of the Ornstein-Uhlenbeck diffusion.

**Theorem S10.** *Let $N \in \mathbb{N}$ with $N \geq 2$. Assume that $p_{\text{data}} \in \mathrm{C}^N(\mathbb{R}^d, \mathbb{R})$ and that there exist $\mathtt{m}_0 > 0$, $d_0, C_0 \geq 0$ such that for any $x_0 \in \mathbb{R}^d$ we have*

$$\langle x_0, \nabla \log p_{\text{data}}(x_0) \rangle \leq -\mathtt{m}_0 \|x_0\|^2 + d_0 \|x_0\|, \quad \|\nabla \log p_{\text{data}}(x_0)\| \leq C_0(1 + \|x_0\|).$$

*In addition, assume that $p_{\text{data}}$ is bounded and that for any $\ell \in \{1, \ldots, N\}$ there exist $A_\ell \geq 0$ and $\alpha_\ell \in \mathbb{N}$ such that for any $x_0 \in \mathbb{R}^d$*

$$\|\nabla^\ell \log p_{\text{data}}(x_0)\| \leq A_\ell(1 + \|x_0\|^{\alpha_\ell}). \tag{S14}$$

*Then for any $t \geq 0$, $p_t \in \mathrm{C}^N(\mathbb{R}^d, (0, +\infty))$ and for any $\ell \in \{1, \ldots, N\}$, there exist $D_\ell \geq 0$ and $\beta_\ell \in \mathbb{N}$ such that for any $t \geq 0$*

$$\|\nabla^\ell \log p_t(x_t)\| \leq D_\ell(1 + \|x_t\|^{\beta_\ell}).$$

*In particular if $\alpha_1 = 1$ then $\beta_1 = 1$.*

*Proof.* Let $t \geq 0$ and $\ell \in \{1, \ldots, N\}$. Using Proposition S4 and Proposition S6 there exist $D_\ell^1 \geq 0$ and $\beta_\ell^1 \in \mathbb{N}$ such that for any $x_t \in \mathbb{R}^d$ we have

$$\|\nabla^\ell \log p_t(x_t)\| \leq D_\ell^1 c_t^{-2\beta_\ell^1}(1 + \|x_t\|^{\beta_\ell^1}).$$

Similarly, using Proposition S7 and Proposition S9 there exist $D_\ell^2 \geq 0$ and $\beta_\ell^2 \in \mathbb{N}$ such that for any $x_t \in \mathbb{R}^d$ we have

$$\|\nabla^\ell \log p_t(x_t)\| \leq D_\ell^2 (\alpha^{1/2}\sigma_t)^{-2\beta_\ell^2}(1 + \|x_t\|^{\beta_\ell^2}).$$

Therefore, there exist $\tilde{D}_\ell \geq 0$ and $\beta_\ell \in \mathbb{N}$ such that for any $x_t \in \mathbb{R}^d$ we have

$$\|\nabla^\ell \log p_t(x_t)\| \leq \tilde{D}_\ell \min(\alpha^{-1}\sigma_t^{-2}, c_t^{-2})^{\beta_\ell}(1 + \|x_t\|^{\beta_\ell}).$$

Since for any $c_t^{-2} = \exp[2\alpha t]$ and $\alpha^{-1}\sigma_t^{-2} = (1 - \exp[-2\alpha t])^{-1}$. Hence we have

$$\min(\alpha^{-1}\sigma_t^{-2}, c_t^{-2})^{\beta_\ell} \leq \max\{\min(1/u, 1/(1-u)) : u \in [0,1]\} \leq 2^{\beta_\ell},$$

which concludes the first part proof. We now show that if $\alpha_1 = 1$ then $\beta_1 = 1$. Recall that for any $t \geq 0$ and $x_t \in \mathbb{R}^d$ we have

$$p_t(x_t) = \int_{\mathbb{R}^d} p_{\text{data}}(x_0) \mathrm{g}(x_t - c_t x_0) \mathrm{d}x_0,$$

with for any $\tilde{x} \in \mathbb{R}^d$

$$c_t = \exp[-\alpha t], \qquad \mathrm{g}(\tilde{x}) = (2\pi\sigma_t^2)^{-d/2} \exp[-\|\tilde{x}\|^2/(2\sigma_t^2)], \quad \sigma_t^2 = (1 - \exp[-2\alpha t])/\alpha.$$

Therefore, using the dominated convergence theorem we get that for any $x_t \in \mathbb{R}^d$

$$\nabla \log p_t(x_t) = \sigma_t^{-2} \int_{\mathbb{R}^d} (x_t - c_t x_0) p_{0|t}(x_0|x_t) \mathrm{d}x_0 = \sigma_t^{-2} \int_{\mathbb{R}^d} (x_t - c_t x_0) q_{0|t}(x_0|x_t) \mathrm{d}x_0. \tag{S15}$$

Similarly, using the dominate convergence theorem and change of variable $z = x_t - c_t x_0$, we have for any $x_t \in \mathbb{R}^d$

$$\nabla \log p_t(x_t) = c_t^{-1} \int_{\mathbb{R}^d} \nabla \log p_{\text{data}}(x_0) p_{0|t}(x_0|x_t) \mathrm{d}x_0.$$

We conclude the proof upon combining this result, (S15), (S14) with $\alpha_1 = 1$, Proposition S9 and Proposition S6. In particular, we use that $\beta_1 = 1$. $\qquad \square$

### S3.2 Proof of Theorem 1

We start by recalling the following basic lemma.

**Lemma S11.** *Let $(\mathsf{E}, \mathcal{E})$ and $(\mathsf{F}, \mathcal{F})$ be two measurable spaces and $\mathrm{K} : \mathsf{E} \times \mathcal{F} \to [0, 1]$ be a Markov kernel. Then for any $\mu_0, \mu_1 \in \mathscr{P}(\mathsf{E})$ we have*

$$\|\mu_0 \mathrm{K} - \mu_1 \mathrm{K}\|_{\mathrm{TV}} \leq \|\mu_0 - \mu_1\|_{\mathrm{TV}}.$$

*In addition, for any $\varphi : \mathsf{E} \to \mathsf{F}$ measurable we get that*

$$\|\varphi_{\#}\mu_0 - \varphi_{\#}\mu_1\|_{\mathrm{TV}} \leq \|\mu_0 - \mu_1\|_{\mathrm{TV}},$$

*with equality if $\varphi$ is injective.*

*Proof.* We divide the proof into two parts.

(a) Note that for any $f : \mathsf{F} \to \mathbb{R}$ such that $\|f\|_\infty \leq 1$ we have $\|\mathrm{K}f\|_\infty \leq 1$. Using this result we get

$$\|\mu_0 \mathrm{K} - \mu_1 \mathrm{K}\|_{\mathrm{TV}} = \sup\{\textstyle\int_\mathsf{F} f(y)\mathrm{d}(\mu_0\mathrm{K})(y) - \int_\mathsf{F} f(y)\mathrm{d}(\mu_1\mathrm{K})(y) \ : \ \|f\|_\infty \leq 1\}$$
$$= \sup\{\textstyle\int_\mathsf{E} \mathrm{K}f(x)\mathrm{d}\mu_0(x) - \int_\mathsf{E} \mathrm{K}f(x)\mathrm{d}\mu_0(x) \ : \ \|f\|_\infty \leq 1\} \leq \|\mu_0 - \mu_1\|_{\mathrm{TV}}.$$

(b) We have

$$\|\varphi_{\#}\mu_0 - \varphi_{\#}\mu_1\|_{\mathrm{TV}} = \sup\{\textstyle\int_\mathsf{E} f(\varphi(x))\mathrm{d}\mu_0(x) - \int_\mathsf{E} f(\varphi(x))\mathrm{d}\mu_1(x) \ : \ \|f\|_\infty \leq 1\}$$
$$\leq \sup\{\textstyle\int_\mathsf{E} f(x)\mathrm{d}\mu_0(x) - \int_\mathsf{E} f(x)\mathrm{d}\mu_1(x) \ : \ \|f\|_\infty \leq 1\} \leq \|\mu_0 - \mu_1\|_{\mathrm{TV}}.$$

If $\varphi$ is injective then there exists $\varphi^{-1} : \mathsf{F} \to \mathsf{F}$ (measurable) such that $\varphi^{-1} \circ \varphi = \mathrm{Id}$. Therefore, for any $f : \mathsf{E} \to \mathbb{R}$ with $\|f\|_\infty \leq 1$ we have $f = (f \circ \varphi^{-1}) \circ \varphi$ and $\|f \circ \varphi^{-1}\|_\infty \leq 1$. Hence we have

$$\|\mu_0 - \mu_1\|_{\mathrm{TV}} = \sup\{\textstyle\int_\mathsf{E} f(x)\mathrm{d}\mu_0(x) - \int_\mathsf{E} f(x)\mathrm{d}\mu_1(x) \ : \ \|f\|_\infty \leq 1\}$$
$$\leq \sup\{\textstyle\int_\mathsf{E} f(\varphi(x))\mathrm{d}\mu_0(x) - \int_\mathsf{E} f(\varphi(x))\mathrm{d}\mu_1(x) \ : \ \|f\|_\infty \leq 1\} \leq \|\varphi_{\#}\mu_0 - \varphi_{\#}\mu_1\|_{\mathrm{TV}},$$

which concludes the proof.

$\square$

We will also make use of the following inequality.

**Lemma S12.** *Let $\varepsilon > 0$, $x, y \in \mathbb{R}^d$, $t > 2/\varepsilon$ and $\varphi : [0,1] \to \mathbb{R}$ such that for any $s \in [0,1]$, $\varphi(s) = \exp[-\|x - sy\|^2/(4t)]$. Then $\varphi \in \mathrm{C}^1([0,1], \mathbb{R})$ and we have for any $s \in [0,1]$*

$$|\varphi'(s)| \leq 2(1 + \varepsilon^{-1})(1 + \|x\|)\exp[-\|x\|^2/(8t)]\exp[\varepsilon\|y\|^2]/t.$$

*Proof.* Let $s \in [0,1]$, we have

$$\varphi'(s) = (\langle x, y \rangle - s\|y\|^2)\exp[-\|x - sy\|^2/(4t)]/(2t).$$

Using the Cauchy-Schwarz inequality and that for any $a, b \in \mathbb{R}^d$, $-\|a + b\|^2 \leq -\|a\|^2/2 + \|b\|^2$ we get

$$|\varphi'(s)| \leq (\|x\|\|y\| + \|y\|^2)\exp[-\|x\|^2/(8t) + \|y\|^2/(4t)]/(2t). \tag{S16}$$

In addition, we have

$$\|y\|\exp[\|y\|^2/(4t)] \leq \|y\|\exp[\varepsilon\|y\|^2/2] \leq (1 + \|y\|^2)\exp[\varepsilon\|y\|^2/2] \leq 2(1 + \varepsilon^{-1})\exp[\varepsilon\|y\|^2]. \tag{S17}$$

Finally we also have $\|y\|^2\exp[\|y\|^2/(4t)] \leq (1 + \varepsilon^{-1})\exp[\varepsilon\|y\|^2]$. Combining this result, (S16) and (S17) concludes the proof. $\square$

Finally we show the following lemma which is a straightforward consequence of Girsanov's theorem (Liptser and Shiryaev, 2001, Theorem 7.7). A similar version of this lemma can be found in the proof of (Durmus and Moulines, 2017, Proposition 2) and in (Laumont et al., 2021, Lemma 26) (version where the dependence of the drift in $w \in \mathrm{C}([0,T], \mathbb{R}^d)$ is replaced by a (simpler) dependence in $x \in \mathbb{R}^d$). We refer to (Liptser and Shiryaev, 2001, Section 4) for the definitions of semi-group, non-anticipative processes and diffusion type processes.

**Lemma S13.** *Let $T > 0$, $b_1, b_2 : [0, +\infty) \times \mathrm{C}([0,T], \mathbb{R}^d) \to \mathbb{R}^d$ measurable such that for any $i \in \{1,2\}$ and $x \in \mathbb{R}^d$, $\mathrm{d}\mathbf{X}_t^{(i)} = b_i(t, (\mathbf{X}_s^{(i)})_{s \in [0,T]})\mathrm{d}t + \sqrt{2}\mathrm{d}\mathbf{B}_t$ admits a unique strong solution with $\mathbf{X}_0^{(i)} = x$ and $(b_i(t, (\mathbf{X}_s^{(i)})))_{t \in [0,T]}$ is non-anticipative, with Markov semi-group $(\mathrm{P}_t^{(i)})_{t \geq 0}$. In addition, assume that for any $x \in \mathbb{R}^d$ and $i \in \{1,2\}$, $\mathbb{P}(\int_0^T \{\|b_i(t, (\mathbf{X}_s^{(i)})_{s \in [0,T]})\|^2 + \|b_i(t, (\mathbf{B}_s)_{s \in [0,T]})\|^2\}\mathrm{d}t < +\infty) = 1$. Then for any $x \in \mathbb{R}^d$ we have*

$$\|\delta_x \mathrm{P}_T^{(1)} - \delta_x \mathrm{P}_T^{(2)}\|_{\mathrm{TV}}^2 \leq (1/2)\int_0^T \mathbb{E}[\|b_1(t, (\mathbf{X}_s^{(1)})_{s \in [0,T]}) - b_2(t, (\mathbf{X}_s^{(1)})_{s \in [0,T]})\|^2]\mathrm{d}t.$$

*Proof.* Let $T > 0$ and $x \in \mathbb{R}^d$. For any $i \in \{1, 2\}$, denote $\mu_{(i)}^x$ the distribution of $(\mathbf{X}_t^{(i)})_{t \in [0,T]}$ on the Wiener space $(\mathcal{C}, \mathcal{B}(\mathcal{C}))$ with $\mathbf{X}_0^{(i)} = x$. Similarly denote $\mu_B^x$ the distribution of $(\mathbf{B}_t)_{t \in [0,T]}$ with $\mathbf{B}_0 = x$, where we recall that $(\mathbf{B}_t)_{t \in [0,T]}$ is a $d$-dimensional Brownian motion. Using Pinsker's inequality (Bakry et al., 2014, Equation 5.2.2) and the transfer theorem (Kullback, 1997, Theorem 4.1) we get that

$$\|\delta_x \mathrm{P}_T^{(1)} - \delta_x \mathrm{P}_T^{(2)}\|_{\mathrm{TV}}^2 \leq 2\,\mathrm{KL}(\mu_{(1)}|\mu_{(2)}).$$

Since for any $i \in \{1, 2\}$, $\mathbb{P}(\int_0^T \{\|b_i(t, (\mathbf{X}_s^{(i)})_{s \in [0,T]})\|^2 + \|b_i(t, (\mathbf{B}_s)_{s \in [0,T]})\|^2\}\mathrm{d}t < +\infty) = 1$ and the processes $(\mathbf{X}_t^{(i)})_{t \in [0,T]}$ are of diffusion type for $i \in \{1, 2\}$ we can apply Girsanov's theorem (Liptser and Shiryaev, 2001, Theorem 7.7) and $\mu_B$-almost surely for any $w \in \mathrm{C}([0,T],\mathbb{R})$ we get

$$(\mathrm{d}\mu_{(1)}^x / \mathrm{d}\mu_B^x)((w_t)_{t \in [0,T]})$$
$$= \exp[(1/2)\textstyle\int_0^T \langle b_1(t, (w_s)_{s \in [0,T]}), \mathrm{d}w_t\rangle - (1/4)\int_0^T \|b_1(t, (w_s)_{s \in [0,T]})\|^2 \mathrm{d}t]$$
$$(\mathrm{d}\mu_B^x / \mathrm{d}\mu_{(2)}^x)((w_t)_{t \in [0,T]})$$
$$= \exp[-(1/2)\textstyle\int_0^T \langle b_2(t, (w_s)_{s \in [0,T]})), \mathrm{d}w_t\rangle + (1/4)\int_0^T \|b_2(t, (w_s)_{s \in [0,T]}))\|^2 \mathrm{d}t].$$

Hence, we obtain that

$$\mathrm{KL}(\mu_{(1)}^x | \mu_{(2)}^x) = \mathbb{E}[\log((\mathrm{d}\mu_{(1)}^x / \mathrm{d}\mu_{(2)}^x)((\mathbf{X}_t^{(1)})_{t \in [0,T]}))]$$
$$= (1/4)\textstyle\int_0^T \mathbb{E}[\|b_1(t, (\mathbf{X}_s^{(1)})_{s \in [0,T]}) - b_2(t, (\mathbf{X}_s^{(1)})_{s \in [0,T]})\|^2]\mathrm{d}t$$

which concludes the proof. $\qquad\square$

We study distributions satisfying some curvature assumption and show that they are sub-Gaussian. More precisely, we show the following proposition.

**Lemma S14.** *Let $q \in \mathrm{C}^1(\mathbb{R}^d, (0, +\infty))$ and $\mathtt{m} > 0$ and $c \geq 0$ such that for any $x \in \mathbb{R}^d$ we have $\langle \nabla \log q(x), x\rangle \leq -\mathtt{m}\|x\|^2 + c\|x\|$. Then for any $\varepsilon \in [0, \mathtt{m}/2)$ we have*

$$\textstyle\int_{\mathbb{R}^d} \exp[\varepsilon\|x\|^2]q(x)\mathrm{d}x < +\infty.$$

*Proof.* For any $x \in \mathbb{R}^d$ we have

$$\log q(x) = \log q(0) + \textstyle\int_0^1 \langle \nabla \log q(tx), x\rangle \mathrm{d}t$$
$$\leq \log q(0) - \mathtt{m}\textstyle\int_0^1 t\|x\|^2\,\mathrm{d}t + c\|x\| \leq \log q(0) + c\|x\| - \mathtt{m}\|x\|^2.$$

which concludes the proof. $\qquad\square$

Finally, we will use the following basic lemma.

**Lemma S15.** *Let $\mu \in \mathscr{P}(\mathbb{R}^d)$, $\alpha_1 \in \mathbb{R}$, $\beta_1 > 0$ and $(\mathbf{X}_t)_{t \geq 0}$ such that $\mathbf{X}_0$ has distribution $\mu$ and*

$$\mathrm{d}\mathbf{X}_t = \alpha_1 \mathbf{X}_t \mathrm{d}t + \beta_1^{1/2} \mathrm{d}\mathbf{B}_t,$$

*where $(\mathbf{B}_t)_{t \geq 0}$ is a Brownian motion. Then for any $\alpha_2 \in \mathbb{R}$ and $\beta_2 > 0$ we have that $(\mathbf{Y}_t)_{t \geq 0}$ given for any $t \geq 0$ by $\mathbf{Y}_t = \alpha_2 \mathbf{X}_{\beta_2 t}$ satisfies*

$$\mathrm{d}\mathbf{Y}_t = \beta_2 \alpha_1 \mathbf{Y}_t \mathrm{d}t + \alpha_2 (\beta_2 \beta_1)^{1/2} \mathrm{d}\tilde{\mathbf{B}}_t,$$

*where $(\tilde{\mathbf{B}}_t)_{t \geq 0}$ is a Brownian motion, and $\mathbf{Y}_0$ has distribution $(\tau_{\alpha_2})_{\#}\mu$, where for any $x \in \mathbb{R}^d$, $\tau_{\alpha_2}(x) = \alpha_2 x$.*

*Proof.* Let $t \geq 0$. Using the change of variable $u \mapsto \beta_2 u$ the following equalities hold in distribution

$$\mathbf{Y}_t = \alpha_2 \alpha_1 \textstyle\int_0^{\beta_2 t} \mathbf{X}_s \mathrm{d}s + \alpha_2 \beta_1^{1/2} \mathbf{B}_{\beta_2 t}$$
$$= \beta_2 \alpha_2 \alpha_1 \textstyle\int_0^t \mathbf{X}_{\beta_2 s} \mathrm{d}s + \alpha_2 (\beta_1 \beta_2)^{1/2} \mathbf{B}_t = \beta_2 \alpha_1 \int_0^t \mathbf{Y}_s \mathrm{d}s + \alpha_2 (\beta_1 \beta_2)^{1/2} \mathbf{B}_t,$$

which concludes the proof. $\qquad\square$

We now turn to the proof of Theorem 1

*Proof.* Let $\alpha \geq 0$. For any $k \in \{1, \ldots, N\}$, denote $R_k$ the Markov kernel such that for any $x \in \mathbb{R}^d$, $A \in \mathcal{B}(\mathbb{R}^d)$ and $k \in \{0, \ldots, N - 1\}$ we have

$$R_{k+1}(x, A) = (4\pi\gamma_{k+1})^{-1/2} \int_A \exp[-\|\tilde{x} - \mathcal{T}_{k+1}(x)\|^2 / (4\gamma_{k+1})]d\tilde{x},$$

where for any $x \in \mathbb{R}^d$, $\mathcal{T}_{k+1}(x) = x + \gamma_{k+1}\{\alpha x + 2s_\theta(t_k, x)\}$, where $t_k = \sum_{\ell=0}^{k-1} \gamma_\ell$. Define for any $k_0, k_1 \in \{1, \ldots, N\}$ with $k_1 \geq k_0$ $Q_{k_0, k_1} = \prod_{\ell=k_0}^{k_1} R_\ell$. Finally, for ease of notation, we also define for any $k \in \{1, \ldots, N\}$, $Q_k = Q_{1,k}$. Note that for any $k \in \{1, \ldots, N\}$, $Y_k$ has distribution $\pi_\infty Q_k$, where $\pi_\infty \in \mathscr{P}(\mathbb{R}^d)$ with density w.r.t. the Lebesgue measure $p_{\text{data}}$. Let $\mathbb{P} \in \mathscr{P}(\mathcal{C})$ be the probability measure associated with the diffusion

$$d\mathbf{X}_t = -\alpha\mathbf{X}_t dt + \sqrt{2}d\mathbf{B}_t, \quad \mathbf{X}_0 \sim \pi_0,$$

where $\pi_0 \in \mathscr{P}(\mathbb{R}^d)$ admits a density w.r.t. the Lebesgue measure given by $p_{\text{data}}$. First note that using that $\mathbb{P}_0 = \pi_0$ we have for any $A \in \mathcal{B}(\mathbb{R}^d)$

$$\pi_0 \mathbb{P}_{T|0}(\mathbb{P}^R)_{T|0}(A) = \mathbb{P}_T(\mathbb{P}^R)_{T|0}(A) = (\mathbb{P}^R)_0(\mathbb{P}^R)_{T|0}(A) = (\mathbb{P}^R)_T(A) = \pi_0(A).$$

Hence $\pi_0 = \pi_0 \mathbb{P}_{T|0}(\mathbb{P}^R)_{T|0}$. Using this result and Lemma S11, we have

$$\|\pi_0 - \pi_\infty Q_N\|_{\text{TV}} = \|\pi_0 \mathbb{P}_{T|0}(\mathbb{P}^R)_{T|0} - \pi_\infty Q_N\|_{\text{TV}}$$
$$\leq \|\pi_0 \mathbb{P}_{T|0}(\mathbb{P}^R)_{T|0} - \pi_\infty(\mathbb{P}^R)_{T|0}\|_{\text{TV}} + \|\pi_\infty(\mathbb{P}^R)_{T|0} - \pi_\infty Q_N\|_{\text{TV}}$$
$$\leq \|\pi_0 \mathbb{P}_{T|0} - \pi_\infty\|_{\text{TV}} + \|\pi_\infty(\mathbb{P}^R)_{T|0} - \pi_\infty Q_N\|_{\text{TV}}.$$

Note that $\mathcal{L}(X_0) = \mathcal{L}(Y_N) = \pi_\infty Q_N$ and therefore

$$\|\mathcal{L}(X_0) - \pi_0\|_{\text{TV}} \leq \|\pi_0 \mathbb{P}_{T|0} - \pi_\infty\|_{\text{TV}} + \|\pi_\infty(\mathbb{P}^R)_{T|0} - \pi_\infty Q_N\|_{\text{TV}}.$$

We now bound each one of these terms.

(a) First, assume that $\alpha > 0$. Let $T_\alpha = \alpha T$ and $\tilde{\mathbb{P}} \in \mathscr{P}(\text{C}([0, T_\alpha], \mathbb{R}^d))$ be associated with $(\mathbf{Z}_t)_{t \in [0, T_\alpha]}$ the classical Ornstein-Uhlenbeck process with $\mathbf{Z}_0 \sim (\tau_\alpha)_\# \pi_0$, where for any $x \in \mathbb{R}^d$ we have $\tau_\alpha(x) = \alpha^{1/2} x$, satisfying the following SDE: $d\mathbf{Z}_t = -\mathbf{Z}_t dt + \sqrt{2}d\mathbf{B}_t$. We denote $\pi_0^\alpha = (\tau_\alpha)_\# \pi_0$, $\mu = (\tau_\alpha)_\# \pi_\infty$. Note that since $p_{\text{prior}}$ is the Gaussian density with zero mean and covariance matrix $(1/\alpha) \text{Id}$, $\mu$ is the Gaussian distribution with zero mean and identity covariance matrix.

First, using (Bakry et al., 2014, Proposition 4.1.1, Proposition 4.3.1, Theorem 4.2.5), we get that for any $t \in [0, T_\alpha]$, $f \in L^1(\mu)$ and $x \in \mathbb{R}^d$

$$\int_{\mathbb{R}^d} (\tilde{\mathbb{P}}_{t|0} g(x))^2 d\mu(x) \leq \exp[-2t] \int_{\mathbb{R}^d} g^2(x) d\mu(x), \quad \text{with } g(x) = f(x) - \int_{\mathbb{R}^d} f(\tilde{x}) d\mu(\tilde{x}). \quad \text{(S18)}$$

Recall that $(\mathbf{X}_t)_{t \geq 0}$ satisfies $d\mathbf{X}_t = -\alpha\mathbf{X}_t + d\mathbf{B}_t$. Using Lemma S15 we have that for any $t \in [0, T]$, $\mathbf{Z}_t$ and $\alpha^{1/2}\mathbf{X}_{\alpha^{-1}t}$ have the same distribution. Hence for any $t \in [0, T]$ we have $\mathbb{P}_t = (\tau_\alpha^{-1})_\# \tilde{\mathbb{P}}_{\alpha t}$. Therefore, using that $(\tau_\alpha)_\# \pi_\infty = \mu$, that $\tilde{\mathbb{P}}$ is Markov and Lemma S11, we get that

$$\|\pi_0 \mathbb{P}_{t|0} - \pi_\infty\|_{\text{TV}} = \|\mathbb{P}_t - \pi_\infty\|_{\text{TV}} = \|(\tau_\alpha)_\# \mathbb{P}_t - (\tau_\alpha)_\# \pi_\infty\|_{\text{TV}}$$
$$= \|\tilde{\mathbb{P}}_{\alpha t} - \mu\|_{\text{TV}} = \|\tilde{\mathbb{P}}_{\alpha t_0} \tilde{\mathbb{P}}_{\alpha(t-t_0)|0} - \mu\|_{\text{TV}}.$$

Finally, note that we have for any $t \geq t_0 \in [0, T]$ and $x \in \mathbb{R}^d$

$$(d(\tilde{\mathbb{P}}_{\alpha t_0} \tilde{\mathbb{P}}_{\alpha(t-t_0)|0})/d\mu)(x) = \tilde{\mathbb{P}}_{\alpha(t-t_0)|0} f(x), \quad \text{with } f(x) = (d\tilde{\mathbb{P}}_{\alpha t_0}/d\mu)(x). \quad \text{(S19)}$$

Let $g = f - 1$. Using (S19), (S18) and that $(\tau_\alpha)_\# \pi_\infty = \mu$, we get that for any $t \geq t_0$ with $t \in [0, T]$

$$\|\pi_0 \mathbb{P}_{t|0} - \pi_\infty\|_{\text{TV}} \leq \|\tilde{\mathbb{P}}_{\alpha t_0} \tilde{\mathbb{P}}_{\alpha(t-t_0)|0} - \mu\|_{\text{TV}} \quad \text{(S20)}$$
$$\leq \int_{\mathbb{R}^d} |\tilde{\mathbb{P}}_{\alpha(t-t_0)|0} f(x) - 1| d\mu(x)$$
$$\leq (\int_{\mathbb{R}^d} (\tilde{\mathbb{P}}_{\alpha(t-t_0)|0} g(x))^2 d\mu(x))^{1/2}$$

$$\leq \exp[-\alpha(t - t_0)](\textstyle\int_{\mathbb{R}^d} g^2(x)\mathrm{d}\mu(x))^{1/2}$$
$$\leq \exp[-\alpha(t - t_0)](\textstyle\int_{\mathbb{R}^d} g^2(\alpha^{1/2}x)\mathrm{d}\pi_\infty(x))^{1/2}.$$

In addition, we have for any $\varphi \in \mathrm{C}_c(\mathbb{R}^d, \mathbb{R})$

$$\textstyle\int_{\mathbb{R}^d} \varphi(x)f(\alpha^{1/2}x)\mathrm{d}\pi_\infty(x) = \int_{\mathbb{R}^d} \varphi(\alpha^{-1/2}x)f(x)\mathrm{d}\mu(x)$$
$$= \textstyle\int_{\mathbb{R}^d} \varphi(\alpha^{-1/2}x)\mathrm{d}\tilde{\mathbb{P}}_{\alpha t_0}(x) = \int_{\mathbb{R}^d} \varphi(x)\mathrm{d}\mathbb{P}_{t_0}(x).$$

Hence, for any $x \in \mathbb{R}^d$, $g(\alpha^{1/2}x) = (\mathrm{d}\mathbb{P}_{t_0}/\mathrm{d}\pi_\infty)(x) - 1$. Combining this result and (S20) we get that for any $t \geq t_0$ with $t \in [0, T]$

$$\|\pi_0\mathbb{P}_{t|0} - \pi_\infty\|_{\mathrm{TV}} \leq \sqrt{2}\exp[-\alpha(t - t_0)]\left(1 + \textstyle\int_{\mathbb{R}^d}(\mathrm{d}\mathbb{P}_{t_0}/\mathrm{d}\pi_\infty)(x)^2\mathrm{d}\pi_\infty(x)\right)^{1/2}. \qquad (\text{S21})$$

Let $t_0 \in [0, T]$. We now derive an upper bound for $\int_{\mathbb{R}^d}(\mathrm{d}\mathbb{P}_{t_0}/\mathrm{d}\pi_\infty)(x)^2\mathrm{d}\pi_\infty(x)$. We recall that $\mathbb{P}_{t_0}$ and $\pi_\infty$ admit density w.r.t. the Lebesgue measure denoted $p_{t_0}$ and $p_\infty$ such that for any $x \in \mathbb{R}^d$

$$p_{t_0}(x) = \textstyle\int_{\mathbb{R}^d}\mathrm{G}_{t_0}(x, \tilde{x})\mathrm{d}\pi_0(\tilde{x}), \quad p_\infty(x) = (2\pi/\alpha)^{-d/2}\exp[-\alpha\|x\|^2/2],$$

where for any $x, \tilde{x} \in \mathbb{R}^d$

$$\mathrm{G}_{t_0}(x, \tilde{x}) = (2\pi\sigma_{t_0}^2)^{-d/2}\exp[-\|x - m_{t_0}(\tilde{x})\|^2/(2\sigma_{t_0}^2)],$$
$$\sigma_{t_0}^2 = (1 - \exp[-2\alpha t_0])/\alpha, \qquad m_{t_0}(\tilde{x}) = \exp[-\alpha t_0]\tilde{x}.$$

Combining this result and Jensen's inequality we get

$$\textstyle\int_{\mathbb{R}^d} p_{t_0}^2(x)p_\infty^{-1}(x)\mathrm{d}x \leq \alpha^{-d/2}(2\pi)^{-d/2}\sigma_{t_0}^{-2d}\int_{\mathbb{R}^d}\exp[-\|x - m_{t_0}(\tilde{x})\|^2/\sigma_{t_0}^2 + \alpha\|x\|^2/2]\mathrm{d}x\mathrm{d}\pi_0(\tilde{x}). \qquad (\text{S22})$$

For any $x, \tilde{x} \in \mathbb{R}^d$ we have

$$\|x - m_{t_0}(\tilde{x})\|^2/\sigma_{t_0}^2 - \alpha\|x\|^2/2 = \|x - m_{t_0}(\tilde{x})(2\tilde{\sigma}_{t_0}^2/\sigma_{t_0}^2)\|^2/(2\tilde{\sigma}_{t_0}^2) - \|\tilde{x}\|^2\phi(\alpha, t_0)/\sigma_{t_0}^2,$$

with $\tilde{\sigma}_{t_0}^2 = (\sigma_{t_0}^2/2)(1 - \alpha\sigma_{t_0}^2/2)^{-1}$ and $\phi(\alpha, t_0) = \alpha\sigma_{t_0}^2(1 - \sigma_{t_0}^2\alpha)/(2 - \sigma_{t_0}^2\alpha)$. Using this result, we get that

$$\textstyle\int_{\mathbb{R}^d}\exp[-\|x - m_{t_0}(\tilde{x})\|^2/\sigma_{t_0}^2 + \alpha\|x\|^2/2]\mathrm{d}x\mathrm{d}\pi_0(\tilde{x}) \leq (2\pi\tilde{\sigma}_{t_0}^2)^{d/2}\int_{\mathbb{R}^d}\exp[\phi(\alpha, t_0)\|\tilde{x}\|^2]\mathrm{d}\pi_0(\tilde{x}),$$

Let $\varepsilon = \mathtt{m}/4$ and $t_0 \geq 0$ such that $\phi(\alpha, t_0) \leq \varepsilon$. Using Lemma S14, we get that

$$\textstyle\int_{\mathbb{R}^d}\exp[-\|x - m_{t_0}(\tilde{x})\|^2/\sigma_{t_0}^2 + \alpha\|x\|^2/2]\mathrm{d}x\mathrm{d}\pi_0(\tilde{x}) \leq (2\pi\tilde{\sigma}_{t_0}^2)^{d/2}\int_{\mathbb{R}^d}\exp[\varepsilon\|\tilde{x}\|^2]\mathrm{d}\pi_0(\tilde{x}).$$

Combining this result, the fact that $\sigma_{t_0}^2 \leq \alpha^{-1}$, (S22) and that for any $t \geq 0$, $(1 - \mathrm{e}^{-t})^{-1} \leq 1 + 1/t$, we obtain

$$\begin{aligned}
\textstyle\int_{\mathbb{R}^d} p_{t_0}^2(x)p_\infty^{-1}(x)\mathrm{d}x &\leq (\alpha^{-1}\tilde{\sigma}_{t_0}^2\sigma_{t_0}^{-4})^{d/2}\textstyle\int_{\mathbb{R}^d}\exp[\varepsilon\|\tilde{x}\|^2]\mathrm{d}\pi_0(\tilde{x}) \\
&\leq (1 - \exp[-2\alpha t_0])^{-d/2}\textstyle\int_{\mathbb{R}^d}\exp[\varepsilon\|\tilde{x}\|^2]\mathrm{d}\pi_0(\tilde{x}) \\
&\leq (1 + 1/(2\alpha t_0))^{d/2}\textstyle\int_{\mathbb{R}^d}\exp[\varepsilon\|\tilde{x}\|^2]\mathrm{d}\pi_0(\tilde{x}).
\end{aligned}$$

Combining this result and (S21), we get that for any $t > t_0$

$$\|\pi_0\mathbb{P}_{t|0} - \pi_\infty\|_{\mathrm{TV}} \leq C_1^a\exp[-\alpha t],$$

with

$$C_1^a = \sqrt{2}(1 + 1/(2\alpha t_0))^{d/2}(1 + (\textstyle\int_{\mathbb{R}^d}\exp[\varepsilon\|\tilde{x}\|^2]\mathrm{d}\pi_0(\tilde{x}))^{1/2})\exp[\alpha t_0].$$

For $t \leq t_0$, using that $\|\pi_0\mathbb{P}_{t|0} - \pi_\infty\|_{\mathrm{TV}} \leq 1$ we have

$$\|\pi_0\mathbb{P}_{t|0} - \pi_\infty\|_{\mathrm{TV}} \leq C_1^b\exp[-\alpha t], \qquad \text{with } C_1^b = \exp[\alpha t_0].$$

Let $C_1 = C_1^a + C_1^b$ and we have that for any $t \in [0, T]$

$$\|\pi_0\mathbb{P}_{t|0} - \pi_\infty\|_{\mathrm{TV}} \leq C_1\exp[-\alpha t]. \qquad (\text{S23})$$

(b) Second assume that $\alpha = 0$.

$$\|\pi_0 \mathbb{P}_{T|0} - \pi_\infty\|_{\mathrm{TV}} \leq \int_{\mathbb{R}^d} \int_{\mathbb{R}^d} (4\pi T)^{-d/2} |\exp[-\|x - \tilde{x}\|^2 /(4T)] - \exp[-\|x\|^2 /(4T)]| \mathrm{d}x \mathrm{d}\pi_0(\tilde{x}).$$

For any $x, \tilde{x} \in \mathbb{R}^d$, let $\varphi \in \mathrm{C}^1([0,1], \mathbb{R})$ with for any $s \in [0,1]$, $\varphi(s) = \exp[-\|x - s\tilde{x}\|^2 /(4T)]$.
First, assume that $T \geq 2/\varepsilon$. Using Lemma S12, we get that for any $s \in [0,1]$

$$|\varphi'(s)| \leq (1 + \varepsilon^{-1})(1 + \|x\|) \exp[-\|x\|^2 /(8T)] \exp[\varepsilon \|y\|^2]/T.$$

Using this result we get that

$$\|\pi_0 \mathbb{P}_{T|0} - \pi_\infty\|_{\mathrm{TV}} \leq \int_{\mathbb{R}^d} \int_{\mathbb{R}^d} (4\pi T)^{-d/2} |\exp[-\|x - \tilde{x}\|^2 /(4T)] - \exp[-\|x\|^2 /(4T)]| \mathrm{d}x \mathrm{d}\pi_0(\tilde{x})$$

$$\leq \int_{\mathbb{R}^d} \int_{\mathbb{R}^d} (4\pi T)^{-d/2} (1 + \varepsilon^{-1})(1 + \|x\|) \exp[-\|x\|^2 /(8T)] \exp[\varepsilon \|\tilde{x}\|^2]/T \mathrm{d}x \mathrm{d}\pi_0(\tilde{x})$$

$$\leq 2^{d/2}(1 + \varepsilon^{-1}) \int_{\mathbb{R}^d} (8\pi T)^{-d/2} (1 + \|x\|) \exp[-\|x\|^2 /(8T)] \mathrm{d}x \int_{\mathbb{R}^d} \exp[\varepsilon \|\tilde{x}\|^2]/T \mathrm{d}\pi_0(\tilde{x})$$

$$\leq 2^{d/2}(1 + \varepsilon^{-1})(1 + 2\sqrt{2}d^{1/2}T^{1/2}) \int_{\mathbb{R}^d} \exp[\varepsilon \|\tilde{x}\|^2]/T \mathrm{d}\pi_0(\tilde{x}).$$

In addition, if $T \leq 2/\varepsilon$ then

$$\|\pi_0 \mathbb{P}_{T|0} - \pi_\infty\|_{\mathrm{TV}} \leq (\varepsilon/2 + (\varepsilon/2)^{1/2})^{-1}(T^{-1} + T^{-1/2}).$$

Hence, we get that there exists $C_2 \geq 0$ such that

$$\|\pi_0 \mathbb{P}_{T|0} - \pi_\infty\|_{\mathrm{TV}} \leq C_2(T^{-1} + T^{-1/2}), \tag{S24}$$

with

$$C_2 = (\varepsilon/2 + (\varepsilon/2)^{1/2})^{-1} + 2^{d/2}(1 + \varepsilon^{-1})(1 + 2\sqrt{2}d^{1/2}) \int_{\mathbb{R}^d} \exp[\varepsilon \|\tilde{x}\|^2] \mathrm{d}\pi_0(\tilde{x}).$$

(c) Recall that $\mathbb{P}^R$ is associated with the diffusion $(\mathbf{Y}_t)_{t\geq 0}$ such that for any $t \in [0, T]$ and $x \in \mathbb{R}^d$

$$\mathrm{d}\mathbf{Y}_t = b_1(t, \mathbf{Y}_t)\mathrm{d}t + \sqrt{2}\mathbf{B}_t, \quad b_1(t, x) = \alpha x + 2\nabla \log p_{T-t}(x).$$

Similarly, for any $k \in \{1, \ldots, N\}$ we have $\mathbb{Q}_k = \mathbb{Q}_{t_k}$ where $\mathbb{Q}$ is associated with the diffusion $(\bar{\mathbf{Y}}_t)_{t\in[0,T]}$ such that for any $(w_t)_{t\in[0,T]} \in \mathrm{C}([0,T], \mathbb{R}^d)$ we have

$$\mathrm{d}\bar{\mathbf{Y}}_t = b_2(t, (\bar{\mathbf{Y}}_s)_{s\in[0,T]})\mathrm{d}t + \sqrt{2}\mathbf{B}_t,$$

$$b_2(t, (w_t)_{t\in[0,T]}) = \sum_{k=0}^{N-1} \mathbb{1}_{[t_k, t_{k+1})}(t) \{2\alpha w_{t_k} + s_\theta(t_k, w_{t_k})\}$$

where for any $k \in \{0, \ldots, N\}$, $t_k = \sum_{\ell=0}^{k-1} \gamma_{\ell+1}$. Recall that for any $i \in \{1, 2, 3\}$ there exist $A_i \geq 0$ and $\alpha_i \in \mathbb{N}$ such that for any $x_0 \in \mathbb{R}^d$

$$\|\nabla^i \log p_0(x)\| \leq A_i(1 + \|x_0\|^{\alpha_i}),$$

with $\alpha_1 = 1$. Using this result and Theorem S10 we get that for any $i \in \{1, 2, 3\}$ there exist $B_i \geq 0$ and $\beta_i \in \mathbb{N}$ with $\beta_1 = 1$ such that for any $x_t \in \mathbb{R}^d$ and $t \in [0, T]$

$$\|\nabla^i \log p_t(x_t)\| \leq B_i(1 + \|x_t\|^{\beta_i}). \tag{S25}$$

In addition, for any $t \in [0, T]$ and $x \in \mathbb{R}^d$ we have

$$\partial_t p_t(x) = -\mathrm{div}(b p_t)(x) + \Delta p_t(x),$$

with $b(x) = -\alpha x$. Therefore, since $\log p \in \mathrm{C}^\infty((0, T] \times \mathbb{R}^d, \mathbb{R})$ we obtain that for any $t \in (0, T]$ and $x_t \in \mathbb{R}^d$

$$\partial_t \log p_t(x_t) = -\mathrm{div}(b \log p_t)(x_t) + \Delta \log p_t(x_t) + \|\nabla \log p_t(x_t)\|^2.$$

Finally, we get that for any $t \in (0, T]$ and $x_t \in \mathbb{R}^d$

$$\partial_t \nabla \log p_t(x_t) = -\nabla \mathrm{div}(b \log p_t)(x_t) + \nabla \Delta \log p_t(x_t) + \nabla \|\nabla \log p_t\|^2 (x_t).$$

Therefore combining this result and (S25) there exist $\tilde{A} \geq 0$ and $\beta \in \mathbb{N}$ such that for any $x_t \in \mathbb{R}^d$ and $t \in (0, T]$, $\|\partial_t \nabla \log p_t(x_t)\| \leq \tilde{A}(1 + \|x_t\|^\beta)$. Hence, for any $t_1, t_2 \in [0, T]$ and $x \in \mathbb{R}^d$

$$\|\nabla \log p_{t_2}(x) - \nabla \log p_{t_1}(x)\| \leq \tilde{A} |t_2 - t_1| (1 + \|x\|^\beta). \tag{S26}$$

In addition, using (S25), we have for any $t \in [0, T]$ and $x_1, x_2 \in \mathbb{R}^d$

$$\|\nabla \log p_t(x_1) - \nabla \log p_t(x_2)\| \le \int_0^1 \|\nabla^2 \log p_t((1-s)x_1 + sx_2)\| \mathrm{d}s \|x_1 - x_2\| \quad \text{(S27)}$$
$$\le B_2(1 + \int_0^1 \|(1-s)x_1 + sx_2\|^{\beta_2} \mathrm{d}s) \|x_1 - x_2\|$$
$$\le B_2(1 + \|x_1\|^{\beta_2} + \|x_2\|^{\beta_2}) \|x_1 - x_2\|.$$

Since $s_\theta \in \mathrm{C}([0,T] \times \mathbb{R}^d, \mathbb{R}^d)$ and $\nabla \log p \in \mathrm{C}([0,T] \times \mathbb{R}^d, \mathbb{R}^d)$ we have using Lemma S13, (S26), (S27) and the Cauchy-Schwarz inequality

$$\|\pi_\infty(\mathbb{P}^R)_{T|0} - \pi_\infty Q_N\|_{\mathrm{TV}}^2 \le (1/2)\int_0^T \mathbb{E}[\|b_1(t, \mathbf{Y}_t) - b_2(t, (\mathbf{Y}_t)_{t \in [0,T]})\|^2]\mathrm{d}t \quad \text{(S28)}$$
$$\le 2\sum_{k=0}^{N-1} \int_{t_k}^{t_{k+1}} \mathbb{E}[\|\nabla \log p_{T-t}(\mathbf{Y}_t) - s_\theta(\mathbf{Y}_{t_k})\|^2]\mathrm{d}t$$
$$+ \sum_{k=0}^{N-1} \int_{t_k}^{t_{k+1}} \alpha^2 \mathbb{E}[\|\mathbf{Y}_t - \mathbf{Y}_{t_k}\|^2]\mathrm{d}t$$
$$\le 6\sum_{k=0}^{N-1} \int_{t_k}^{t_{k+1}} \mathbb{E}[\|\nabla \log p_{T-t}(\mathbf{Y}_t) - \nabla \log p_{T-t}(\mathbf{Y}_{t_k})\|^2]\mathrm{d}t$$
$$+ 6\sum_{k=0}^{N-1} \int_{t_k}^{t_{k+1}} \mathbb{E}[\|\nabla \log p_{T-t}(\mathbf{Y}_{t_k}) - \nabla \log p_{T-t_k}(\mathbf{Y}_{t_k})\|^2]\mathrm{d}t$$
$$+ 6\sum_{k=0}^{N-1} \int_{t_k}^{t_{k+1}} \mathbb{E}[\|\nabla \log p_{T-t_k}(\mathbf{Y}_{t_k}) - s_\theta(t_k, \mathbf{Y}_{t_k})\|^2]\mathrm{d}t$$
$$+ \sum_{k=0}^{N-1} \int_{t_k}^{t_{k+1}} \alpha^2 \mathbb{E}[\|\mathbf{Y}_t - \mathbf{Y}_{t_k}\|^2]\mathrm{d}t$$
$$\le 18\sqrt{2}B_2^2(1 + 2N_T(4\beta_2))^{1/2} \sum_{k=0}^{N-1} \int_{t_k}^{t_{k+1}} \mathbb{E}[\|\mathbf{Y}_t - \mathbf{Y}_{t_k}\|^4]^{1/2}\mathrm{d}t$$
$$+ 12\tilde{A}^2(1 + N_T(2\beta))\sum_{k=0}^{N-1} \int_{t_k}^{t_{k+1}} (t - t_k)^2\mathrm{d}t + 6T\mathtt{M}^2$$
$$+ \sum_{k=0}^{N-1} \int_{t_k}^{t_{k+1}} \alpha^2 \mathbb{E}[\|\mathbf{Y}_t - \mathbf{Y}_{t_k}\|^2]\mathrm{d}t$$
$$\le \{18\sqrt{2}B_2^2(1 + 2N_T(4\beta_2))^{1/2} + \alpha^2\} \sum_{k=0}^{N-1} \int_{t_k}^{t_{k+1}} \mathbb{E}[\|\mathbf{Y}_t - \mathbf{Y}_{t_k}\|^4]^{1/2}\mathrm{d}t$$
$$+ 4\tilde{A}^2(1 + N_T(2\beta))\sum_{k=0}^{N-1} (t_{k+1} - t_k)^3 + 6T\mathtt{M}^2$$
$$\le \{18\sqrt{2}B_2^2(1 + 2N_T(4\beta_2))^{1/2} + \alpha^2\} \sum_{k=0}^{N-1} \int_{t_k}^{t_{k+1}} \mathbb{E}[\|\mathbf{Y}_t - \mathbf{Y}_{t_k}\|^4]^{1/2}\mathrm{d}t$$
$$+ 4\tilde{A}^2(1 + N_T(2\beta))T\bar{\gamma}^2 + 6T\mathtt{M}^2,$$

where for any $\ell \in \mathbb{N}$, $N_T(\ell) = \sup_{t \in [0,T]} \mathbb{E}[\|\mathbf{Y}_t\|^\ell]$. For any $t \in [0, T]$, let $\mathcal{A}_t : \mathrm{C}^2(\mathbb{R}^d) \to \mathrm{C}^2(\mathbb{R}^d, \mathbb{R})$ the generator given for any $t \ge 0$, $\varphi \in \mathrm{C}^2(\mathbb{R}^d, \mathbb{R})$ and $x \in \mathbb{R}^d$ by

$$\mathcal{A}_t(\varphi)(x) = \langle \alpha x + 2\nabla \log p_{T-t}(x), \nabla \varphi(x) \rangle + \Delta \varphi(x).$$

For any $\ell \in \mathbb{N}$, let $V_\ell(x) = \|x\|^{2\ell}$. Hence, for any $\ell \in \mathbb{N}$, $x \in \mathbb{R}^d$ and $t \in [0, T]$ we have using (S25)

$$\mathcal{A}_t(V_\ell)(x) = 2\ell\alpha \|x\|^{2\ell} + 2\ell B_1 \|x\|^{2\ell-1} + 2\ell B_1 \|x\|^{2\ell} + 2\ell(2\ell - 1) \|x\|^{2(\ell-1)}.$$

Hence, for any $\ell \in \mathbb{N}$ there exist $\tilde{B}_\ell$ such that $x \in \mathbb{R}^d$ and $t \in [0, T]$

$$|\mathcal{A}_t(V_\ell)(x)| \le \tilde{B}_\ell(1 + V_\ell(x)). \quad \text{(S29)}$$

For any $\ell \in \mathbb{N}$, $(M_{\ell,t})_{t \in [0,T]} = (V_\ell(\mathbf{Y}_t) - V_\ell(\mathbf{Y}_0) - \int_0^t \mathcal{A}_t(V_\ell)(\mathbf{Y}_s)\mathrm{d}s)_{t \in [0,T]}$ is a local martingale. For any $\ell \in \mathbb{N}$, there exists $(\tau_{\ell,k})_{k \in \mathbb{N}}$ a sequence of stopping times such that $\lim_{k \to +\infty} \tau_{\ell,k} = T$ and $(M_{\ell,t \wedge \tau_{\ell,k}})_{t \in [0,T]}$ is a martingale. Using (S29), we have for any $t \in [0, T]$, $\ell \in \mathbb{N}$ and $k \in \mathbb{N}$

$$\mathbb{E}[V_\ell(\mathbf{Y}_{t \wedge \tau_{\ell,k}})] \le \mathbb{E}[V_\ell(\mathbf{Y}_0)] + \tilde{B}_\ell \int_0^t (1 + \mathbb{E}[V_\ell(\mathbf{Y}_{s \wedge \tau_{\ell,k}})])\mathrm{d}s.$$

Hence, using Grönwall's lemma we get that for any $\ell \in \mathbb{N}$, $\sup_{k \in \mathbb{N}} \mathbb{E}[V_\ell(\mathbf{Y}_{t \wedge \tau_{\ell,k}})] < +\infty$. Therefore for any $\ell \in \mathbb{N}$, $((M_{\ell,t \wedge \tau_k})_{t \in [0,T]})_{k \in \mathbb{N}}$ is uniformly integrable and we have that for any $\ell \in \mathbb{N}$, $(M_{\ell,t})_{t \in [0,T]}$ is a martingale. Therefore we get that for any $t \in [0, T]$, $\ell \in \mathbb{N}$

$$\mathbb{E}[V_\ell(\mathbf{Y}_t)] \le \mathbb{E}[V_\ell(\mathbf{Y}_0)] + \tilde{B}_\ell \int_0^t (1 + \mathbb{E}[V_\ell(\mathbf{Y}_s)])\mathrm{d}s.$$

Using Grönwall's lemma we get that for any $\ell \in \mathbb{N}$ there exist $\tilde{C}_\ell \ge 0$ such that

$$N_T(\ell) = \sup_{t \in [0,T]} \mathbb{E}[\|\mathbf{Y}_t\|^{2\ell}] \le \tilde{C}_\ell \exp[\tilde{B}_\ell T]. \quad \text{(S30)}$$

We have that for any $s, t \in [0, T]$

$$\mathbf{Y}_t = \mathbf{Y}_s + \int_s^t \{\alpha \mathbf{Y}_u + 2\nabla \log p_{T-u}(\mathbf{Y}_u)\} \mathrm{d}u + \sqrt{2} \int_s^t \mathrm{d}\mathbf{B}_u.$$

Using (S26) and Cauchy-Schwarz inequality we have for any $s, t \in [0, T]$

$$
\begin{aligned}
\mathbb{E}[\|\mathbf{Y}_t - \mathbf{Y}_s\|^4] &\leq 64(t-s)^3 \int_s^t \{\alpha^4 \mathbb{E}[\|\mathbf{Y}_u\|^4] + 16\mathbb{E}[\|\nabla \log p_{T-u}(\mathbf{Y}_u)\|^4]\} \mathrm{d}u + 48\sqrt{2}(t-s)^2 \\
&\leq 64(t-s)^3 \int_s^t \{\alpha^4 \mathbb{E}[\|\mathbf{Y}_u\|^4] + 128B_1^4(1 + \mathbb{E}[\|\mathbf{Y}_u\|^4])\} \mathrm{d}u + 48\sqrt{2}(t-s)^2 \\
&\leq 64(\alpha^4 + 128B_1^4)(1 + N_T(4))(t-s)^4 + 48\sqrt{2}(t-s)^2. \quad\quad \text{(S31)}
\end{aligned}
$$

Combining (S30) and (S31) in (S28) we get that there exist $C_3 \geq 0$ such that

$$\|\pi_\infty(\mathbb{P}^R)_{T|0} - \pi_\infty \mathrm{Q}_N\|_{\mathrm{TV}}^2 \leq C_3 \exp[C_3 T](\bar{\gamma} + \mathtt{M}^2), \quad\quad \text{(S32)}$$

We conclude the proof upon combining (S23) and (S32) if $\alpha > 0$ and (S24) and (S32) if $\alpha = 0$. $\quad\square$

### S3.3 General SGM and links with existing works

In this section we describe a general algorithm for SGM in Section S3.3.1 and show that the formulation (6) encompasses the ones of (Song et al., 2021; Ho et al., 2020) in Section S3.3.2.

### S3.3.1 General SGM algorithm

We first present a general algorithm to compute approximate reverse dynamics, *i.e.* to compute the reverse-time Markov chain associated with the forward process

$$\mathrm{d}\mathbf{X}_t = f_t(\mathbf{X}_t)\mathrm{d}t + \sqrt{2}\mathrm{d}\mathbf{B}_t, \qquad \mathbf{X}_0 \sim p_{\mathrm{data}}. \quad\quad \text{(S33)}$$

We use the Euler-Maruyama discretization of (S33), *i.e.* let $X_0 \sim p_{\mathrm{data}}$ and for any $k \in \{0, \ldots, N-1\}$

$$X_{k+1} = X_k + \gamma_{k+1} f_k(X_k) + \sqrt{2\gamma_{k+1}} Z_{k+1}.$$

In general, we do not have that $p(x_k|x_0)$ is a Gaussian density contrary to Song and Ermon (2019); Ho et al. (2020). However, in this case, we obtain that for any $x \in \mathbb{R}^d$,

$$p_{k+1}(x) = (4\pi\gamma_{k+1})^{-d/2} \int_{\mathbb{R}^d} p_k(\tilde{x}) \exp[-\|\mathcal{T}_{k+1}(\tilde{x}) - x\|^2 / (4\gamma_{k+1})] \mathrm{d}\tilde{x},$$

with $\mathcal{T}_{k+1}(x) = \tilde{x} + \gamma_{k+1} f_k(\tilde{x})$. Therefore, we get that for any $x \in \mathbb{R}^d$

$$(2\gamma_{k+1} p_{k+1}(x))\nabla \log p_{k+1}(x) = \int_{\mathbb{R}^d} (\mathcal{T}_{k+1}(\tilde{x}) - x) p_k(\tilde{x}) \exp[-\|\mathcal{T}_{k+1}(\tilde{x}) - x\|^2 / (4\gamma_{k+1})] \mathrm{d}\tilde{x}.$$

Hence, we get that for any $x \in \mathbb{R}^d$

$$\nabla \log p_{k+1}(x) = \mathbb{E}[\mathcal{T}_{k+1}(X_k) - X_{k+1}|X_{k+1} = x]/(2\gamma_{k+1}) = -(2\gamma_{k+1})^{1/2}\mathbb{E}[Z_{k+1}|X_{k+1} = x]. \quad \text{(S34)}$$

From this formula we derive a regression problem similar to the one of Section 2.1. We obtain Algorithm 1. We highlight a few differences between our approach and the ones of Song and Ermon (2019); Ho et al. (2020):

(a) As emphasized in (S34), the regression problem in Algorithm 1 is different from the one usually considered in SGM which restrict themselves to the setting $f_k(x) = \alpha x$ with $\alpha = 0$ (Song and Ermon, 2019) or $\alpha > 0$ (Ho et al., 2020).

(b) In the present algorithm we do not use any corrector step (Song et al., 2021) at sampling time. Note that the use of a corrector step is only justified in the context of classical SGM algorithms and not the DSB method introduced in Section 3.3. This is because, we do not have access to the marginal of the time-reverse density during the IPF iterations contrary to classical SGMs.

(c) Finally, we do not present the Exponential Moving Average (EMA) procedure Song and Ermon (2020) which is key to prevent the network from oscillating. Contrary to the corrector step, this technique can easily be incorporated in Algorithm 1.

Further comments and additional techniques are presented in Section S9.

---

**Algorithm 1** Generalized score-matching

---

1: **Inputs:** $(b_k)_{k \in \{0,\dots,N-1\}}$, $N \in \mathbb{N}$ (nb. of iterations), $M \in \mathbb{N}$ (batch size), $N_{\text{epochs}}$ (nb. of epochs), $(\gamma_k)_{k \in \{0,\dots,N-1\}}$ (stepsizes), $\{s_\theta : \theta \in \Theta\}$ (neural network), $\text{opt}$ (optimizer), $p_{\text{prior}}$ (prior distribution), $\lambda(k)$ (weights)

2: **for** $n_{\text{epoch}} = 0, \dots, N_{\text{epoch}} - 1$ **do**

3:    **for** $j \in \{1, \dots, M\}$ **do**

4:       $X_0^j \sim p_{\text{data}}$

5:       **for** $k \in \{0, \dots, N-1\}$ **do**

6:          $X_{k+1}^j = X_k^j + \gamma_{k+1} f_k(X_k^j) + \sqrt{2\gamma_{k+1}} Z_{k+1}^j$

7:       **end for**

8:    **end for**

9:    $\widehat{\ell}(\theta) = M^{-1} \sum_{j=1}^{M} \sum_{k=0}^{N-1} \lambda(k)/(2\gamma_{k+1}) \sum_{j=1}^{M} \|\sqrt{2\gamma_{k+1}} s_\theta(k+1, X_{k+1}^j) + Z_{k+1}^j\|^2$

10:    $\theta_{n_{\text{epoch}}+1} = \text{opt}(\ell, \theta_{n_{\text{epoch}}})$

11: **end for**

12: $X_N \sim p_{\text{prior}}$

13: **for** $k \in \{N-1, \dots, 0\}$ **do**

14:    $X_k = X_{k+1} + \gamma_{k+1}\{-f_k(X_{k+1}) + 2s_{\theta_{N_{\text{epoch}}}}(k+1, X_{k+1})\} + \sqrt{2\gamma_{k+1}} Z_{k+1}$

15: **end for**

16: **Output:** $X_0$

---

### S3.3.2   Links with existing work

In this section, we show that we can recover the training and sampling algorithm of Song and Ermon (2019) and Ho et al. (2020) by reversing homogeneous diffusions. Note that Song et al. (2021) identified links with non-homogeneous SDEs. We explicitly characterize the fundamental difference between the approaches of Song and Ermon (2019); Ho et al. (2020) by identifying the two corresponding forward homogeneous processes (Brownian motion or Ornstein-Ulhenbeck).

**Brownian motion**    First, we show that we can recover the sampling procedure and the loss function of Song and Ermon (2019) by reversing a Brownian motion. Assume that we have

$$\mathrm{d}\mathbf{X}_t = \sqrt{2}\mathrm{d}\mathbf{B}_t, \qquad \mathbf{X}_0 \sim p_{\text{data}}. \tag{S35}$$

In what follows we define $\{Y_k\}_{k=0}^{N-1}$ such that $\{Y_k\}_{k=0}^{N-1}$ approximates $\{\mathbf{X}_{T-t_k}\}_{k=0}^{N-1}$ for a specific sequence of times $\{t_k\}_{k=0}^{N-1} \in [0, T]^N$. We recall that the time-reversal of (S35) is associated with the following SDE

$$\mathrm{d}\mathbf{Y}_t = 2\nabla \log p_{T-t}(\mathbf{Y}_t) + \sqrt{2}\mathrm{d}\mathbf{B}_t. \tag{S36}$$

The Euler-Maruyama discretization of (S36) yields for any $k \in \{0, \dots, N-1\}$

$$\tilde{Y}_{k+1} = \tilde{Y}_k + 2\gamma_{k+1}\nabla \log p_{T-t_k}(\tilde{Y}_k) + \sqrt{2\gamma_{k+1}} Z_{k+1}.$$

where $\{\gamma_{k+1}\}_{k=0}^{N-1}$ is a sequence of stepsizes and for any $k \in \{0, \dots, N\}$, $t_k = \sum_{j=0}^{k-1} \gamma_{j+1}$. A close form for $\{\nabla \log p_{T-t_k}\}_{k=0}^{N-1}$ is not available and in practice we consider

$$Y_{k+1} = Y_k + 2\gamma_{k+1} s_{\theta^\star}(T-t_k, Y_k) + \sqrt{2\gamma_{k+1}} Z_{k+1}, \tag{S37}$$

where for any $k \in \{0, \dots, N-1\}$, $s_{\theta^\star}(T-t_k, \cdot)$ is an approximation of $\nabla \log p_{T-t_k}$. The sampling procedure (S37) is similar to the one of Song and Ermon (2019) upon setting (with the notations of Song and Ermon (2019)) $T \leftarrow 1$ in (Song and Ermon, 2019, Algorithm 1) (no corrector step), $\alpha_k/2 \leftarrow \gamma_k$ and $\mathbf{s}_{\boldsymbol{\theta}}(\cdot, \sigma_{k+1}) \leftarrow 2s_{\theta^\star}(T-t_k, \cdot)$. It remains to show that $2s_{\theta^\star}$ is the solution to the same regression problem as $\mathbf{s}_{\boldsymbol{\theta}}$ in (Song and Ermon, 2019, Equation 6). First, note that for any $t > 0$ and $x_t \in \mathbb{R}^d$ we have

$$p_t(x_t) = (4\pi t)^{-d/2} \int_{\mathbb{R}^d} p_{\text{data}}(x_0) \exp[-\|x_t - x_0\|^2/(4t)]\mathrm{d}x_0.$$

Therefore, we get that for any $t > 0$ and $x_t \in \mathbb{R}^d$

$$\nabla \log p_t(x_t) = \int_{\mathbb{R}^d} (x_0 - x_t)/(2t) p_{0|t}(x_0|x_t)\mathrm{d}x_0 = \mathbb{E}[\mathbf{X}_0 - \mathbf{X}_t | \mathbf{X}_t = x_t]/(2t).$$

Hence, we have that $\theta^\star$ satisfies the following regression problem

$$\theta^\star = \arg\min_\theta \sum_{k=0}^{N-1} \lambda(k) \mathbb{E}[\|(\mathbf{X}_0 - \mathbf{X}_{T-t_k})/(T-t_k) - 2s_\theta(T-t_k, \mathbf{X}_{T-t_k})\|].$$

Note that this loss function is similar to the one of (Song and Ermon, 2019, Equation 6) upon letting $\sigma_{k+1}^2 \leftarrow 2(T-t_k)$ and $L \leftarrow N$. Hence, the two recursions approximately define the same scheme if for any $k \in \{0, \ldots, N-1\}$, $\sigma_1^2 - \sigma_{k+1}^2 \approx (1/2)\sum_{j=0}^{k-1} \alpha_{j+1}$ since $t_0 = 0$ implies $T = (1/2)\sigma_1^2$. In Song and Ermon (2019) we have for any $k \in \{0, \ldots, N-1\}$, $\sigma_k^2 = \kappa^{N-k}\sigma_N^2$ (recall that $N = L$) with $\kappa > 1$. In addition, we have for any $k \in \{0, \ldots, N-1\}$, $\alpha_k = \varepsilon \sigma_k^2/\sigma_N^2$ for some $\varepsilon > 0$. We get that

$$\begin{aligned}
(1/2)\sum_{j=0}^{k-1} \alpha_{j+1} &= (\varepsilon/2)\kappa^{N-1}\sum_{j=0}^{k-1} \kappa^{-j} \\
&= (\varepsilon/2)(\kappa^{N-1} - \kappa^{N-k-1})/(1 - \kappa^{-1}) \\
&= \varepsilon/(2(1-\kappa^{-1})\sigma_N^2)(\sigma_1^2 - \sigma_{k+1}^2).
\end{aligned}$$

Hence, the two schemes are identical if $\varepsilon = 2(1-\kappa^{-1})\sigma_N^2$. In practice in Song and Ermon (2019) the authors choose $N = 10$, $\sigma_N = 10^{-2}$, $\sigma_1 = 1$ (hence $\kappa = 10^{4/9}$) and $\varepsilon = 2 \times 10^{-5}$. We have $2(1-\kappa^{-1})\sigma_N^2 \approx 1.3 \times 10^{-4}$ which has one order of difference with $\varepsilon$.

**Ornstein-Ulhenbeck**    Second, we show that we can recover the sampling procedure and the loss function of Ho et al. (2020) by reversing an Ornstein-Ulhenbeck process. Contrary to the previous analysis we do not show a strict equivalence between the two recursions but instead that our algorithm can be seen as a first order approximation of the one of Ho et al. (2020).

In this section, we consider the following diffusion

$$\mathrm{d}\mathbf{X}_t = -\alpha \mathbf{X}_t \mathrm{d}t + \sqrt{2}\mathrm{d}\mathbf{B}_t, \qquad \mathbf{X}_0 \sim p_{\text{data}}. \tag{S38}$$

In what follows we define $\{Y_k\}_{k=0}^{N-1}$ such that $\{Y_k\}_{k=0}^{N-1}$ approximates $\{\mathbf{X}_{T-t_k}\}_{k=0}^{N-1}$ for a specific sequence of times $\{t_k\}_{k=0}^{N-1} \in [0, T]^N$. We recall that the time-reversal of (S38) is associated with the following SDE

$$\mathrm{d}\mathbf{Y}_t = \{\alpha \mathbf{Y}_t + 2\nabla \log p_{T-t}(\mathbf{Y}_t)\}\mathrm{d}t + \sqrt{2}\mathrm{d}\mathbf{B}_t. \tag{S39}$$

In what follows, we fix $\alpha = 1$. The Euler-Maruyama discretization of (S39) yields for any $k \in \{0, \ldots, N-1\}$

$$\tilde{Y}_{k+1} = (1 + \gamma_{k+1})\tilde{Y}_k + 2\gamma_{k+1}\nabla \log p_{T-t_k}(\tilde{Y}_k) + \sqrt{2\gamma_{k+1}}Z_{k+1}.$$

where $\{\gamma_{k+1}\}_{k=0}^{N-1}$ is a sequence of stepsizes and for any $k \in \{0, \ldots, N-1\}$, $t_k = \sum_{j=0}^{k-1} \gamma_{j+1}$. A close form for $\{\nabla \log p_{T-t_k}\}_{k=0}^{N-1}$ is not available and in practice we consider

$$Y_{k+1} = (1 + \gamma_{k+1})Y_k + 2\gamma_{k+1}s_{\theta^\star}(T-t_k, Y_k)\mathrm{d}t + \sqrt{2\gamma_{k+1}}Z_{k+1}. \tag{S40}$$

In (Ho et al., 2020, Equation 11) the backward recursion is given for any $k \in \{0, \ldots, N-1\}$

$$Y_{k+1} = \alpha_{N-k}^{-1/2}(Y_k - \beta_{N-k}/(1 - \bar{\alpha}_{N-k})^{1/2}\boldsymbol{\epsilon}_\theta(Y_k, T-t_k)) + \sigma_{N-k}Z_{k+1}. \tag{S41}$$

In (S41) we set $\sigma_k^2 = \beta_k$ as suggested in Ho et al. (2020) where for any $k \in \{0, \ldots, N-1\}$

$$\sigma_{k+1}^2 = \beta_{k+1}, \quad \alpha_{k+1} = 1 - \beta_{k+1}, \quad \bar{\alpha}_{k+1} = \prod_{i=1}^{k+1} \alpha_i.$$

We consider a first-order expansion of (S41) with respect to $\{\beta_{k+1}\}_{k=0}^{N-1}$. We obtain the following recursion for any $k \in \{0, \ldots, N-1\}$

$$Y_{k+1} = (1 + \beta_{N-k}/2)Y_k - \beta_{N-k}/(1 - \bar{\alpha}_{N-k})^{1/2}\boldsymbol{\epsilon}_\theta(Y_k, T-t_k) + \sqrt{\beta_{N-k}}Z_{k+1}.$$

This last recursion is equivalent to (S40) upon setting $\beta_{N-k} \leftarrow 2\gamma_{k+1}$ and $-\boldsymbol{\epsilon}_\theta(\cdot, T-t_k)/(1 - \bar{\alpha}_{N-k})^{1/2} \leftarrow -s_{\theta^\star}(T-t_k, \cdot)$. It remains to show that $s_{\theta^\star}$ is the solution to the same regression problem as $\boldsymbol{\epsilon}_\theta/(1 - \bar{\alpha}_{N-\cdot})$ in (Ho et al., 2020, Equation 12). First, note that for any $t > 0$ and $x_t \in \mathbb{R}^d$ we have

$$p_t(x_t) = (2\pi\bar{\sigma}_t^2)^{-d/2} \int_{\mathbb{R}^d} p_{\text{data}}(x_0) \exp[-\|x_t - c_t x_0\|^2/(2\bar{\sigma}_t^2)]\mathrm{d}x_0,$$

with
$$c_t^2 = \exp[-2t], \qquad \bar{\sigma}_t^2 = 1 - \exp[-2t].$$
Therefore we get that for any $t \in [0, T]$ and $x_t \in \mathbb{R}^d$
$$\nabla \log p_t(x_t) = \int_{\mathbb{R}^d} (c_t x_0 - x_t) p_{\text{data}}(x_0) \exp[-\|x_t - c_t x_0\|^2 / (2\bar{\sigma}_t^2)] \mathrm{d}x_0$$
$$= \mathbb{E}[c_t \mathbf{X}_0 - \mathbf{X}_t | \mathbf{X}_t = x_t] / \bar{\sigma}_t^2 = -\mathbb{E}[\mathbf{Z} | \mathbf{X}_t = x_t] / \bar{\sigma}_t,$$
where we recall that $\mathbf{X}_t$ has the same distribution as $c_t \mathbf{X}_0 + \bar{\sigma}_t \mathbf{Z}$, with $\mathbf{Z}$ a $d$-dimensional Gaussian random variable with zero mean and identity covariance matrix. Hence, we have that $\theta^\star$ satisfies the following regression problem
$$\theta^\star = \arg\min_\theta \sum_{k=0}^{N-1} \lambda(k) \mathbb{E}[\|\mathbf{Z}/\sigma_{T-t_k} + s_\theta(T - t_k, \mathbf{X}_{T-t_k})\|].$$
Note that we have
$$\sum_{i=1}^{N-k} \beta_i = \sum_{i=k}^{N-1} \beta_{N-i} = 2 \sum_{i=k}^{N-1} \gamma_{i+1} = 2(T - t_k).$$
Using this result we have for any $k \in \{0, \ldots, N-1\}$
$$1 - \bar{\alpha}_{N-k} = 1 - \exp[-\sum_{i=1}^{N-k} \log(1 - \beta_i)] \approx 1 - \exp[-\sum_{i=1}^{N-k} \beta_i] \approx \bar{\sigma}_{T-t_k}^2.$$
Let $\tilde{\theta}^\star$ the minimizer of (Ho et al., 2020, Equation 12) we have
$$\tilde{\theta}^\star \approx \arg\min_\theta \sum_{k=0}^{N-1} (2\alpha_{N-k}(1 - \alpha_{N-k}))^{-1} \mathbb{E}[\|\mathbf{Z} - \boldsymbol{\epsilon}_\theta(\mathbf{X}_{T-t_k}, T - t_k)\|^2]$$
$$\approx \arg\min_\theta \sum_{k=0}^{N-1} (2\alpha_{N-k})^{-1} \mathbb{E}[\|\mathbf{Z}/(1 - \alpha_{N-k})^{1/2} - \boldsymbol{\epsilon}_\theta(\mathbf{X}_{T-t_k}, T - t_k)/(1 - \alpha_{N-k})^{1/2}\|^2]$$
$$\approx \arg\min_\theta \sum_{k=0}^{N-1} (2\alpha_{N-k})^{-1} \mathbb{E}[\|\mathbf{Z}/\bar{\sigma}_{T-t_k} + s_\theta(T - t_k, \mathbf{X}_{T-t_k})\|^2].$$
Hence the two regression problems are approximately the same (for small values of $\{\beta_{k+1}\}_{k=0}^{N-1}$) if we set $\lambda(k) = (2\alpha_{N-k})^{-1}$.

## S4  Schrödinger bridges with potentials and DSB recursion

In this section, we start by proving an additive formula for the Kullback–Leibler divergence in Section S4.1 following Léonard (2014a). We recall the classical IPF formulation using potentials in Section S4.2. Then, Proposition 2 is proved in Section S4.3. Finally, we highlight a link between our formulation and autoencoders in Section S4.4.

### S4.1  Additive formula for the Kullback–Leibler divergence

In this section, we prove a formula for the Kullback–Leibler divergence following the proof of Léonard (2014a) which extends the result to unbounded measures defined on the space of right-continuous left-limited functions from $[0, T]$. We recall that a Polish space is a complete metric separable space.

We start with the following disintegration theorem for probability measures.

**Theorem S16.** *Let $(\mathsf{X}, \mathcal{X})$ and $(\mathsf{Y}, \mathcal{Y})$ be two Polish spaces. Let $\pi \in \mathscr{P}(\mathsf{X})$ and $\varphi : \mathsf{X} \to \mathsf{Y}$ measurable. Then there exists a Markov kernel $\mathrm{K}_\varphi^\pi : \mathsf{Y} \times \mathcal{X} \to [0, 1]$ such that the following hold:*

*(a) For any $y \in \mathsf{Y}$, $\mathrm{K}_\varphi^\pi(y, \varphi^{-1}(\{y\})) = 1$.*

*(b) For any $f : \mathsf{X} \to [0, +\infty)$ measurable we have $\int_\mathsf{X} f(x)\mathrm{d}\pi(x) = \int_\mathsf{Y} \mathrm{K}_\varphi^\pi(y, f)\mathrm{d}\pi_\varphi(y),$*

*where $\pi_\varphi = \varphi_\#\pi$.*

*Proof.* See (Dellacherie and Meyer, 1988, III-70) for instance. □

$\mathrm{K}_\varphi^\pi$ is called the disintegration of $\pi$ w.r.t. $\varphi$ and is unique, see (Dellacherie and Meyer, 1988, III-70). In particular, for any $\mathsf{X}$-valued random variable $X$ with distribution $\pi$ we have $\mathbb{E}[f(X)|\varphi(X)] = \mathrm{K}_\varphi^\pi(\varphi(X), f)$. Next we prove the following proposition, see (Léonard, 2014a, Proposition A.13) for an extension to unbounded measures. In what follows, for any $\varphi : \mathsf{X} to \mathbb{R}$ measurable we denote $\pi_\varphi = \varphi_\#\pi$.

**Proposition S17.** *Let* $(\mathsf{X}, \mathcal{X})$ *and* $(\mathsf{Y}, \mathcal{Y})$ *be two Polish spaces. Let* $\pi, \mu \in \mathscr{P}(\mathsf{X})$ *and* $\varphi : \mathsf{X} \to \mathsf{Y}$ *measurable. Assume that* $\pi \ll \mu$. *Then the following holds:*

*(a)* $\pi_\varphi \ll \mu_\varphi$

*(b)* *There exists* $\mathsf{A} \in \mathcal{Y}$ *with* $\pi_\varphi(\mathsf{A}) = 1$ *such that for any* $y \in \mathsf{A}$, $\mathrm{K}^\pi_\varphi(y, \cdot) \ll \mathrm{K}^\mu_\varphi(y, \cdot)$.

*In addition, we have for any* $y \in \mathsf{Y}$, $y' \in \mathsf{A}$ *and* $x \in \mathsf{X}$

$$(\mathrm{d}\pi_\varphi/\mathrm{d}\mu_\varphi)(y) = \mathrm{K}^\mu_\varphi(y, (\mathrm{d}\pi/\mathrm{d}\mu)), \quad (\mathrm{d}\mathrm{K}^\pi_\varphi(y', \cdot)/\mathrm{d}\mathrm{K}^\mu_\varphi(y', \cdot))(x) = (\mathrm{d}\pi/\mathrm{d}\mu)(x)/(\mathrm{d}\pi_\varphi/\mathrm{d}\mu_\varphi)(y').$$

*Finally, there exists* $\mathsf{C} \in \mathcal{X}$ *with* $\pi(\mathsf{C}) = 1$ *such that for any* $x \in \mathsf{C}$ *we have*

$$(\mathrm{d}\pi/\mathrm{d}\mu)(x) = (\mathrm{d}\pi_\varphi/\mathrm{d}\mu_\varphi)(\varphi(x))(\mathrm{d}\mathrm{K}^\pi_\varphi(\varphi(x), \cdot)/\mathrm{d}\mathrm{K}^\mu_\varphi(\varphi(x), \cdot))(x).$$

*Proof.* Let $f : \mathsf{X} \to [0, +\infty)$ measurable. Using Theorem S16 we have

$$\pi_\varphi[f] = \int_\mathsf{X} f(\varphi(x))\mathrm{d}\pi(x) = \int_\mathsf{X} f(\varphi(x))(\mathrm{d}\pi/\mathrm{d}\mu)(x)\mathrm{d}\mu(x) = \int_\mathsf{X} f(y)\mathrm{K}^\mu_\varphi(y, (\mathrm{d}\pi/\mathrm{d}\mu))\mathrm{d}\mu_\varphi(y),$$

which concludes the first part of the proof. For the second part of the proof, let $\mathsf{B} = \{y \in \mathsf{Y} : (\mathrm{d}\pi_\varphi/\mathrm{d}\mu_\varphi)(y) = 0\}$. We have

$$0 = \int_\mathsf{Y} \mathbb{1}_\mathsf{B}(y)(\mathrm{d}\pi_\varphi/\mathrm{d}\mu_\varphi)(y)\mathrm{d}\mu_\varphi(y) = \pi_\varphi(\mathsf{B}).$$

Therefore, there exists $\mathsf{A}_1 \in \mathcal{Y}$ such that $\pi_\varphi(\mathsf{A}_1) = 1$ and for any $y \in \mathsf{A}_1$, $(\mathrm{d}\pi_\varphi/\mathrm{d}\mu_\varphi)(y) > 0$. Let $g : \mathsf{Y} \to [0, +\infty)$. Using Theorem S16 we have

$$\int_\mathsf{X} g(\varphi(x))f(x)\mathrm{d}\pi(x) = \int_\mathsf{X} g(\varphi(x))f(x)(\mathrm{d}\pi/\mathrm{d}\mu)(x)\mathrm{d}\mu(x) = \int_\mathsf{Y} g(y)\mathrm{K}^\mu_\varphi(y, f \times (\mathrm{d}\pi/\mathrm{d}\mu))\mathrm{d}\mu_\varphi(y).$$

Similarly, using Theorem S16 we have

$$\int_\mathsf{X} g(\varphi(x))f(x)\mathrm{d}\pi(x) = \int_\mathsf{Y} g(y)\mathrm{K}^\pi_\varphi(y, f)\mathrm{d}\pi_\varphi(y) = \int_\mathsf{Y} g(y)\mathrm{K}^\pi_\varphi(y, f)(\mathrm{d}\pi_\varphi/\mathrm{d}\mu_\varphi)(y)\mathrm{d}\pi_\varphi(y).$$

Hence, we get that there exists $\mathsf{A}_2 \in \mathcal{Y}$ with $\mu_\varphi(\mathsf{A}_2) = 1$ (hence $\pi_\varphi(\mathsf{A}_2) = 1$) such that for any $y \in \mathsf{A}_2$ we have

$$\mathrm{K}^\pi_\varphi(y, f)(\mathrm{d}\pi_\varphi/\mathrm{d}\mu_\varphi)(y) = \mathrm{K}^\mu_\varphi(y, f \times (\mathrm{d}\pi/\mathrm{d}\mu)).$$

We conclude upon letting $\mathsf{A} = \mathsf{A}_1 \cap \mathsf{A}_2$ and using the fact that for any $y \in \mathsf{A}$, $(\mathrm{d}\pi_\varphi/\mathrm{d}\mu_\varphi)(y) > 0$. Finally, since $\pi_\varphi(\mathsf{A}) = 1$ if and only if $\pi(\varphi^{-1}(\mathsf{A})) = 1$, we have for any $x \in \varphi^{-1}(\mathsf{A})$

$$(\mathrm{d}\pi/\mathrm{d}\mu)(x) = (\mathrm{d}\pi_\varphi/\mathrm{d}\mu_\varphi)(\varphi(x))(\mathrm{d}\mathrm{K}^\pi_\varphi(\varphi(x), \cdot)/\mathrm{d}\mathrm{K}^\mu_\varphi(\varphi(x), \cdot))(x),$$

which concludes the proof. $\qquad\square$

We are now ready to state the additive formula.

**Proposition S18.** *Let* $(\mathsf{X}, \mathcal{X})$ *and* $(\mathsf{Y}, \mathcal{Y})$ *be two Polish spaces and* $\pi, \mu \in \mathscr{P}(\mathsf{X})$ *with* $\pi \ll \mu$. *Then for any* $\varphi : \mathsf{X} \to \mathsf{Y}$ *we have*

$$\mathrm{KL}(\pi|\mu) = \mathrm{KL}(\pi_\varphi|\mu_\varphi) + \int_\mathsf{Y} \mathrm{KL}(\mathrm{K}^\pi_\varphi(y, \cdot)|\mathrm{K}^\mu_\varphi(y, \cdot))\mathrm{d}\pi_\varphi(y).$$

*Proof.* First assume that $\int_\mathsf{X} |\log((\mathrm{d}\pi/\mathrm{d}\mu)(x))| \, \mathrm{d}\pi(x) = +\infty$. Then, using Proposition S17 we have $\int_\mathsf{X} |\log((\mathrm{d}\pi_\varphi/\mathrm{d}\mu_\varphi)(\varphi(x)))| \, \mathrm{d}\pi(x) = +\infty$ or $\int_\mathsf{X} |\log((\mathrm{d}\mathrm{K}^\pi_\varphi(\varphi(x), \cdot)/\mathrm{d}\mathrm{K}^\mu_\varphi(\varphi(x), \cdot))(x))| \, \mathrm{d}\pi(x) = +\infty$, *i.e.* either $\mathrm{KL}(\pi_\varphi|\mu_\varphi) = +\infty$ or $\int_\mathsf{X} \mathrm{KL}(\mathrm{K}^\pi_\varphi(\varphi(x), \cdot)|\mathrm{K}^\mu_\varphi(\varphi(x), \cdot))\mathrm{d}\pi(x) = +\infty$ using Theorem S16, which concludes the first part of the proof. Second, assume that $\int_\mathsf{X} |\log((\mathrm{d}\pi/\mathrm{d}\mu)(x))| \, \mathrm{d}\pi(x) < +\infty$. Using Pinsker's inequality (Bakry et al., 2014, Equation 5.2.2) we get that $\mathrm{KL}(\pi_\varphi|\mu_\varphi) < +\infty$, *i.e.* $\int_\mathsf{X} |\log((\mathrm{d}\pi_\varphi/\mathrm{d}\mu_\varphi)(\varphi(x)))| \, \mathrm{d}\pi(x) < +\infty$. Hence, we get that $\int_\mathsf{X} |\log((\mathrm{d}\mathrm{K}^\pi_\varphi(\varphi(x), \cdot)/\mathrm{d}\mathrm{K}^\mu_\varphi(\varphi(x), \cdot))(x))| \, \mathrm{d}\pi(x) < +\infty$. Therefore we have

$$\mathrm{KL}(\pi|\mu) = \mathrm{KL}(\pi_\varphi|\mu_\varphi) + \int_\mathsf{Y} \mathrm{KL}(\mathrm{K}^\pi_\varphi(y, \cdot)|\mathrm{K}^\mu_\varphi(y, \cdot))\mathrm{d}\pi_\varphi(y)$$

which concludes the proof $\qquad\square$

We emphasize that in the case where $\mathsf{X} = \mathbb{R}^d \times \mathbb{R}^d$, $\varphi = \mathrm{proj}_0$ the projection on the first variable and $\pi$, $\mu$ admit densities w.r.t. the Lebesgue measure denoted $p$ and $q$ such that for any $x, y \in \mathbb{R}^d$, $p(x, y) = p_0(x)p_{1|0}(y|x)$ and $q(x, y) = q_0(x)q_{1|0}(y|x)$ then one can avoid using disintegration theory and Proposition S18 can be proved directly.

## S4.2 Iterative Proportional Fitting via potentials

In this section, before recalling the usual definition of the IPF via potentials we provide a condition under which the IPF sequence is well-defined which is used throughout Section 3.2.

**Proposition S19.** *Assume that there exists $\tilde{\pi} \in \mathscr{P}_{N+1}$ such that $\tilde{\pi}_0 = p_{\text{data}}$, $\tilde{\pi}_N = p_{\text{prior}}$ and $\mathrm{KL}(\tilde{\pi}|\pi^0) < +\infty$. Then the IPF sequence is well-defined.*

*Proof.* We prove the existence of the IPF sequence by recursion. First, note that $\pi^1$ is well-defined since $\tilde{\pi} \in \mathscr{P}_{N+1}$ with $\tilde{\pi}_N = p_{\text{prior}}$ and $\mathrm{KL}(\tilde{\pi}|\pi^0) < +\infty$. Second, assume that the sequence is well-defined up to $n$ with $n \in \mathbb{N}$. Using (Csiszár, 1975, Theorem 2.2) we have

$$\mathrm{KL}(\tilde{\pi}|\pi^0) = \mathrm{KL}(\tilde{\pi}|\pi^n) + \sum_{j=0}^{n-1} \mathrm{KL}(\pi^{j+1}|\pi^j).$$

Hence $\mathrm{KL}(\tilde{\pi}|\pi^n) < +\infty$. Using that $\tilde{\pi}_0 = p_{\text{data}}$ if $n$ is odd and that $\tilde{\pi}_N = p_{\text{prior}}$ if $n$ is even, we get that $\pi^{n+1}$ is well-defined, which concludes the proof. $\square$

We now introduce the IPF using potentials. This construction is not new and can be found in Bernton et al. (2019); Chen et al. (2016, 2021); Pavon et al. (2021); Peyré and Cuturi (2019) for instance (in continuous state spaces). In discrete settings the recursion can be found in the following earlier works Kruithof (1937); Deming and Stephan (1940); Fortet (1940); Sinkhorn and Knopp (1967); Kullback (1968); Ruschendorf et al. (1995). The IPF is defined by the following recursion $\pi^0 = p$ given in (1) and for $n \geq 0$

$$\pi^{2n+1} = \arg\min \left\{ \mathrm{KL}(\pi|\pi^{2n}) \; : \; \pi \in \mathscr{P}_{N+1}, \; \pi_N = p_{\text{prior}} \right\},$$
$$\pi^{2n+2} = \arg\min \left\{ \mathrm{KL}(\pi|\pi^{2n+1}) \; : \; \pi \in \mathscr{P}_{N+1}, \; \pi_0 = p_{\text{data}} \right\}.$$

In the classical IPF presentation we obtain under mild assumptions that $\pi^{2n+1}$ admits a density $q^n$ w.r.t the Lebesgue measure and that $\pi^{2n}$ admits a density $p^n$ w.r.t the Lebesgue measure, given by the following expressions

$$q^n(x_{0:N}) = p_{\text{data}}^n(x_0) \prod_{k=0}^{N-1} p_{k+1|k}^{n+1}(x_{k+1}|x_k), \tag{S42}$$
$$p^{n+1}(x_{0:N}) = p_{\text{data}}(x_0) \prod_{k=0}^{N-1} p_{k+1|k}^{n+1}(x_{k+1}|x_k),$$

where $(p_{\text{data}}^n(x_0))_{n \in \mathbb{N}}$ and $(p_{k+1|k}^n(x_{k+1}|x_k))_{n \in \mathbb{N}}$ are densities which are iteratively computed, with $p_{k+1|k}^0 = p_{k+1|k}$.

In the context of generative modelling the derivation (S42) is not useful because it does not provide a generative model, *i.e.* a probabilistic transition from $p_{\text{prior}}$ to $p_{\text{data}}$ but instead defines a transition from $p_{\text{data}}$ to $p_{\text{prior}}$. Therefore, in this section only, we reverse the roles of $p_{\text{prior}}$ and $p_{\text{data}}$ and consider a reference density $\bar{p}$ such that for any $x_{0:N} \in \mathcal{X}$ we have

$$\bar{p}(x_{0:N}) = p_{\text{prior}}(x_0) \prod_{k=0}^{N-1} \bar{p}_{k+1|k}(x_{k+1}|x_k). \tag{S43}$$

Then, we consider the following recursion $\pi^0 = \bar{p}$ given in (S43) and for $n \in \mathbb{N}$

$$\pi^{2n+1} = \arg\min \left\{ \mathrm{KL}(\pi|\pi^{2n}) \; : \; \pi \in \mathscr{P}_{N+1}, \; \pi_N = p_{\text{data}} \right\}, \tag{S44}$$
$$\pi^{2n+2} = \arg\min \left\{ \mathrm{KL}(\pi|\pi^{2n+1}) \; : \; \pi \in \mathscr{P}_{N+1}, \; \pi_0 = p_{\text{prior}} \right\}.$$

Again, we emphasize that the roles of $p_{\text{prior}}$ and $p_{\text{data}}$ are exchanged in this formulation. Using the classical IPF presentation we obtain the following expressions under mild assumptions

$$\bar{q}^n(x_{0:N}) = p_{\text{prior}}^n(x_0) \prod_{k=0}^{N-1} \bar{p}^{n+1}(x_{k+1}|x_k), \tag{S45}$$
$$\bar{p}^{n+1}(x_{0:N}) = p_{\text{prior}}(x_0) \prod_{k=0}^{N-1} \bar{p}^{n+1}(x_{k+1}|x_k).$$

In this case, we get that $\pi^{2n+1}$ (approximately) defines a generative model for large values of $n \in \mathbb{N}$ since it provides a transition from to $p_{\text{prior}}$ to (approximately) $p_{\text{data}}$. In the following proposition we give the precise statement corresponding to (S45). We assume that $\bar{p}^0 = \bar{p}$.

**Proposition S20.** *Assume that* $\mathrm{KL}(p_{\mathrm{prior}} \otimes p_{\mathrm{data}} | \bar{p}_{0,N}) < +\infty$. *Then* $(\pi^n)_{n \in \mathbb{N}}$ *given by* (S44) *is well-defined and for any* $n \in \mathbb{N}$ *we have that* $\pi^{2n+1}$ *and* $\pi^{2n+2}$ *admit a density w.r.t. the Lebesgue measures denoted* $\bar{q}^n$ *and* $\bar{p}^{n+1}$. *In addition, we have for any* $n \in \mathbb{N}$ *and* $x_{0:N} \in \mathcal{X}$

$$\bar{q}^n(x_{0:N}) = p_{\mathrm{prior}}^n(x_0) \textstyle\prod_{k=0}^{N-1} \bar{p}^{n+1}(x_{k+1}|x_k),$$

$$\bar{p}^{n+1}(x_{0:N}) = p_{\mathrm{prior}}(x_0) \textstyle\prod_{k=0}^{N-1} \bar{p}^{n+1}(x_{k+1}|x_k),$$

*where for any* $n \in \mathbb{N}$ *we have for any* $x_{0:N} \in \mathcal{X}$ *and* $k \in \{0, \ldots, N-1\}$

$$p_{\mathrm{prior}}^n(x_0) = \psi_0^n(x_0) p_{\mathrm{prior}}(x_0), \quad \bar{p}^{n+1}(x_{k+1}|x_k) = \bar{p}^n(x_{k+1}|x_k)\psi_{k+1}^n(x_{k+1})/\psi_k^n(x_k),$$

*with*

$$\psi_N^n(x_N) = p_{\mathrm{data}}(x_N)/\bar{p}_N^n(x_N), \quad \psi_k^n(x_k) = \textstyle\int_{\mathbb{R}^d} \psi_{k+1}^n(x_{k+1})\bar{p}^n(x_{k+1}|x_k)\mathrm{d}x_{k+1}.$$

*Proof.* Let $\tilde{\pi} = (p_{\mathrm{prior}} \otimes p_{\mathrm{data}})\bar{p}_{|0,N}$. Using Proposition S18 we get that $\mathrm{KL}(\tilde{\pi}|\bar{p}) = \mathrm{KL}(p_{\mathrm{prior}} \otimes p_{\mathrm{data}}|\bar{p}_{0,N}) < +\infty$. Using Proposition S19 the IPF sequence is well-defined. In addition, using (Csiszár, 1975, Theorem 3.1) for any $n \in \mathbb{N}$ there exists $\psi_N^n : \mathbb{R}^d \to [0, +\infty)$ such that for any $x_{0:N} \in \mathsf{A}$ with $\tilde{\pi}(\mathsf{A}) = 1$ we have

$$\bar{q}^n(x_{0:N}) = \bar{p}^n(x_{0:N})\psi_N^n(x_N).$$

Since $\tilde{\pi}$ is equivalent to the Lebesgue measure we get that for any $x_{0:N} \in \mathbb{R}^d$

$$\bar{q}^n(x_{0:N}) = \bar{p}^n(x_{0:N})\psi_N^n(x_N).$$

Let $n \in \mathbb{N}$. We have for any $x_N \in \mathbb{R}^d$, $p_{\mathrm{data}}(x_N) = \bar{q}^n(x_N) = \bar{p}_N^n(x_N)\psi_N^n(x_N)$. Hence, we get that for any $N \in \mathbb{N}$, $\psi_N^n(x_N) = p_{\mathrm{data}}(x_N)/\bar{p}_N^n(x_N)$. For any $x_{0:N} \in \mathcal{X}$ and $k \in \{0, \ldots, N-1\}$ let

$$\psi_k^n(x_k) = \textstyle\int_{\mathbb{R}^d} \psi_{k+1}^n(x_{k+1})\bar{p}^n(x_{k+1}|x_k)\mathrm{d}x_{k+1}.$$

We obtain that for any $x_{0:N} \in \mathcal{X}$

$$\bar{q}^n(x_{0:N}) = p_{\mathrm{prior}}(x_0)\psi_0(x_0) \textstyle\prod_{k=0}^{N-1}(\bar{p}^n(x_{k+1}|x_k)\psi_{k+1}(x_{k+1})/\psi_k(x_k)).$$

Hence, we get that for any $x_{0:N} \in \mathcal{X}$, $\bar{q}^n(x_0) = p_{\mathrm{prior}}^n(x_0) \prod_{k=0}^{N-1} \bar{p}^{n+1}(x_{k+1}|x_k)$. Using Proposition S18 we get that for any $x_{0:N} \in \mathcal{X}$, $\bar{p}^{n+1}(x_0) = p_{\mathrm{prior}}(x_0) \prod_{k=0}^{N-1} \bar{p}^{n+1}(x_{k+1}|x_k)$, which concludes the proof. $\qquad\square$

The previous expression is not symmetric and the IPF iterations appear as a policy refinement of the original forward dynamic $\bar{p}$. In the next proposition we present another potential formulation of the IPF iterations which is symmetric.

**Proposition S21.** *Assume that* $\mathrm{KL}(p_{\mathrm{prior}} \otimes p_{\mathrm{data}} | q_{0,N}) < +\infty$. *Then* $(\pi^n)_{n \in \mathbb{N}}$ *given by* (S44) *is well-defined and for any* $n \in \mathbb{N}$ *we have that* $\pi^{2n+1}$ *and* $\pi^{2n+2}$ *admit a density w.r.t. the Lebesgue measures denoted* $\bar{q}^n$ *and* $\bar{p}^{n+1}$. *In addition, we have for any* $n \in \mathbb{N}$ *and* $x_{0:N} \in \mathcal{X}$

$$\bar{q}^n(x_{0:N}) = \varphi_0^n(x_0) \textstyle\prod_{k=0}^{N-1} \bar{p}(x_{k+1}|x_k)\psi_N^n(x_N),$$

$$\bar{p}^{n+1}(x_{0:N}) = \varphi_0^{n+1}(x_0) \textstyle\prod_{k=0}^{N-1} \bar{p}(x_{k+1}|x_k)\psi_N^n(x_N),$$

*where for any* $n \in \mathbb{N}$ *we have for any* $x_{0:N} \in \mathcal{X}$ *and* $k \in \{0, \ldots, N-1\}$

$$\psi_N^n(x_N) = p_{\mathrm{data}}(x_N)/\varphi_N^n(x_N), \quad \psi_k^n(x_k) = \textstyle\int_{\mathbb{R}^d} \psi_{k+1}^n(x_{k+1})\bar{p}(x_{k+1}|x_k)\mathrm{d}x_{k+1},$$

$$\varphi_0^{n+1}(x_0) = p_{\mathrm{prior}}(x_0)/\psi_0^n(x_0), \quad \varphi_{k+1}^{n+1}(x_{k+1}) = \textstyle\int_{\mathbb{R}^d} \varphi_k^{n+1}(x_k)\bar{p}(x_{k+1}|x_k)\mathrm{d}x_k,$$

*and* $\varphi_0^0 = p_{\mathrm{prior}}$ *and* $\psi_N^{-1} = 1$.

*Proof.* Let $\tilde{\pi} = (p_{\mathrm{prior}} \otimes p_{\mathrm{data}})q_{|0,N}$. Using Proposition S18 we get that $\mathrm{KL}(\tilde{\pi}|q) = \mathrm{KL}(p_{\mathrm{prior}} \otimes p_{\mathrm{data}}|p_{0,N}) < +\infty$. Using Proposition S19 the IPF sequence is well-defined. In addition, using (Csiszár, 1975, Theorem 3.1) for any $n \in \mathbb{N}$ there exists $\psi_N^n : \mathbb{R}^d \to [0, +\infty)$ such that for any $x_{0:N} \in \mathsf{A}$ with $\tilde{\pi}(\mathsf{A}) = 1$ we have

$$\bar{q}^n(x_{0:N}) = \bar{p}^n(x_{0:N})\tilde{\psi}_N^n(x_N), \qquad \bar{p}^{n+1}(x_{0:N}) = \bar{q}^n(x_{0:N})\tilde{\varphi}_0^n(x_0).$$

Since $\tilde{\pi}$ is equivalent to the Lebesgue measure we get that for any $x_{0:N} \in \mathbb{R}^d$

$$\bar{q}^n(x_{0:N}) = \bar{p}^n(x_{0:N})\tilde{\psi}_N^n(x_N), \qquad \bar{p}^{n+1}(x_{0:N}) = \bar{q}^n(x_{0:N})\tilde{\varphi}_0^n(x_0).$$

For any $n \in \mathbb{N}$, let $\psi_N^n = \psi_N^{n-1}\tilde{\psi}_N^n$ and $\varphi_0^{n+1} = \varphi_0^n\tilde{\varphi}_0^n$. By recursion, we get that for any $n \in \mathbb{N}$ and $x_{0:N} \in \mathcal{X}$

$$\bar{q}^n(x_{0:N}) = \varphi_0^n(x_0) \prod_{k=0}^{N-1} \bar{p}(x_{k+1}|x_k)\psi_N^n(x_N),$$

$$\bar{p}^{n+1}(x_{0:N}) = \varphi_0^{n+1}(x_0) \prod_{k=0}^{N-1} \bar{p}(x_{k+1}|x_k)\psi_N^n(x_N).$$

Let $n \in \mathbb{N}$. For any $x_N \in \mathbb{R}^d$ we have

$$\bar{q}_N^n(x_N) = p_{\text{data}}(x_N) = \bar{p}_N^n(x_N)\tilde{\psi}_N^n(x_n). \tag{S46}$$

In addition, for any $k \in \{0, \dots, N-1\}$ and $x_{0:N} \in \mathcal{X}$ we define $\varphi_{k+1}^{n+1}(x_{k+1}) = \int_{\mathbb{R}^d} \varphi_k^{n+1}(x_k)\bar{p}(x_{k+1}|x_k)\mathrm{d}x_k$. We have for any $x_N \in \mathbb{R}^d$, $\bar{p}_N^n(x_N) = \varphi_N^n(x_N)\psi_N^{n-1}(x_n)$. Combining this result with (S46) we get that for any $x_N \in \mathbb{R}^d$

$$\psi_N^n(x_N) = p_{\text{data}}(x_N)/\varphi_N^n(x_N).$$

Similarly, we get that for any $x_0 \in \mathbb{R}^d$, $\varphi_0^{n+1}(x_0) = p_{\text{prior}}(x_0)/\psi_0^n(x_0)$, which concludes the proof. $\qquad\square$

### S4.3   Proof of Proposition 2

Let $\tilde{\pi} = (p_{\text{prior}} \otimes p_{\text{data}})p_{|0,N}$. Using Proposition S18 we get that $\mathrm{KL}(\tilde{\pi}|p) = \mathrm{KL}(p_{\text{prior}} \otimes p_{\text{data}}|p_{0,N}) < +\infty$. Using Proposition S19 the IPF sequence is well-defined. Note that $\pi^0$ admits a density w.r.t. the Lebesgue measure given by $p > 0$. Let $n \in \mathbb{N}$ and assume that $p^n > 0$ is given for any $x_{0:N} \in \mathcal{X}$ by

$$p^n(x_{0:N}) = p_{\text{data}}(x_0) \prod_{k=0}^{N-1} q^{n-1}(x_{k+1}|x_k). \tag{S47}$$

Using Proposition S18 we get that for any $\pi \in \mathscr{P}_{N+1}$ such that $\pi_N = p_{\text{prior}}$ we have

$$\mathrm{KL}(\pi|\pi^{2n}) = \mathrm{KL}(p_{\text{prior}}|\pi_0^{2n}) + \int_{\mathbb{R}^d} \mathrm{KL}(\pi_{|N}|\pi_{|N}^{2n})p_{\text{prior}}(x_N)\mathrm{d}x_N.$$

Hence, we have that $\pi^{2n+1} = p_{\text{prior}}\pi_{|N}^{2n}$. Since $p^n > 0$ we get that for any $\pi_{|N}^{2n}$ satisfies for any $\mathsf{A} \in \mathcal{B}(\mathcal{X})$ and $x_N \in \mathbb{R}^d$

$$\pi_{|N}^{2n}(\mathsf{A}|x_N) = \int_{\mathsf{A}} p^n(x_{0:N})/p^n(x_N)\mathrm{d}x_{0:N}\delta_{x_N}(\mathsf{A}_N).$$

Therefore, $\pi^{2n+1}$ admits a density w.r.t. the Lebesgue measure denoted $q^n$ and given for any $x_{0:N} \in \mathcal{X}$ by

$$q^n(x_{0:N}) = p^n(x_{0:N})p_{\text{prior}}(x_N)/p^n(x_N)$$

$$= p_{\text{prior}}(x_N) \prod_{k=0}^{N-1} p^n(x_{k+1}|x_k)p^n(x_k)/p^n(x_{k+1}) = p_{\text{prior}}(x_N) \prod_{k=0}^{N-1} p^n(x_k|x_{k+1}),$$

where we have used (S47). Note that $q^n > 0$. Similarly, we get that for any $x_{0:N} \in \mathcal{X}$

$$p^{n+1}(x_{0:N}) = p_{\text{data}}(x_0) \prod_{k=0}^{N-1} q^n(x_{k+1}|x_k).$$

Note that again that $p^{n+1} > 0$. We conclude by recursion.

### S4.4   Link with autoencoders

Consider the maximum likelihood problem

$$q^\star = \arg\max\{\mathbb{E}_{p_{\text{data}}}[\log q_0(X_0)] \,:\, q \in \mathscr{P}_d(\mathcal{X}), \ q_N = p_{\text{prior}}\},$$

where $\mathscr{P}_d(\mathcal{X})$ is the subset of the probability distribution over $\mathcal{X}$ which admit a density w.r.t. the Lebesgue measure. Using Jensen's inequality we have for any $q \in \mathscr{P}_d(\mathcal{X})$

$$\mathbb{E}_{p_{\text{data}}}[\log q_0(X_0)] = \int_{\mathbb{R}^d} \log\left(\int_{(\mathbb{R}^d)^{N-1}} q(x_{0:N})p(x_{1:N}|x_0)/p(x_{1:N}|x_0)\mathrm{d}x_{1:N}\right)p_0(x_0)\mathrm{d}x_0$$

$$\geq \int_{\mathcal{X}} \log(q(x_{0:N})/p(x_{1:N}|x_0))p(x_{0:N})\mathrm{d}x_{0:N} \geq -\mathrm{KL}(p|q) - \mathrm{H}(p_0).$$

This Evidence Lower Bound (ELBO) is similar to the one identified in Ho et al. (2020). Maximizing this ELBO is equivalent to solving the following problem

$$q^0 = \arg\min\{\mathrm{KL}(q|p) \,:\, q \in \mathscr{P}_d(\mathcal{X}), \ q_N = p_{\text{prior}}\},$$

which is the first step of IPF. Hence subsequent steps can be obtained by maximizing ELBOs associated with the following maximum likelihood problems for any $n \in \mathbb{N}$

$$q^\star = \arg\max\{\mathbb{E}_{p_{\text{data}}}[\log q_0(X_0)] \,:\, q \in \mathscr{P}_d(\mathcal{X}), \ q_N = p_{\text{prior}}\},$$

$$p^\star = \arg\max\{\mathbb{E}_{p_{\text{prior}}}[\log p_N(X_N)] \,:\, p \in \mathscr{P}_d(\mathcal{X}), \ p_0 = p_{\text{data}}\}.$$

## S5 Alternative variational formulations

In this section, we draw links between IPF and score-matching techniques. We start by proving Proposition 3 in Section S5.1. We then present alternative variational formulations in Section S5.2.

### S5.1 Proof of Proposition 3

We only prove (12) since the proof (13) is similar. Let $n \in \mathbb{N}$ and $k \in \{0, \ldots, N-1\}$. For any $x_{k+1} \in \mathbb{R}^d$ we have

$$p_{k+1}^n(x_{k+1}) = (4\pi\gamma_{k+1})^{-d/2} \int_{\mathbb{R}^d} p^n(x_k) \exp[-\|F_k^n(x_k) - x_{k+1}\|^2/(4\gamma_{k+1})]\mathrm{d}x_k,$$

with $F_k^n(x_k) = x_k + \gamma_{k+1}f_k^n(x_k)$. Since $p_k^n > 0$ is bounded using the dominated convergence theorem we have for any $x_{k+1} \in \mathbb{R}^d$

$$\nabla \log p_{k+1}^n(x_{k+1}) = \int_{\mathbb{R}^d}(F_k^n(x_k) - x_{k+1})/(2\gamma_{k+1})p_{k|k+1}(x_k|x_{k+1})\mathrm{d}x_k.$$

Therefore we get that for any $x_{k+1} \in \mathbb{R}^d$

$$b_{k+1}^n(x_{k+1}) = \int_{\mathbb{R}^d}(F_k^n(x_k) - F_k^n(x_{k+1}))/\gamma_{k+1}p_{k|k+1}(x_k|x_{k+1})\mathrm{d}x_k.$$

This is equivalent to

$$B_{k+1}^n(x_{k+1}) = \mathbb{E}[X_{k+1} + F_k^n(X_k) - F_k^n(X_{k+1})|X_{k+1} = x_{k+1}],$$

with $(X_k, X_{k+1}) \sim p_{k,k+1}(x_k, x_{k+1})$. Hence, we get that

$$B_{k+1}^n = \arg\min_{\mathrm{B} \in \mathrm{L}^2(\mathbb{R}^d, \mathbb{R}^d)} \mathbb{E}_{p_{k,k+1}^n}[\|\mathrm{B}(X_{k+1}) - (X_{k+1} + F_k^n(X_k) - F_k^n(X_{k+1}))\|^2],$$

which concludes the proof.

### S5.2 Variational formulas

In Proposition 3 and Section 3.3 we present a variational formula for $B_{k+1}^n$ and $F_k^{n+1}$ for any $n \in \mathbb{N}$ and $k \in \{0, \ldots, N-1\}$, where we recall that for any $x \in \mathbb{R}^d$ we have

$$B_{k+1}^n(x) = x + \gamma_{k+1}b_{k+1}^n(x), \qquad F_k^{n+1} = x + \gamma_{k+1}f_k^{n+1}(x),$$

where we have

$$b_{k+1}^n(x) = -f_k^n(x) + 2\nabla \log p_{k+1}^n(x), \qquad f_k^{n+1}(x) = -b_{k+1}^n(x) + 2\nabla \log q_k^n(x). \tag{S48}$$

In the rest of this section we assume that for any $n \in \mathbb{N}$, $k \in \{0, \ldots, N-1\}$ and $x \in \mathbb{R}^d$ we have

$$q_{k|k+1}^n(x_k|x_{k+1}) = (4\pi\gamma_{k+1})^{-d/2} \exp[-\|x_k - B_{k+1}^n(x_{k+1})\|^2/(4\gamma_{k+1})],$$

$$p_{k+1|k}^{n+1}(x_{k+1}|x_k) = (4\pi\gamma_{k+1})^{-d/2} \exp[-\|x_{k+1} - F_k^{n+1}(x_k)\|^2/(4\gamma_{k+1})].$$

We recall that in this case Proposition 3 ensures that for any $n \in \mathbb{N}$ and $k \in \{0, \ldots, N-1\}$

$$B_{k+1}^n = \arg\min_{\mathrm{B} \in \mathrm{L}^2(\mathbb{R}^d, \mathbb{R}^d)} \mathbb{E}_{p_{k,k+1}^n}[\|\mathrm{B}(X_{k+1}) - (X_{k+1} + F_k^n(X_k) - F_k^n(X_{k+1}))\|^2],$$

$$F_k^{n+1} = \arg\min_{\mathrm{F} \in \mathrm{L}^2(\mathbb{R}^d, \mathbb{R}^d)} \mathbb{E}_{q_{k,k+1}^n}[\|\mathrm{F}(X_k) - (X_k + B_{k+1}^n(X_{k+1}) - B_{k+1}^n(X_k))\|^2].$$

In the rest of this section we derive other variational formulas and discuss their practical limitations/advantages.

#### S5.2.1 Score-matching formula and sum of networks

First, using (S48) we have for any $n \in \mathbb{N}$, $k \in \{0, \ldots, N-1\}$ and $x \in \mathbb{R}^d$

$$b_{k+1}^n(x) = \alpha x + 2 \sum_{j=0}^n \nabla \log p_{k+1}^j(x) - 2 \sum_{j=0}^{n-1} \nabla \log q_k^j(x), \tag{S49}$$

$$f_k^n(x) = -\alpha x + 2 \sum_{j=0}^{n-1} \nabla \log q_k^j(x) - 2 \sum_{j=0}^{n-1} \nabla \log p_{k+1}^j(x). \tag{S50}$$

In the following proposition we derive a variational formula for $\nabla \log p_{k+1}^n$ and $\nabla \log q_k^n(x)$ for any $n \in \mathbb{N}$ and $k \in \{0, \ldots, N-1\}$.

**Proposition S22.** *For any $n \in \mathbb{N}$ and $k \in \{0, \ldots, N-1\}$ we have*

$$\nabla \log p_{k+1}^n = \arg \min_{u \in \mathrm{L}^2(\mathbb{R}^d, \mathbb{R}^d)} \mathbb{E}_{p_{k,k+1}^n}[\|u(X_{k+1}) - (F_k^n(X_k) - X_{k+1})/(2\gamma_{k+1})\|^2], \quad \text{(S51)}$$

$$\nabla \log q_k^n = \arg \min_{v \in \mathrm{L}^2(\mathbb{R}^d, \mathbb{R}^d)} \mathbb{E}_{q_{k,k+1}^n}[\|v(X_k) - (B_{k+1}^n(X_{k+1}) - X_k)/(2\gamma_{k+1})\|^2]. \quad \text{(S52)}$$

*Proof.* The proof is similar to the one of Proposition 3 but is provided for completeness. We only prove (S53) since the proof (S54) is similar. Let $n \in \mathbb{N}$ and $k \in \{0, \ldots, N-1\}$. For any $x_{k+1} \in \mathbb{R}^d$ we have

$$p_{k+1}^n(x_{k+1}) = (4\pi\gamma_{k+1})^{-d/2} \int_{\mathbb{R}^d} p^n(x_k) \exp[-\|F_k^n(x_k) - x_{k+1}\|^2/(4\gamma_{k+1})]\mathrm{d}x_k,$$

with $F_k^n(x_k) = x_k + \gamma_{k+1}f_k^n(x_k)$. Since $p_k^n > 0$ is bounded using the dominated convergence theorem we have for any $x_{k+1} \in \mathbb{R}^d$

$$\nabla \log p_{k+1}^n(x_{k+1}) = \int_{\mathbb{R}^d}(F_k^n(x_k) - x_{k+1})/(2\gamma_{k+1})p_{k|k+1}(x_k|x_{k+1})\mathrm{d}x_k.$$

This is equivalent to

$$\nabla \log p_{k+1}^n(x_{k+1}) = \mathbb{E}[(F_k^n(X_k) - X_{k+1})/(2\gamma_{k+1})|X_{k+1} = x_{k+1}],$$

with $(X_k, X_{k+1}) \sim p_{k,k+1}(x_k, x_{k+1})$. Hence, we get that

$$\nabla \log p_{k+1}^n = \arg \min_{u \in \mathrm{L}^2(\mathbb{R}^d, \mathbb{R}^d)} \mathbb{E}_{p_{k,k+1}^n}[\|u(X_{k+1}) - (F_k^n(X_k) - X_{k+1})/(2\gamma_{k+1})\|^2],$$

which concludes the proof. $\qquad \square$

Note that (S53) and (S54) can be simplified upon remarking that for any $n \in \mathbb{N}$ and $k \in \{0, \ldots, N-1\}$

$$X_{k+1}^n = F_k^n(X_k^n) + \sqrt{2\gamma_{k+1}}Z_{k+1}^n, \quad \tilde{X}_k^n = F_k^n(\tilde{X}_{k+1}^n) + \sqrt{2\gamma_{k+1}}\tilde{Z}_{k+1}^n,$$

with $\{X_k^n\}_{k=0}^N \sim p^n$, $\{\tilde{X}_k^n\}_{k=0}^N \sim q^n$ and $\{(Z_{k+1}^n, \tilde{Z}_{k+1}^n) : n \in \mathbb{N}, k \in \{0, \ldots, N-1\}\}$ a family of independent Gaussian random variables with zero mean an identity covariance matrix. Using this result we get that for any $n \in \mathbb{N}$ and $k \in \{0, \ldots, N-1\}$

$$\nabla \log p_{k+1}^n = \arg \min_{u \in \mathrm{L}^2(\mathbb{R}^d, \mathbb{R}^d)} \mathbb{E}_{p_{k,k+1}^n}[\|u(X_{k+1}) - Z_{k+1}^n/\sqrt{2\gamma_{k+1}}\|^2], \quad \text{(S53)}$$

$$\nabla \log q_k^n = \arg \min_{v \in \mathrm{L}^2(\mathbb{R}^d, \mathbb{R}^d)} \mathbb{E}_{q_{k,k+1}^n}[\|v(X_k) - \tilde{Z}_{k+1}^n/\sqrt{2\gamma_{k+1}}\|^2]. \quad \text{(S54)}$$

In practice, neural networks $u_{\alpha^n}(k, x) \approx \nabla \log p_k^n(x)$, and $v_{\beta^n}(k, x) \approx \nabla \log q_k^n(x)$ are used. Hence, we sample approximately from $q^n$ and $p^n$ for any $n \in \mathbb{N}$ using the following recursion:

$$\tilde{X}_k^n = \tilde{\tau}_{k+1}\tilde{X}_{k+1}^n + 2\gamma_{k+1}\{\sum_{j=0}^n u_{\alpha^j}(k+1, \tilde{X}_{k+1}^n) - \sum_{j=0}^{n-1} v_{\beta^j}(k, \tilde{X}_{k+1}^n)\} + \sqrt{2\gamma_{k+1}}\tilde{Z}_{k+1}^n,$$

$$X_{k+1}^n = \tau_{k+1}X_k^n + 2\gamma_{k+1}\{\sum_{j=0}^n u_{\alpha^j}(k+1, X_k^n) - \sum_{j=0}^n v_{\beta^j}(k, X_k^n)\} + \sqrt{2\gamma_{k+1}}Z_{k+1}^n, \quad \text{(S55)}$$

where $\tilde{\tau}_{k+1} = 1 + \alpha\gamma_{k+1}$, $\tau_{k+1} = 1 - \alpha\gamma_{k+1}$ and $X_0^n \sim p_{\mathrm{data}}$, $\tilde{X}_N^n \sim p_{\mathrm{prior}}$.

### S5.2.2 Drift-matching formula

In Proposition 3 we have given a variational formula for $B_{k+1}^n$ and $F_k^{n+1}$ for any $n \in \mathbb{N}$ and $k \in \{0, \ldots, N-1\}$. In Proposition S22 we have given a variational formula for $\nabla \log p_{k+1}^n$ and $\nabla \log q_k^n$ for any $n \in \mathbb{N}$ and $k \in \{0, \ldots, N-1\}$. In the following proposition we give a variational formula for the drifts $b_{k+1}^n$ and $f_k^{n+1}$.

**Proposition S23.** *For any $n \in \mathbb{N}$ and $k \in \{0, \ldots, N-1\}$ we have*

$$b_{k+1}^n = \arg \min_{b \in \mathrm{L}^2(\mathbb{R}^d, \mathbb{R}^d)} \mathbb{E}_{p_{k,k+1}^n}[\|b(X_{k+1}) - (F_k^n(X_k) - F_k^n(X_{k+1}))/\gamma_{k+1}\|^2] \quad \text{(S56)}$$

$$f_k^{n+1} = \arg \min_{f \in \mathrm{L}^2(\mathbb{R}^d, \mathbb{R}^d)} \mathbb{E}_{q_{k,k+1}^n}[\|f(X_k) - (B_{k+1}^n(X_{k+1}) - B_{k+1}^n(X_k))/\gamma_{k+1}\|^2] \quad \text{(S57)}$$

*Proof.* The proof is similar to the one of Proposition 3 but is provided for completeness. We only prove (S56) since the proof (S57) is similar. Let $n \in \mathbb{N}$ and $k \in \{0, \ldots, N-1\}$. For any $x_{k+1} \in \mathbb{R}^d$ we have

$$p_{k+1}^n(x_{k+1}) = (4\pi\gamma_{k+1})^{-d/2} \int_{\mathbb{R}^d} p^n(x_k) \exp[-\|F_k^n(x_k) - x_{k+1}\|^2/(4\gamma_{k+1})]\mathrm{d}x_k,$$

with $F_k^n(x_k) = x_k + \gamma_{k+1} f_k^n(x_k)$. Since $p_k^n > 0$ is bounded using the dominated convergence theorem we have for any $x_{k+1} \in \mathbb{R}^d$

$$\nabla \log p_{k+1}^n(x_{k+1}) = \int_{\mathbb{R}^d} (F_k^n(x_k) - x_{k+1})/(2\gamma_{k+1}) p_{k|k+1}(x_k|x_{k+1}) \mathrm{d}x_k.$$

Therefore we get that for any $x_{k+1} \in \mathbb{R}^d$

$$b_{k+1}^n(x_{k+1}) = \int_{\mathbb{R}^d} (F_k^n(x_k) - F_k^n(x_{k+1}))/\gamma_{k+1} p_{k|k+1}(x_k|x_{k+1}) \mathrm{d}x_k.$$

This is equivalent to

$$b_{k+1}^n(x_{k+1}) = \mathbb{E}[(F_k^n(X_k) - F_k^n(X_{k+1}))/\gamma_{k+1}|X_{k+1} = x_{k+1}],$$

with $(X_k, X_{k+1}) \sim p(x_k, x_{k+1})$. Hence, we get that

$$b_{k+1}^n = \arg\min_{b \in \mathrm{L}^2(\mathbb{R}^d, \mathbb{R}^d)} \mathbb{E}_{p_{k,k+1}^n}[\|b(X_{k+1}) - (F_k^n(X_k) - F_k^n(X_{k+1}))/\gamma_{k+1}\|^2],$$

which concludes the proof. $\square$

In practice, neural networks $b_{\beta^n}(k, x) \approx b_k^n(x)$, and $f_{\alpha^n}(k, x) \approx f_k^n(x)$ are used. Hence, we sample approximately from $q^n$ and $p^n$ for any $n \in \mathbb{N}$ using the following recursion:

$$\tilde{X}_k^n = \tilde{X}_{k+1}^n + \gamma_{k+1} b_{\beta^n}(k+1, \tilde{X}_{k+1}^n) + \sqrt{2\gamma_{k+1}} \tilde{Z}_{k+1}^n,$$
$$X_{k+1}^n = X_k^n + \gamma_{k+1} f_{\alpha^n}(k, X_k^n) + \sqrt{2\gamma_{k+1}} Z_{k+1}^n,$$

with $X_0^n \sim p_{\text{data}}$, $\tilde{X}_N^n \sim p_{\text{prior}}$.

### S5.2.3 Discussion

We identify three variational formulas associated with Proposition 3, Proposition S22 and Proposition S23. In practice we discard the approach of Section S5.2.1 because it requires storing $2n$ neural networks to sample from $p^n$, see (S55). Hence the algorithm requires more memory as $n$ increases and the sampling procedure requires $\mathcal{O}(nN)$ passes through a neural network. The approaches described in Proposition 3 and Proposition S23 yield sampling procedures which only require $\mathcal{O}(N)$ passes through a neural network and have fixed memory cost for any $n \in \mathbb{N}$. In practice we observed that the approach of Proposition 3 yields better results. We conjecture that this favorable behavior is mainly due to the architecture of the neural networks used to approximate $B_{k+1}^n$ and $F_k^{n+1}$ which have residual connections and therefore are better suited at representing functions of the $x \mapsto x + \Phi(x)$ where $\Phi$ is a perturbation.

## S6 Theoretical study of Schrödinger bridges and the IPF

In this section, we explore some of the theoretical properties of Schrödinger bridges and the IPF procedure. Proposition 4 and Proposition 5 are proved in Section S6.1 and Section S6.2 respectively.

### S6.1 Proof of Proposition 4

In this section, we prove Proposition 4. First we gather novel monotonicity results for the IPF in Proposition S25, see Section S6.1.1. Then we prove our quantitative convergence bounds in Theorem S30, see Section S6.1.2.

### S6.1.1 Monotonicity results

We consider the static IPF recursion: $\pi^0 = \mu \in \mathscr{P}_2$ and

$$\pi^{2n+1} = \arg\min\left\{\mathrm{KL}(\pi|\pi^{2n}) \ : \ \pi \in \mathscr{P}_2, \ \pi_1 = \nu_1\right\},$$
$$\pi^{2n+2} = \arg\min\left\{\mathrm{KL}(\pi|\pi^{2n+1}) \ : \ \pi \in \mathscr{P}_2, \ \pi_0 = \nu_0\right\},$$

where $\nu_0, \nu_1 \in \mathscr{P}(\mathbb{R}^d)$. We also consider the following assumption.

**B1.** *$\mu$ is absolutely continuous w.r.t. $\mu_0 \otimes \mu_1$ and $\mathrm{KL}(\nu_0 \otimes \nu_1|\mu) < +\infty$. In addition, $\nu_i$ and $\mu_i$ are equivalent for $i \in \{0, 1\}$.*

First we draw links between **A**1 and **B**1.

**Proposition S24.** **A**1 *implies* **B**1 *with* $\mu = p_{0,N}$.

*Proof.* Since $p_N > 0$ we get that $p_N$ and $p_{\text{prior}}$ are equivalent. Hence $\mu_1$ and $\nu_1$ are equivalent and $\mu_0 = \nu_0$. Let us show that $\mu$ is absolutely continuous w.r.t. $\mu_0 \otimes \mu_1$, *i.e.* that $p_{0,N}$ is absolutely continuous w.r.t. $p_{\text{data}} \otimes p_N$. Since $p_N > 0$ we get that $p_{0,N}$ is absolutely continuous w.r.t. $p_{\text{data}} \otimes p_N$ with density $p_{N|0}/p_N$. Finally we have

$$\int_{(\mathbb{R}^d)^2} \log(p_{\text{data}}(x_0)p_{\text{prior}}(x_N)/(p_{\text{data}}(x_0)p_{N|0}(x_N|x_0)))p_{\text{data}}(x_0)p_{\text{prior}}(x_N)\mathrm{d}x_0\mathrm{d}x_N$$

$$= \int_{(\mathbb{R}^d)^2} \log(p_{\text{prior}}(x_N)/p_{N|0}(x_N|x_0))p_{\text{data}}(x_0)p_{\text{prior}}(x_N)\mathrm{d}x_0\mathrm{d}x_N$$

$$\leq |\mathrm{H}(p_{\text{prior}})| + \int_{\mathbb{R}^d} |\log p_{N|0}(x_N|x_0)|p_{\text{data}}(x_0)p_{\text{prior}}(x_N)\mathrm{d}x_0\mathrm{d}x_N < +\infty$$

which concludes the proof. $\qquad\square$

In this section we prove the following proposition.

**Proposition S25.** *Assume* **B**1. *Then, the IPF sequence is well-defined and for any* $n \in \mathbb{N}$ *with* $n \geq 1$ *we have*

$$\mathrm{KL}(\pi^{n+1}|\pi^n) \leq \mathrm{KL}(\pi^{n-1}|\pi^n), \qquad \mathrm{KL}(\pi^n|\pi^{n+1}) \leq \mathrm{KL}(\pi^n|\pi^{n-1}). \qquad \text{(S58)}$$

*In addition, the following results hold:*

*(a)* $(\|\pi^{n+1} - \pi^n\|_{\mathrm{TV}})_{n \in \mathbb{N}}$ *and* $(\mathrm{J}(\pi^{n+1}, \pi^n))_{n \in \mathbb{N}}$ *are non-increasing.*

*(b)* $(\mathrm{KL}(\pi^{2n+1}|\pi^{2n}))_{n \in \mathbb{N}}$ *and* $(\mathrm{KL}(\pi^{2n+2}|\pi^{2n+1}))_{n \in \mathbb{N}}$ *are non-increasing.*

*(c)* $(\mathrm{KL}(\pi_1^{2n+1}|\nu_1))_{n \in \mathbb{N}}$ *and* $(\mathrm{KL}(\pi_0^{2n}|\nu_0))_{n \in \mathbb{N}}$ *are non-increasing.*

*(d)* $(\|\pi_1^{2n+1} - \nu_1\|_{\mathrm{TV}})_{n \in \mathbb{N}}$ *and* $(\|\pi_0^{2n} - \nu_0\|_{\mathrm{TV}})_{n \in \mathbb{N}}$ *are non-increasing.*

First, we show that under **B**1, the IPF sequence is well-defined and is associated with a sequence of potentials.

**Proposition S26.** *Assume* **B**1. *Then, the IPF sequence is well-defined and there exist* $(a_n)_{n \in \mathbb{N}}$ *and* $(b_n)_{n \in \mathbb{N}}$ *such that for any* $n \in \mathbb{N}$, $a_n, b_n : \mathbb{R}^d \to (0, +\infty)$ *and for any* $x, y \in \mathbb{R}^d$

$$(\mathrm{d}\pi^{2n+1}/\mathrm{d}(\mu_0 \otimes \mu_1))(x, y) = a_n(x)h(x, y)b_n(y) \qquad \text{(S59)}$$

$$(\mathrm{d}\pi^{2n+2}/\mathrm{d}(\mu_0 \otimes \mu_1))(x, y) = a_{n+1}(x)h(x, y)b_n(y),$$

*and*

$$v_0(x) = a_{n+1}(x)\int_{\mathbb{R}^d} h(x, y)b_n(y)\mathrm{d}\mu_1(y), \quad v_1(y) = b_n(y)\int_{\mathbb{R}^d} h(x, y)a_n(x)\mathrm{d}\mu_0(x), \qquad \text{(S60)}$$

*where* $v_i = \mathrm{d}\nu_i/\mathrm{d}\mu_i$ *for* $i \in \{0, 1\}$.

*Proof.* First, we show that the IPF sequence is well-defined. Note that $\pi^1$ is well-defined since $\mathrm{KL}(\nu_0 \otimes \nu_1|\mu) < +\infty$. Assume that $\{\pi^\ell\}_{\ell=1}^n$ is well-defined. Using (Csiszár, 1975, Theorem 2.2) we have

$$\mathrm{KL}(\nu_0 \otimes \nu_1|\mu) = \mathrm{KL}(\nu_0 \otimes \nu_1|\pi^n) + \sum_{\ell=0}^{n-1} \mathrm{KL}(\pi^{\ell+1}|\pi^\ell).$$

In particular, $\mathrm{KL}(\nu_0 \otimes \nu_1|\pi^n) < +\infty$ and $\pi^{n+1}$ is well-defined. We conclude by recursion.

Using (Csiszár, 1975, Theorem 3.1) and **B**1, there exists $(\tilde{b}_n)_{n \in \mathbb{N}}$ such that for any $n \in \mathbb{N}$, $\tilde{b}_n : \mathbb{R}^d \to [0, +\infty)$ and for any $x, y \in \mathsf{A}_n$, $(\mathrm{d}\pi^{2n+1}/\mathrm{d}\pi^{2n})(x, y) = \tilde{b}_n(y)$ with $\mathsf{A}_n \in \mathcal{B}(\mathbb{R}^d)$, $\tilde{\pi}(\mathsf{A}_n) = 0$ for any $\tilde{\pi}$ such that $\tilde{\pi}_1 = \nu_1$ and $\mathrm{KL}(\tilde{\pi}|\pi^{2n}) < +\infty$. In particular we have $(\nu_0 \otimes \nu_1)(\mathsf{A}_n) = 0$. Since $\nu_i$ is equivalent to $\mu_i$ for any $i \in \{0, 1\}$ we have $(\mu_0 \otimes \mu_1)(\mathsf{A}_n) = 0$. Similarly, there exists $(\tilde{a}_n)_{n \in \mathbb{N}}$ such that for any $n \in \mathbb{N}$, $\tilde{a}_n : \mathbb{R}^d \to [0, +\infty)$ and for any $x, y \in \mathsf{B}_n$, $(\mathrm{d}\pi^{2n+2}/\mathrm{d}\pi^{2n+1})(x, y) = \tilde{a}_{n+1}(x)$ with $\mathsf{B}_n \in \mathcal{B}(\mathbb{R}^d)$ and $(\mu_0 \otimes \mu_1)(\mathsf{B}_n) = 0$. As a result, there exist $(a_n)_{n \in \mathbb{N}}$ and $(b_n)_{n \in \mathbb{N}}$ with $a_n : \mathbb{R}^d \to [0, +\infty)$ and $b_n : \mathbb{R}^d \to [0, +\infty)$ such that for any $n \in \mathbb{N}$ and $x, y \in \mathbb{R}^d$

$$(\mathrm{d}\pi^{2n+1}/\mathrm{d}(\mu_0 \otimes \mu_1))(x, y) = a_n(x)h(x, y)b_n(y)$$

$$(\mathrm{d}\pi^{2n+2}/\mathrm{d}(\mu_0 \otimes \mu_1))(x, y) = a_{n+1}(x)h(x, y)b_n(y),$$

where $h = \mathrm{d}\mu/\mathrm{d}(\mu_0 \otimes \mu_1)$ and $a_0 = 1$. In addition, setting $b_{-1} = 1$, we have for any $x, y \in \mathbb{R}^d$,

$$(\mathrm{d}\pi^0/\mathrm{d}(\mu_0 \otimes \mu_1))(x, y) = a_0(x)h(x,y)b_{-1}(y).$$

Using that $\nu_i$ is absolutely continuous w.r.t. $\mu_i$ for $i \in \{0, 1\}$ with density $v_i : \mathbb{R}^d \to (0, +\infty)$ we get that for any $x, y \in \mathbb{R}^d$ and $n \in \mathbb{N}$

$$v_0(x) = a_{n+1}(x) \int_{\mathbb{R}^d} h(x,y)b_n(y)\mathrm{d}\mu_1(y), \quad v_1(y) = b_n(y) \int_{\mathbb{R}^d} h(x,y)a_n(x)\mathrm{d}\mu_0(x).$$

Since $v_0, v_1 > 0$ for any $n \in \mathbb{N}$, $a_n, b_n > 0$. $\qquad \square$

Note that the system of equations (S60) corresponds to iteratively solving the Schrödinger system, see Léonard (2014b) for a survey. In addition, (S60) has connections with Fortet's mapping (Léonard, 2019; Fortet, 1940).

In the rest of the section we detail the proof of Proposition 4. We start by deriving identities between the marginals of the IPF and its joint distribution both w.r.t. the Kullback-Leibler divergence and the total variation norm in Lemma S27. Second, we establish that $(\|\pi^{n+1} - \pi^n\|_{\mathrm{TV}})_{n \in \mathbb{N}}$ is non-increasing in Lemma S28. Then, we prove (S58) in Lemma S29. We conclude with the proof of Proposition S25.

**Lemma S27.** *Assume* **B**1. *Then, for any $n \in \mathbb{N}$ we have*

$$\|\pi^{2n+1} - \pi^{2n}\|_{\mathrm{TV}} = \|\pi_1^{2n} - \nu_1\|_{\mathrm{TV}}, \qquad \|\pi^{2n+2} - \pi^{2n+1}\|_{\mathrm{TV}} = \|\pi_0^{2n+1} - \nu_0\|_{\mathrm{TV}}. \quad \text{(S61)}$$

*In addition, we have*

$$\mathrm{KL}(\pi^{2n}|\pi^{2n+1}) = \mathrm{KL}(\pi_1^{2n}|\nu_1), \qquad \mathrm{KL}(\pi^{2n+1}|\pi^{2n+2}) = \mathrm{KL}(\pi_0^{2n+1}|\nu_0). \quad \text{(S62)}$$

*Proof.* We divide the proof into two parts. First, we prove (S61). Second, we show that (S62) holds.

(a) We only show that for any $n \in \mathbb{N}$ we have $\|\pi^{2n+1} - \pi^{2n}\|_{\mathrm{TV}} = \|\pi_1^{2n} - \nu_1\|_{\mathrm{TV}}$. The proof that for any $n \in \mathbb{N}$, $\|\pi^{2n+2} - \pi^{2n+1}\|_{\mathrm{TV}} = \|\pi_0^{2n+1} - \nu_0\|_{\mathrm{TV}}$ is similar. Let $n \in \mathbb{N}$. Using (S59) and (S60) we have

$$\|\pi^{2n+1} - \pi^{2n}\|_{\mathrm{TV}} = \int_{(\mathbb{R}^d)^2} |b_n(y) - b_{n-1}(y)| \, a_n(x)h(x,y)\mathrm{d}\mu_0(x)\mathrm{d}\mu_1(y) \qquad \text{(S63)}$$
$$= \int_{\mathbb{R}^d} |1 - b_{n-1}(x)/b_n(x)| \, \mathrm{d}\nu_1(y).$$

In addition, we have that for any $\mathsf{A} \in \mathcal{B}(\mathbb{R}^d)$

$$\pi_1^{2n}(\mathsf{A}) = \int_{\mathbb{R}^d \times \mathsf{A}} a_n(x)b_{n-1}(y)h(x,y)\mathrm{d}\mu_0(x)\mathrm{d}\mu_1(y) = \int_{\mathsf{A}} (b_{n-1}/b_n)(y)\mathrm{d}\nu_1(y).$$

We get that for any $y \in \mathbb{R}^d$, $(\mathrm{d}\pi_1^{2n}/\mathrm{d}\nu_1)(y) = (b_{n-1}/b_n)(y)$. Hence, using (S63) we get that

$$\|\pi_1^{2n} - \nu_1\|_{\mathrm{TV}} = \int_{\mathbb{R}^d} |1 - a_n(x)/a_{n+1}(x)| \, \mathrm{d}\nu_0(x) = \|\pi^{2n+1} - \pi^{2n}\|_{\mathrm{TV}}.$$

(b) We only show that for any $n \in \mathbb{N}$ we have $\mathrm{KL}(\pi^{2n}|\pi^{2n+1}) = \mathrm{KL}(\pi_1^{2n}|\nu_1)$. The proof that for any $n \in \mathbb{N}$, $\mathrm{KL}(\pi^{2n+1}|\pi^{2n+2}) = \mathrm{KL}(\pi_0^{2n+1}|\nu_0)$ is similar. Let $n \in \mathbb{N}$. Using that for any $x, y \in \mathbb{R}^d$, $(\mathrm{d}\pi_1^{2n}/\mathrm{d}\nu_1)(y) = b_{n-1}(y)/b_n(y)$ and that $(\mathrm{d}\pi^{2n+1}/\mathrm{d}\pi^{2n})(x, y) = b_n(y)/b_{n-1}(y)$ we have

$$\mathrm{KL}(\pi^{2n}|\pi^{2n+1}) = -\int_{\mathbb{R}^d} \log(b_n(y)/b_{n-1}(y))\mathrm{d}\pi_1^{2n}(y) = \mathrm{KL}(\pi_1^{2n}|\nu_1).$$

This concludes the proof.

$\qquad \square$

**Lemma S28.** *Assume* **B**1. *Then $(\|\pi^{n+1} - \pi^n\|_{\mathrm{TV}})_{n \in \mathbb{N}}$ is non-increasing.*

*Proof.* We only prove that for any $n \in \mathbb{N}$ with $n \geq 1$, $\|\pi^{2n+1} - \pi^{2n}\|_{\mathrm{TV}} \leq \|\pi^{2n} - \pi^{2n-1}\|_{\mathrm{TV}}$. The proof that for any $n \in \mathbb{N}$, $\|\pi^{2n+2} - \pi^{2n+1}\|_{\mathrm{TV}} \leq \|\pi^{2n+1} - \pi^{2n}\|_{\mathrm{TV}}$ is similar. Let $n \in \mathbb{N}$ with $n \geq 1$. Similarly to the proof of Lemma S27 we have that

$$\|\pi^{2n+1} - \pi^{2n}\|_{\mathrm{TV}} = \int_{\mathbb{R}^d} |1 - b_{n-1}(y)/b_n(y)| \, \mathrm{d}\nu_1(y) = \int_{\mathbb{R}^d} \left|b_n^{-1}(y) - b_{n-1}^{-1}(y)\right| b_{n-1}(y)\mathrm{d}\nu_1(y).$$
$$\text{(S64)}$$

In addition, we have that for any $y \in \mathbb{R}^d$

$$\left|b_{n-1}^{-1}(y) - b_n^{-1}(y)\right| \leq v_1^{-1}(y) \int_{\mathbb{R}^d} h(x, y) \left|a_{n-1}(x) - a_n(x)\right| \mathrm{d}\mu_0(x).$$

Combining this result and (S64) we get that

$$\begin{aligned}
\|\pi^{2n+1} - \pi^{2n}\|_{\mathrm{TV}} &\leq \int_{\mathbb{R}^d} \left|b_{n-1}^{-1}(y) - b_n^{-1}(y)\right| b_{n-1}(y) \mathrm{d}\nu_1(y) \\
&\leq \int_{(\mathbb{R}^d)^2} |a_n(x) - a_{n-1}(x)| \, h(x, y) b_{n-1}(y) \mathrm{d}\mu_0(x) \mathrm{d}\mu_1(y) \\
&\leq \int_{\mathbb{R}^d} |1 - a_{n-1}(x)/a_n(x)| \, \mathrm{d}\nu_0(x) \leq \|\pi^{2n} - \pi^{2n-1}\|_{\mathrm{TV}},
\end{aligned}$$

which concludes the proof. $\qquad\square$

**Lemma S29.** *Assume* **B**1. *Then for any* $n \in \mathbb{N}$ *with* $n \geq 1$ *we have*

$$\mathrm{KL}(\pi^{n+1}|\pi^n) \leq \mathrm{KL}(\pi^{n-1}|\pi^n), \qquad \mathrm{KL}(\pi^n|\pi^{n+1}) \leq \mathrm{KL}(\pi^n|\pi^{n-1}).$$

*Proof.* Using Lemma S27 and the data processing theorem (Ambrosio et al., 2008, Lemma 9.4.5) we get that for any $n \in \mathbb{N}$

$$\mathrm{KL}(\pi^{2n}|\pi^{2n+1}) = \mathrm{KL}(\pi_1^{2n}|\nu_1) \leq \mathrm{KL}(\pi^{2n}|\pi^{2n+1}).$$

Similarly, we get that for any $n \in \mathbb{N}$, $\mathrm{KL}(\pi^{2n+1}|\pi^{2n+2}) \leq \mathrm{KL}(\pi^{2n+1}|\pi^{2n})$. Hence, we get that for any $n \in \mathbb{N}$, $\mathrm{KL}(\pi^n|\pi^{n+1}) \leq \mathrm{KL}(\pi^n|\pi^{n-1})$.

In addition, using that for any $n \in \mathbb{N}$ with $n \geq 1$ and $x, y \in \mathbb{R}^d$, we have that $\pi_1^{2n+1} = \nu_1$ and $(\mathrm{d}\pi^{2n+1}/\mathrm{d}\pi^{2n})(x, y) = b_n(y)/b_{n-1}(y)$ we get for any $n \in \mathbb{N}$ with $n \geq 1$

$$\mathrm{KL}(\pi^{2n+1}|\pi^{2n}) = -\int_{\mathbb{R}^d} \log(b_{n-1}(y)/b_n(y)) \mathrm{d}\nu_1(y). \tag{S65}$$

Using Jensen's inequality we have for any $n \in \mathbb{N}$

$$\begin{aligned}
-\log(b_{n-1}(y)/b_n(y)) &\leq -\log\left(\int_{\mathbb{R}^d} h(x, y) a_n(x) \mathrm{d}\mu_0(x) \middle/ \int_{\mathbb{R}^d} h(x, y) a_{n-1}(x) \mathrm{d}\mu_0(x)\right) \\
&\leq -\log\left(\int_{\mathbb{R}^d} (a_n(x)/a_{n-1}(x)) h(x, y) a_{n-1}(x) \mathrm{d}\mu_0(y) \middle/ \int_{\mathbb{R}^d} h(x, y) a_{n-1}(x) \mathrm{d}\mu_0(x)\right) \\
&\leq -\int_{\mathbb{R}^d} \log(a_n(x)/a_{n-1}(x)) b_{n-1}(y) h(x, y) a_{n-1}(x)/v_1(y) \mathrm{d}\mu_0(x).
\end{aligned}$$

Combining this result, (S65), Fubini's theorem and that for any $n \in \mathbb{N}$ with $n \geq 1$ and $x \in \mathbb{R}^d$, $(\mathrm{d}\pi_0^{2n-1}/\mathrm{d}\nu_0)(x) = a_{n-1}(x)/a_n(x)$ we get that for any $n \in \mathbb{N}$ with $n \geq 1$

$$\begin{aligned}
\mathrm{KL}(\pi^{2n+1}|\pi^{2n}) &\leq \int_{(\mathbb{R}^d)^2} \log(a_{n-1}(x)/a_n(x)) a_{n-1}(x) h(x, y) b_{n-1}(y) \mathrm{d}\mu_1(y) \mathrm{d}\mu_0(x) \\
&\leq \int_{(\mathbb{R}^d)^2} \log(a_{n-1}(x)/a_n(x))(a_{n-1}(x)/a_n(x)) \mathrm{d}\nu_0(x) \leq \mathrm{KL}(\pi_0^{2n-1}|\nu_0).
\end{aligned}$$

Using Lemma S27 (or the data processing theorem) we get that for any $n \in \mathbb{N}$ with $n \geq 1$, $\mathrm{KL}(\pi^{2n+1}|\pi^{2n}) \leq \mathrm{KL}(\pi^{2n-1}|\pi^{2n})$. Similarly, we get that for any $n \in \mathbb{N}$, $\mathrm{KL}(\pi^{2n+2}|\pi^{2n+1}) \leq \mathrm{KL}(\pi^{2n}|\pi^{2n+1})$, which concludes the proof. $\qquad\square$

We now turn to the proof of Proposition S25

*Proof.* First, (S58) is a direct consequence of Lemma S29. Using Lemma S28 we get that $(\|\pi^{n+1} - \pi^n\|_{\mathrm{TV}})_{n \in \mathbb{N}}$ is non-increasing. Since for any $\eta_0, \eta_1 \in \mathscr{P}(\mathbb{R}^d)$ we have $\mathrm{J}(\eta_0, \eta_1) = (1/2)\{\mathrm{KL}(\eta_0|\eta_1) + \mathrm{KL}(\eta_0|\eta_1)\}$ and using (S58), we get that $(\mathrm{J}(\pi_{n+1}, \pi_n))_{n \in \mathbb{N}}$ is non-increasing which proves Proposition S25-(a). Proposition S25-(b) is a straightforward consequence of (S58). Proposition S25-(c) is a consequence of Lemma S27 and Proposition S25-(a). Finally, Proposition S25-(c) is a consequence of Lemma S27 and (S58). $\qquad\square$

Note that we also have that for any $n \in \mathbb{N}$, $(\mathrm{KL}(\pi^{2n}|\pi^{2n+1}))_{n \in \mathbb{N}}$ and $(\mathrm{KL}(\pi^{2n+1}|\pi^{2n+2}))_{n \in \mathbb{N}}$ are non-increasing.

### S6.1.2 Quantitative convergence bounds

In this section we prove the following theorem.

**Theorem S30.** *Assume* **B**1. *Then, the IPF sequence* $(\pi^n)_{n\in\mathbb{N}}$ *is well-defined. Then, the following hold:*

*(a)* $\lim_{n\to+\infty} n^{1/2}\left\{\|\pi_0^n - \nu_0\|_{\mathrm{TV}} + \|\pi_1^n - \nu_1\|_{\mathrm{TV}}\right\} = 0.$

*(b)* $\lim_{n\to+\infty} n\left\{\mathrm{KL}(\pi_0^n|\nu_0) + \mathrm{KL}(\pi_1^n|\nu_1)\right\} = 0.$

We begin with Lemma S31 which is an adaption of (Ruschendorf et al., 1995, Proposition 2.1). Then we state and prove Lemma S32 which is a classical lemma from real analysis. Combining these two lemmas and the monotonicity results from Proposition S25 conclude the proof.

**Lemma S31.** *Assume* **B**1. *Then,* $(\pi^n)_{n\in\mathbb{N}}$ *is well-defined and we have* $\sum_{n\in\mathbb{N}} \mathrm{KL}(\pi^{n+1}|\pi^n) < +\infty.$

*Proof.* The sequence is well-defined using Proposition S26. In addition, using (Csiszár, 1975, Theorem 2.2) we have for any $n \in \mathbb{N}$

$$\mathrm{KL}(\pi^\star|\pi^0) = \mathrm{KL}(\pi^\star|\pi^n) + \sum_{k=0}^{n-1} \mathrm{KL}(\pi^{k+1}|\pi^k),$$

which concludes the proof. $\qquad\square$

**Lemma S32.** *Let* $(c_n)_{n\in\mathbb{N}} \in [0, +\infty)^{\mathbb{N}}$ *a non-increasing sequence such that* $\sum_{n\in\mathbb{N}} c_n < +\infty$. *Then* $\lim_{n\to+\infty} c_n n = 0.$

*Proof.* Let $\varepsilon > 0$ and $n_0 \in \mathbb{N}$ such that for any $n \geq n_0$, $\sum_{k=n}^{+\infty} c_k \leq \varepsilon$. Let $n \in \mathbb{N}$ with $n \geq 2n_0$. Note that $n - n_0 \geq n/2 \geq n_0$. Therefore we have $\varepsilon \geq (n - n_0)c_n \geq (n/2)c_n$. Hence, for any $n \in \mathbb{N}$ with $n \geq 2n_0$, $c_n n \leq 2\varepsilon$, which concludes the proof. $\qquad\square$

We now conclude with the proof of Theorem S30.

*Proof.* Since $(\mathrm{KL}(\pi^{2n+1}|\pi^{2n}))_{n\in\mathbb{N}}$ and $(\mathrm{KL}(\pi^{2n+2}|\pi^{2n+1}))_{n\in\mathbb{N}}$ are non-increasing by Proposition S25, using Lemma S32, we get that

$$\lim_{n\to+\infty} n\left\{\mathrm{KL}(\pi_0^n|\nu_0) + \mathrm{KL}(\pi_1^n|\nu_1)\right\} = 0.$$

We conclude upon using Pinsker's inequality (Bakry et al., 2014, Equation 5.2.2). $\qquad\square$

### S6.2 Proof of Proposition 5

Similarly to Section S6.1, we consider the static IPF recursion: $\pi^0 = \mu \in \mathscr{P}_2$ and

$$\pi^{2n+1} = \arg\min\left\{\mathrm{KL}(\pi|\pi^{2n}) \,:\, \pi \in \mathscr{P}_2, \ \pi_1 = \nu_1\right\},$$
$$\pi^{2n+2} = \arg\min\left\{\mathrm{KL}(\pi|\pi^{2n+1}) \,:\, \pi \in \mathscr{P}_2, \ \pi_0 = \nu_0\right\},$$

where $\nu_0, \nu_1 \in \mathscr{P}(\mathbb{R}^d)$. We recall that in this context if the Schrödinger bridge $\pi^\star$ exists it is given by

$$\pi^\star = \arg\min\{\mathrm{KL}(\pi|\mu) \,:\, \pi \in \mathscr{P}_2, \ \pi_0 = \nu_0, \ \pi_1 = \nu_1\}.$$

In this section, we prove the following proposition which directly implies Proposition 5.

**Proposition S33.** *Assume* **B**1 *and denote* $h = \mathrm{d}\mu/(\mathrm{d}\mu_0 \otimes \mu_1)$. *Assume that* $h \in \mathrm{C}(\mathbb{R}^d \times \mathbb{R}^d, (0, +\infty])$ *and that there exist* $\Phi_0, \Phi_1 \in \mathrm{C}(\mathbb{R}^d, (0, +\infty))$ *such that for any* $x, y \in \mathbb{R}^d$

$h(x, y) \leq \Phi_0(x)\Phi_1(y)$, *and*

$\int_{\mathbb{R}^d \times \mathbb{R}^d} (|\log h(x_0, x_1)| + |\log \Phi_0(x_0)| + |\log \Phi_1(x_1)|)\mathrm{d}\mu_0(x_0)\mathrm{d}\mu_1(x_1) < +\infty.$ (S66)

*Then there exists a solution* $\pi^\star$ *to the Schrödinger bridge. Assume that the IPF sequence satisfies* $\lim_{n\to+\infty} \|\pi^n - \pi^\infty\|_{\mathrm{TV}} = 0$ *with* $\pi^\infty \in \mathscr{P}_2$. *If* $\mu$ *is absolutely continuous w.r.t.* $\pi^\infty$ *then* $\pi^\infty = \pi^\star$.

We begin with an adaptation of (Rüschendorf and Thomsen, 1993, Proposition 2).

**Proposition S34.** *Let $\mu \in \mathscr{P}_2$ and assume that $\mu$ is absolutely continuous w.r.t. $\mu_0 \otimes \mu_1$. Let $(a_n)_{n\in\mathbb{N}}$ and $(b_n)_{n\in\mathbb{N}}$ such that for any $n \in \mathbb{N}$, $a_n : \mathbb{R}^d \to (0, +\infty)$ and $b_n : \mathbb{R}^d \to (0, +\infty)$. Assume that there exists $\Phi : (\mathbb{R}^d)^2 \to [0, +\infty)$ and $\mathsf{A} \in \mathcal{B}(\mathbb{R}^d) \otimes \mathcal{B}(\mathbb{R}^d)$ with $\mu(\mathsf{A}) = 1$ such that for any $(x, y) \in \mathsf{A}$*

$$\lim_{n\to+\infty} a_n(x) b_n(y) = \Phi(x, y).$$

*Then, there exist $a : \mathbb{R}^d \to [0, +\infty)$, $b : \mathbb{R}^d \to [0, +\infty)$ and $\mathsf{B} \in \mathcal{B}(\mathbb{R}^d) \otimes \mathcal{B}(\mathbb{R}^d)$ with $\mu(\mathsf{B}) = 1$ such that for any $x, y \in \mathsf{B}$*

$$\Phi(x, y) = a(x) b(y), \qquad or \quad \Phi(x, y) = 0.$$

*Proof.* Let $\tilde{\mathsf{A}} = \{(x, y) \in (\mathbb{R}^d)^2 : \Phi(x, y) = 0\}$ and $\mathsf{A}_a = \tilde{\mathsf{A}} \cap \mathsf{A}$ and $\mathsf{A}_b = \tilde{\mathsf{A}}^c \cap \mathsf{A}$. If $\mathsf{A}_b = \emptyset$, we conclude the proof. Otherwise, let $(x_0, y_0) \in \mathsf{A}_b$. Let $\mathsf{C}_0, \mathsf{C}_1 \in \mathcal{B}(\mathbb{R}^d) \otimes \mathcal{B}(\mathbb{R}^d)$ be given by

$$\mathsf{C}_0^0 = \{x \in \mathbb{R}^d : \lim_{n\to+\infty} a_n^0(x) = a^0(x) \text{ exists and } a^0(x) > 0\}, \tag{S67}$$

$$\mathsf{C}_1^0 = \{y \in \mathbb{R}^d : \lim_{n\to+\infty} b_n^0(y) = b^0(y) \text{ exists and } b^0(y) > 0\},$$

where for any $n \in \mathbb{N}$ and $x, y \in \mathbb{R}^d$, $a_n^0(x) = a_n(x)/a_n(x_0)$ and $b_n^0(y) = b_n(y)a_n(x_0)$, which is well-defined since for any $n \in \mathbb{N}$, $a_n(x_0) > 0$. Note that $x_0 \in \mathsf{C}_0^0$ and that $y_0 \in \mathsf{C}_1^0$. If $\mathsf{A}_b \subset \mathsf{C}_0^0 \times \mathsf{C}_1^0$, we conclude the proof. Otherwise, let $(x_1, y_1) \in \mathsf{A}_b \cap (\mathsf{C}_0^0 \times \mathsf{C}_1^0)^c$ and define

$$\mathsf{C}_0^1 = \{x \in \mathbb{R}^d : \lim_{n\to+\infty} a_n^1(x) = a^1(x) \text{ exists and } a^1(x) > 0\},$$

$$\mathsf{C}_1^1 = \{y \in \mathbb{R}^d : \lim_{n\to+\infty} b_n^1(y) = b^1(y) \text{ exists and } b^1(y) > 0\},$$

where for any $n \in \mathbb{N}$ and $x, y \in \mathbb{R}^d$, $a_n^1(x) = a_n(x)/a_n(x_1)$ and $b_n^1(y) = b_n(y)a_n(x_1)$, which is well-defined since for any $n \in \mathbb{N}$, $a_n(x_1) > 0$. Note that $\mathsf{C}_0^0 \cap \mathsf{C}_0^1 = \emptyset$ and $\mathsf{C}_1^0 \cap \mathsf{C}_1^1 = \emptyset$. Indeed, if there exists $x \in \mathsf{C}_0^0 \cap \mathsf{C}_0^1$, then $a^0(x) = \lim_{n\to+\infty} a_n(x)/a_n(x_0) > 0$ and $a^1(x) = \lim_{n\to+\infty} a_n(x)/a_n(x_1) > 0$ exists. Therefore $\lim_{n\to+\infty} a_n(x_1)/a_n(x_0) > 0$ exists and $\lim_{n\to+\infty} b_n(y_1)a_n(x_0) > 0$ exists. Hence $(x_1, y_1) \in \mathsf{C}_0^0 \times \mathsf{C}_1^0$ which is absurd. Similarly, if there exists $y \in \mathsf{C}_1^0 \cap \mathsf{C}_1^1$ then $(x_1, y_1) \in \mathsf{C}_0^0 \times \mathsf{C}_1^0$ which is absurd. Hence, we consider $T : \mathsf{A}_b \to 2^{(\mathbb{R}^d)^2}$ such that for any $(x, y) \in \mathsf{A}_b$, $T(x, y) = \mathsf{C}_0^{(x,y)} \times \mathsf{C}_1^{(x,y)}$, where $\mathsf{C}_0^{(x,y)} \times \mathsf{C}_1^{(x,y)}$ is constructed as in (S67) replacing $(x_0, y_0)$ by $(x, y)$.

Consider a well order on $(\mathsf{A}_b, \leq)$, which is possible by the well-ordering principle (Enderton, 1977, p. 196). For any $(x, y) \in \mathbb{R}^d$, let $\mathsf{A}_b^{(x,y)} = \{(x', y') \in (\mathbb{R}^d)^2 : (x', y') < (x, y)\}$. Using the transfinite recursion theorem (Enderton, 1977, p. 175) there exists $f : \mathsf{A}_b \to \{0, 1\}$ such that for any $(x, y) \in \mathsf{A}_b$ if there exists $(x', y') \in (\mathbb{R}^d)^2$ such that $(x', y') < (x, y)$, $f(x', y') = 1$ and $(x, y) \in T(x', y')$ then $f(x, y) = 0$ and $f(x, y) = 1$ otherwise. Let $I = f^{-1}(\{1\})$. Let $(x, y), (x', y') \in I$ with $(x, y) \neq (x', y')$ then for $(x, y) < (x', y')$ for instance. Since $f(x, y) = f(x', y') = 1$ we have that $(\mathsf{C}_0^{(x,y)} \times \mathsf{C}_1^{(x,y)}) \cap (\mathsf{C}_0^{(x',y')} \times \mathsf{C}_1^{(x',y')}) = \emptyset$. Let $(x, y) \in \mathsf{A}_b$. If $f(x, y) = 1$ then $(x, y) \in \mathsf{C}_0^{(x,y)} \times \mathsf{C}_1^{(x,y)}$. If $f(x, y) = 0$ then there exists $(x', y') < (x, y)$ such that $(x, y) \in \mathsf{C}_0^{(x',y')} \times \mathsf{C}_1^{(x',y')}$. Therefore, we get that $\{\mathsf{C}^{(x,y)} = (\mathsf{C}_0^{(x,y)} \times \mathsf{C}_1^{(x,y)}) \cap \mathsf{A}_b : (x, y) \in I\}$ is a partition of $\mathsf{A}_b$.

Since $\mu(\mathsf{A}_b) \leq 1$, and $\{\mathsf{C}^{(x,y)} = (\mathsf{C}_0^{(x,y)} \times \mathsf{C}_1^{(x,y)}) \cap \mathsf{A}_b : (x, y) \in I\}$ is a partition of $\mathsf{A}_b$, we get that $J = \{\mathsf{C}^{(x,y)} : (x, y) \in I, \mu_0(\mathsf{C}_0^{(x,y)})\mu_1(\mathsf{C}_1^{(x,y)}) > 0\}$ is countable. Denote $\mathsf{A}_c = \cup_{(x,y)\in J} \mathsf{C}^{(x,y)}$. Let us show that $\mu(\mathsf{A}_c^c \cap \mathsf{A}_b) = \mu(\cup_{(x,y)\in I\cap J^c} \mathsf{C}^{(x,y)}) = 0$. Let $x \in \mathbb{R}^d$ and define $\mathsf{D}_x = \{y \in \mathbb{R}^d : (x, y) \in \mathsf{A}_b \cap \mathsf{A}_c^c\}$. If $\mathsf{D}_x$ is not empty, then there exists $(x', y') \in I$ such that $x \in \mathsf{C}_0^{(x',y')}$. Then, for any $y \in \mathsf{D}_x$, $y \in \mathsf{C}_1^{(x',y')}$. Hence, $(x', y') \in I \cap J^c$ by definition of $\mathsf{D}_x$ and $\mu_1(\mathsf{D}_x) = 0$. We get that

$$\mu(\mathsf{A}_b \cap \mathsf{A}_c^c) = \int_{\mathbb{R}^d}\left(\int_{\mathsf{D}_x} h(x, y)\mathrm{d}\mu_1(y)\right)\mathrm{d}\mu_0(x) = 0,$$

where $h$ is the density of $\mu$ w.r.t. $\mu_0 \otimes \mu_1$. Note that this is the only instance in the proof, where we use that $\mu$ is absolutely continuous w.r.t. $\mu_0 \otimes \mu_1$. For any $(x, y) \in \mathsf{A}_c$ define for any $n \in \mathbb{N}$

$$\hat{a}_n(x) = \sum_{(x',y')\in J} \mathbb{1}_{\mathsf{C}_0^{(x',y')}}(x) a_n^{(x',y')}(x), \quad \hat{b}_n(y) = \sum_{(x',y')\in J} \mathbb{1}_{\mathsf{C}_1^{(x',y')}}(x) b_n^{(x',y')}(y).$$

There exist $\hat{a}, \hat{b} : \mathbb{R}^d \to (0, +\infty)$ such that for any $(x, y) \in \mathsf{A}_c$, $\lim_{n \to +\infty} \hat{a}_n(x) = \hat{a}(x)$ and $\lim_{n \to +\infty} \hat{b}_n(y) = \hat{b}(y)$. In addition, for any $(x, y) \in \mathsf{A}_c$, $a_n(x)b_n(y) = \hat{a}_n(x)\hat{b}_n(y)$. Hence, for any $(x, y) \in \mathsf{A}_c$, $\Phi(x, y) = \hat{a}(x)\hat{b}(y)$. Since $\mathsf{A}_a \cap \mathsf{A}_c = \emptyset$ and $\mu(\mathsf{A}_c) = \mu(\mathsf{A}_b)$, we have

$$\mu(\mathsf{A}_a) + \mu(\mathsf{A}_c) = \mu(\mathsf{A}_a) + \mu(\mathsf{A}_b) = \mu(\mathsf{A}) = 1.$$

We conclude the proof upon remarking that for any $(x, y) \in \mathsf{A}_a$, $\Phi(x, y) = 0$ and for any $(x, y) \in \mathsf{A}_c$, $\Phi(x) = \hat{a}(x)\hat{b}(y)$. $\qquad\square$

In what follows we prove Proposition S33.

*Proof.* Since $\lim_{n \to +\infty} \|\pi^n - \pi^\infty\|_{\mathrm{TV}} = 0$, there exist $\mathsf{A}$ with $\mu(\mathsf{A}) = 1$ and $\Phi : (\mathbb{R}^d)^2 \to [0, +\infty)$ such that, up to extraction, for any $x, y \in \mathsf{A}$

$$\lim_{n \to +\infty} a_n(x)b_n(y) = \Phi(x, y),$$

and $(\mathrm{d}\pi^\infty/\mathrm{d}\mu) = \Phi$. Using Proposition S34, there exist $a, b : \mathbb{R}^d \to [0, +\infty)$ and $\mathsf{B}$ with $\pi^\infty(\mathsf{B}) = 1$ such that for any $x, y \in \mathsf{B}$, $(\mathrm{d}\pi^\infty/\mathrm{d}\mu)(x, y) = a(x)b(y)$. Since $\mu$ is absolutely continuous w.r.t. $\pi^\infty$, we get that for any $x, y \in \mathbb{R}^d$, $(\mathrm{d}\pi^\infty/\mathrm{d}(\mu_0 \otimes \mu_1))(x, y) = a(x)b(y)h(x, y)$. In addition, the Schrödinger bridge $\pi^\star \in \mathscr{P}((\mathbb{R}^d)^2)$ exists, see (Rüschendorf and Thomsen, 1993, Theorem 3), and there exist $a', b' : \mathbb{R}^d \to [0, +\infty)$ and $\mathsf{B}'$ with $\mu(\mathsf{B}') = 1$ such that for any $x, y \in \mathsf{B}'$

$$(\mathrm{d}\pi^\star/\mathrm{d}(\mu_0 \otimes \mu_1))(x, y) = a'(x)b'(y)h(x, y).$$

Let $\mathscr{M}_{+,\times}$ be the space of non-negative product measures over $\mathcal{B}(\mathbb{R}^d) \otimes \mathcal{B}(\mathbb{R}^d)$. Let $\Psi_{\bar{h}} : \mathscr{M}_{+,\times} \to \mathscr{M}_{+,\times}$ be given for any $\lambda = \lambda_0 \otimes \lambda_1 \in \mathscr{M}_{+,\times}$ by $\Psi_{\bar{h}}(\lambda) = \Psi_h^\lambda$ where for any $\mathsf{A}, \mathsf{B} \in \mathcal{B}(\mathbb{R}^d)$

$$\Psi_h^\lambda(\mathsf{A} \times \mathsf{B}) = (\textstyle\int_{\mathsf{A} \times \mathbb{R}^d} \bar{h}(x, y)\mathrm{d}\lambda_0(x)\mathrm{d}\lambda_1(x))(\int_{\mathbb{R}^d \times \mathsf{B}} \bar{h}(x, y)\mathrm{d}\lambda_0(x)\mathrm{d}\lambda_1(y))$$

where for any $x, y \in \mathbb{R}^d$, $\bar{h}(x, y) = h(x, y)\Phi_0^{-1}(x)\Phi_1^{-1}(y)$. Note that $\bar{h} \in \mathrm{C}(\mathbb{R}^d \times \mathbb{R}^d, [0, +\infty))$ and is bounded. Hence, using (Beurling, 1960, Theorem 2) and (S66) we get that $\Psi_{\bar{h}}$ is a bijection. Let $\lambda = (a\Phi_0\mu_0, b\Phi_1\mu_1)$ and $\lambda' = (a'\Phi_0\mu_0, b'\Phi_1\mu_1)$. Then, since $\pi_i^\star = \pi_i^\infty = \nu_i$ for $i \in \{0, 1\}$ we get that $\Psi_h(\lambda) = \Psi_h(\lambda')$. Hence $\lambda = \lambda'$ and $\pi^\infty = \pi^\star$ which concludes the proof. $\qquad\square$

In Proposition S36 we derive an alternative proposition to Proposition S33. We start with the following lemma.

**Lemma S35.** *Let $\pi^\star \in \mathscr{P}_2$ with $\pi_i^\star = \nu_i$ for $i \in \{0, 1\}$. Assume that $\mathrm{KL}(\pi^\star|\mu) < +\infty$ and that $\mathrm{L}^1(\nu_0) \oplus \mathrm{L}^1(\nu_1)$ is closed in $\mathrm{L}^1(\pi^\star)$. In addition, assume that there exist $a, b : \mathbb{R}^d \to [0, +\infty)$ and $\mathsf{A}$ with $\pi^\star(\mathsf{A}) = 1$ such that for any $(x, y) \in \mathsf{A}$,*

$$(\mathrm{d}\pi^\star/\mathrm{d}\mu)(x, y) = a(x)b(y).$$

*Then $\pi^\star$ is the Schrödinger bridge.*

*Proof.* Since $\mathrm{KL}(\pi^\star|\mu) < +\infty$ we have that

$$\textstyle\int_{(\mathbb{R}^d)^2} |\log(a(x)b(y))|\, \mathrm{d}\pi^\star(x, y) < +\infty.$$

Using (Kober, 1939, Theorem 1) and that $\pi_i^\star = \nu_i$ for $i \in \{0, 1\}$, we get that

$$\textstyle\int_{\mathbb{R}^d} |\log a(x)|\, \mathrm{d}\nu_0(x) + \int_{\mathbb{R}^d} |\log b(y)|\, \mathrm{d}\nu_1(y) < +\infty. \tag{S68}$$

Let $\pi \in \mathscr{P}_2$ such that $\pi_i = \nu_i$ for $i \in \{1, 2\}$ and $\mathrm{KL}(\pi|\mu) < +\infty$. Using (S68), we have that $\int_{(\mathbb{R}^d)^2} |\log((\mathrm{d}\pi^\star/\mathrm{d}\mu)(x, y))|\, \mathrm{d}\pi(x, y) < +\infty$. Hence, $(\mathrm{d}\pi^\star/\mathrm{d}\mu) > 0$, $\pi$-almost surely. Using this result we have for any $\mathsf{A} \in \mathcal{B}(\mathbb{R}^d)$

$$\begin{aligned}
\pi(\mathsf{A}) &= \textstyle\int_{\mathbb{R}^d} \mathbb{1}_{\mathsf{A}}(x)(\mathrm{d}\pi^\star/\mathrm{d}\mu)(x)(\mathrm{d}\pi^\star/\mathrm{d}\mu)(x)^{-1}\mathrm{d}\pi(x) \\
&= \textstyle\int_{\mathbb{R}^d} \mathbb{1}_{\mathsf{A}}(x)(\mathrm{d}\pi^\star/\mathrm{d}\mu)(x)(\mathrm{d}\pi^\star/\mathrm{d}\mu)(x)^{-1}(\mathrm{d}\pi/\mathrm{d}\mu)(x)\mathrm{d}\mu(x) \\
&= \textstyle\int_{\mathbb{R}^d} \mathbb{1}_{\mathsf{A}}(x)(\mathrm{d}\pi^\star/\mathrm{d}\mu)(x)^{-1}(\mathrm{d}\pi/\mathrm{d}\mu)(x)\mathrm{d}\pi^\star(x).
\end{aligned}$$

Hence we get that $\mathrm{d}\pi/\mathrm{d}\pi^\star = (\mathrm{d}\pi/\mathrm{d}\mu)(\mathrm{d}\pi^\star/\mathrm{d}\mu)^{-1}$. In addition, we have that

$$\mathrm{KL}(\pi^\star|\mu) = \int_{\mathbb{R}^d} \log(a(x))\mathrm{d}\nu_0(x) + \int_{\mathbb{R}^d} \log(b(y))\mathrm{d}\nu_1(y) = \int_{(\mathbb{R}^d)^2} \log((\mathrm{d}\pi^\star/\mathrm{d}\mu)(x,y))\mathrm{d}\pi(x,y).$$

We get that

$$\mathrm{KL}(\pi|\pi^\star) = \int_{\mathbb{R}^d} \log((\mathrm{d}\pi/\mathrm{d}\mu)(\mathrm{d}\pi^\star/\mathrm{d}\mu)(x,y)^{-1})\mathrm{d}\pi(x,y) = \mathrm{KL}(\pi|\mu) - \mathrm{KL}(\pi^\star|\mu).$$

Hence, $\mathrm{KL}(\pi|\mu) \geq \mathrm{KL}(\pi^\star|\mu)$ with equality if and only if $\pi^\star = \pi$. Therefore, $\pi^\star$ is the Schrödinger bridge. $\qquad\square$

The following proposition is an alternative to Proposition S33.

**Proposition S36.** *Assume* **B**1. *Then there exists a solution $\pi^\star$ to the Schrödinger bridge. Assume that the IPF sequence $(\pi^n)_{n\in\mathbb{N}}$ satisfies $\lim_{n\to+\infty} \|\pi^n - \pi^\infty\|_{\mathrm{TV}} = 0$ with $\pi^\infty \in \mathscr{P}_2$. If $\mathrm{L}^1(\nu_0) \oplus \mathrm{L}^1(\nu_1)$ is closed in $\mathrm{L}^1(\pi^\infty)$ then $\pi^\infty = \pi^\star$.*

*Proof.* Since $\lim_{n\to+\infty} \|\pi^n - \pi^\infty\|_{\mathrm{TV}} = 0$ there exist $\mathsf{A}$ with $\mu(\mathsf{A}) = 1$ and $\Phi : (\mathbb{R}^d)^2 \to [0, +\infty)$ such that, up to extraction, for any $x, y \in \mathsf{A}$

$$\lim_{n\to+\infty} a_n(x)b_n(y) = \Phi(x,y),$$

and $(\mathrm{d}\pi^\infty/\mathrm{d}\mu) = \Phi$. Using Proposition S34, there exist $a, b : \mathbb{R}^d \to [0, +\infty)$ and $\mathsf{B}$ with $\pi^\infty(\mathsf{B}) = 1$ such that for any $x, y \in \mathsf{B}$, $(\mathrm{d}\pi^\infty/\mathrm{d}\mu)(x,y) = a(x)b(y)$. We conclude upon using Lemma S35. $\qquad\square$

# S7  Geometric convergence rates and convergence to ground-truth

In this section, we derive geometric convergence rates in Section S7.1 in a Gaussian setting. In particular, we provide an explicit upper-bound on the convergence rate that depends only on the covariance of the reference measure and the target. In Section S7.2, we show that DSB (with Brownian reference measure) converges towards the Schrödinger bridge in a Gaussian setting where the ground-truth is available. In Section 4 we show that our implementation actually recovers the Schrödinger bridge in this setting.

## S7.1  Geometric convergence rates

In the following proposition we show that we recover a geometric convergence rate in a Gaussian setting and derive intuition from this case study. We set $N = 1$ and assume that for any $x_0, x_N \in \mathbb{R}^d$ we have

$$p(x_0, x_N) \propto \exp[-\|x_0\|^2 + 2\alpha\langle x_0, x_N\rangle - \|x_N\|^2],$$

with $\alpha \in [0, 1)$. In this case assume that there exists $\beta > 0$ such that the target marginals are given for any $x_0, x_N \in \mathbb{R}^d$ by

$$p_{\mathrm{data}}(x_0) \propto \exp[-\beta\|x_0\|^2], \qquad p_{\mathrm{prior}}(x_N) \propto \exp[-\beta\|x_N\|^2].$$

**Proposition S37.** *Let $\alpha \in (0, 1)$ and $\beta > 0$. Then the Schrödinger bridge $\pi^\star$ exists and there exists $C \geq 0$ (explicit in the proof) such that for any $n \in \mathbb{N}$, $\mathrm{KL}(\pi^\star|\pi^n) \leq C\kappa^{2n}$, with $\kappa < 1$ given by $\kappa = \rho/(1+\rho)$ and $\rho = 2\alpha/\beta^2$. In addition, $\pi^\star$ admits a density w.r.t. the Lebesgue measure denoted $p^\star$ and given for any $x, y \in \mathbb{R}^d$ by*

$$p^\star(x, y) = \exp[-\gamma^\star\|x\|^2 + 2\alpha\langle x, y\rangle - \gamma^\star\|y\|^2] / \int_{\mathbb{R}^d} \exp[-\gamma^\star\|x\|^2 + 2\alpha\langle x, y\rangle - \gamma^\star\|y\|^2]\mathrm{d}x\mathrm{d}y,$$

*with $\gamma^\star = (\beta^2/2)(1 + (1 + 4\alpha^2/\beta^2)^{1/2})$.*

Remark that if $\beta^2 = 1 - \alpha^2$ then $\gamma^\star$ and $p^\star = p$, *i.e.* the IPF leaves $\mu$ invariant. Note that the performance of the IPF improves if $\kappa$ is close to $0$, *i.e.* if $\rho = 2\alpha/\beta^2$ is close to $0$. This is the case if $\alpha \approx 0$ (the marginals are almost independent) or if $\beta \approx +\infty$ (the target distribution is close to $\delta_0$), see Figure S1. This behavior is in accordance with the limit case where the marginals are independent or one of the target distribution is a Dirac mass in which case the IPF converges in two iterations.

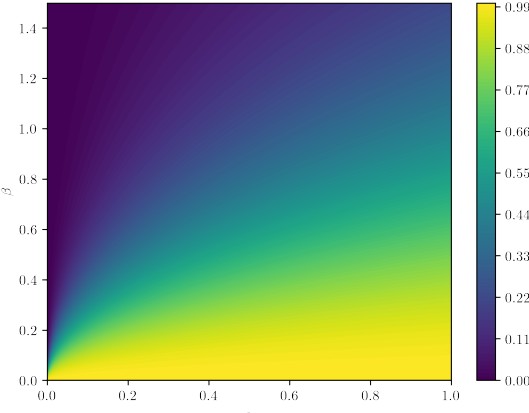

Figure S1: Evolution of $\kappa^2$ depending on $\alpha$ and $\beta$.

Also, note that the convergence rate does not depend on the dimension but only on the constants of the problem. In what follows we first derive the IPF sequence for this Gaussian problem and establish that $\alpha$ controls the amount of information shared by the marginals. Then we prove Proposition S37. In the rest of this section, we let $\mu \in \mathscr{P}_2$ with density $p$ w.r.t. the Lebesgue measure such that for any $x_0, x_1 \in \mathbb{R}^d$

$$p(x_0, x_1) = \exp[-\|x_0\|^2 + 2\alpha\langle x_0, x_1\rangle - \|x_1\|^2]/\int_{\mathbb{R}^d} \exp[-\|x_0\|^2 + 2\alpha\langle x_0, x_1\rangle - \|x_1\|^2]\mathrm{d}x_0\mathrm{d}x_1.$$

We have that $\mu$ is the Gaussian distribution with zero mean and covariance matrix $\Sigma$ such that

$$\Sigma = (2(1 - \alpha^2))^{-1}\begin{pmatrix} \mathrm{Id} & \alpha\,\mathrm{Id} \\ \alpha\,\mathrm{Id} & \mathrm{Id} \end{pmatrix}.$$

We have that $\det(\Sigma) = 2^{2d}(1 - \alpha^2)^{-d}$ using Schur complement (Petersen et al., 2008, Section 9.1.2). Hence we get that for any $x_0, x_1 \in \mathbb{R}^d$

$$p(x_0, x_1) = \pi^{-d}(1 - \alpha^2)^{d/2}\exp[-\|x_0\|^2 + 2\alpha\langle x_0, x_1\rangle - \|x_1\|^2].$$

In what follows, we denote $C = \pi^d(1 - \alpha^2)^{-d/2}$. Similarly, we get that $\mu_0 = \mu_1$ and that they admit the density $p_0$ w.r.t. the Lebesgue measure given for any $x \in \mathbb{R}^d$ by

$$p_0(x) = \pi^{-d/2}(1 - \alpha^2)^{d/2}\exp[-\|x\|^2(1 - \alpha^2)].$$

In what follows, we denote $C_0 = \pi^{d/2}(1 - \alpha^2)^{-d/2}$. In this case note that $\mu$ admits a density w.r.t. $\mu_0 \otimes \mu_1$ given for any $x_0, x_1 \in \mathbb{R}^d$ by

$$h(x_0, x_1) = (\mathrm{d}\mu/\mathrm{d}(\mu_0 \otimes \mu_1))(x_0, x_1) = (1 - \alpha^2)^{-d/2}\exp[-\alpha^2\|x_0\|^2 - 2\alpha\langle x_0, x_1\rangle - \alpha^2\|x_1\|^2].$$

Remark that $p_{\mathrm{prior}} = p_{\mathrm{data}} = q$ with for any $x \in \mathbb{R}^d$, $q(x) = \pi^{-d/2}\beta^{d/2}\exp[-\beta\|x\|^2]$. We have for any $x_1, x_0 \in \mathbb{R}^d$

$$p_{1|0}(x_1|x_0) = p(x_0, x_1)/p_0(x_0) = \pi^{-d/2}(1 - \alpha^2)^{d/2}\exp[-\alpha^2\|x_0\|^2 + 2\alpha\langle x_0, x_1\rangle - \|x_1\|^2].$$

Hence, we have that **A**1 holds and the IPF sequence is well-defined and converges using Proposition 5. In what follows we start to show that $\alpha$ controls the amount of information shared by the two marginals $\mu_0$ and $\mu_1$, *i.e.* the mutual information. More precisely we have the following result.

**Proposition S38.** *For any $\alpha \in (0, 1)$ we have* $\mathrm{KL}(\mu|\mu_0 \otimes \mu_1) = -(d/2)\log(1 - \alpha^2)$.

*Proof.* For any $x, y \in \mathbb{R}^d$ we have

$$(\mathrm{d}\mu/(\mathrm{d}\mu_0 \otimes \mathrm{d}\mu_1))(x, y) = \exp[-\alpha^2\|x\|^2 + 2\alpha\langle x, y\rangle - \alpha^2\|y\|^2](1 - \alpha^2)^{-d/2}.$$

We have that

$$\int_{\mathbb{R}^d \times \mathbb{R}^d}(-\alpha^2\|x\|^2 - \alpha^2\|y\|^2 + 2\alpha\langle x, y\rangle)\mathrm{d}\mu(x, y) = 0.$$

Hence, $\mathrm{KL}(\mu|\mu_0 \otimes \mu_1) = -(d/2)\log(1 - \alpha^2)$, which concludes the proof. $\qquad\square$

In what follows, we denote by $(\pi^n)_{n\in\mathbb{N}}$ the IPFP sequence, defined for any $n \in \mathbb{N}$ we have for any $x, y \in \mathbb{R}^d$

$$(\mathrm{d}\pi^{2n}/\mathrm{d}\mu)(x,y) = a_n(x)b_n(y)h(x,y), \qquad (\mathrm{d}\pi^{2n+1}/\mathrm{d}\mu)(x,y) = a_{n+1}(x)b_n(y)h(x,y),$$

where for any $x, y \in \mathbb{R}^d$

$$a_{n+1}(x) = (\mathrm{d}\nu_0/\mathrm{d}\mu_0)(x) \left( \textstyle\int_{\mathbb{R}^d} h(x,y)b_n(y)\mathrm{d}\mu_1(y) \right)^{-1},$$
$$b_{n+1}(x) = (\mathrm{d}\nu_1/\mathrm{d}\mu_1)(y) \left( \textstyle\int_{\mathbb{R}^d} h(x,y)a_{n+1}(x)\mathrm{d}\mu_0(x) \right)^{-1}.$$

We now turn to the proof of the Proposition S37.

*Proof.* Let $\alpha \in (0,1)$ and $\beta > 1$. We have for any $x, y \in \mathbb{R}^d$

$$(\mathrm{d}\nu_0/\mathrm{d}\mu_0)(x) = \exp[(1-\beta^2-\alpha^2)\|x\|^2]/C_2, \qquad (\mathrm{d}\nu_1/\mathrm{d}\mu_1)(y) = \exp[(1-\beta^2-\alpha^2)\|y\|^2]/C_2,$$

with $C_2 = C_1/C_0$ with $C_1 = \pi^{d/2}\beta^{d/2}$. For any $x \in \mathbb{R}^d$ and $\gamma \geq 0$ we have

$$(\mathrm{d}\nu_0/\mathrm{d}\mu_0)(x) \left( \textstyle\int_{\mathbb{R}^d} \exp[-\gamma\|y\|^2]h(x,y)\mathrm{d}\mu_1(y) \right)^{-1}$$
$$= (C_0C_2)^{-1}C \exp[(1-\beta^2-\alpha^2)\|x\|^2] \left( \textstyle\int_{\mathbb{R}^d} \exp[-\gamma\|y\|^2 - \|y-\alpha x\|^2]\mathrm{d}y \right)^{-1}$$
$$= (C_0C_2)^{-1}C \exp[(1-\beta^2-\alpha^2)\|x\|^2]$$
$$\times \left( \textstyle\int_{\mathbb{R}^d} \exp[-(\gamma+1)\|y-\alpha/(\gamma+1)x\|^2 - \alpha^2(1-1/(\gamma+1))\|x\|^2]\mathrm{d}y \right)^{-1}$$
$$= (C_0C_2)^{-1}C \exp[(1-\beta^2-\alpha^2+\alpha^2\gamma/(\gamma+1))\|x\|^2]$$
$$\times \left( \textstyle\int_{\mathbb{R}^d} \exp[-(\gamma+1)\|y-\alpha/(\gamma+1)x\|^2]\mathrm{d}y \right)^{-1}$$
$$= (C_0C_2\tilde{C}_\gamma)^{-1}C \exp[(1-\beta^2-\alpha^2/(\gamma+1))\|x\|^2],$$

with $\tilde{C}_\gamma = \pi^{d/2}(1+\gamma)^{-d/2}$. Note that $a_0 = b_0 = 1$. Let $n \in \mathbb{N}$ and assume that for any $y \in \mathbb{R}^d$ $b_n(y) = \exp[-\gamma_{2n}\|y\|^2]/C_{2n}$ with $\gamma_{2n} \geq 0$ and $C_{2n} > 0$ then we have for any $x \in \mathbb{R}^d$

$$a_{n+1}(x) = (C_0C_2\tilde{C}_{\gamma_{2n}})^{-1}CC_{2n}\exp[-(1-\beta^2-\alpha^2/(\gamma_{2n}+1))\|x\|^2] = \exp[-\gamma_{2n+1}\|x\|^2]/C_{2n+1},$$

with

$$\gamma_{2n+1} = \beta^2 - 1 + \alpha^2/(\gamma_{2n}+1), \qquad (C_0C_2\tilde{C}_{\gamma_{2n}})/(CC_{2n}) = C_{2n+1}. \tag{S69}$$

Similarly, if we assume that for any $x \in \mathbb{R}^d$ $a_{n+1}(x) = \exp[-\gamma_{2n+1}\|x\|^2]/C_{2n+1}$ with $\gamma_{2n+1} \geq 0$ and $C_{2n+1} > 0$ then we have for any $y \in \mathbb{R}^d$

$$b_{n+1}(y) = (C_0C_2\tilde{C}_{\gamma_{2n+1}})^{-1}(CC_{2n+1})\exp[-(1-\beta^2-\alpha^2/(\gamma_{2n+1}+1))\|y\|^2]$$
$$= \exp[-\gamma_{2n+2}\|y\|^2]/C_{2n+2},$$

with

$$\gamma_{2n+2} = \beta^2 - 1 + \alpha^2/(\gamma_{2n+1}+1), \qquad (C_0C_2\tilde{C}_{\gamma_{2n+1}})/(CC_{2n+1}) = C_{2n+2}.$$

Combining this result, (S69) and using the recursion principle we get that for any $n \in \mathbb{N}$

$$a_{n+1}(x) = \exp[-\gamma_{2n+1}\|x\|^2]/C_{2n+1}, \qquad b_{n+1}(y) = \exp[-\gamma_{2n+2}\|y\|^2]/C_{2n+2}.$$

The recursion can be extended to $a_0$ and $b_0$ by setting $\gamma_{-1} = \gamma_0 = 0$ and $C_{-1} = C_0 = 1$. Therefore, for any $n \in \mathbb{N}$ we have

$$\gamma_{n+1} = \beta^2 - 1 + \alpha^2/(\gamma_n+1). \tag{S70}$$

We now study the convergence of the sequence $(\gamma_n)_{n\in\mathbb{N}}$. By recursion, we have that for any $k, \ell \in \mathbb{N}$, if $\gamma_k \geq \gamma_\ell$ then for any $m \in \mathbb{N}$ with $m$ even we have $\gamma_{m+k} \geq \gamma_{m+\ell}$ and for any $m \in \mathbb{N}$ with $m$ odd we have $\gamma_{m+k} \leq \gamma_{m+\ell}$. We have $\gamma_0 = 0$ and

$$\gamma_1 = \beta^2 + \alpha^2 - 1, \qquad \gamma_2 = \beta^2 - 1 + \alpha^2/(\beta^2+\alpha^2). \tag{S71}$$

We divide the rest of the proof into three parts.

(a) First assume that $\beta^2 > 1 - \alpha^2$. Using (S71) we have that $\gamma_1 > \gamma_0$ and $\gamma_2 > \gamma_0$. Therefore, we obtain that $(\gamma_{2n})_{n\in\mathbb{N}}$ is non-decreasing, that $(\gamma_{2n+1})_{n\in\mathbb{N}}$ is non-increasing and that for any $n \in \mathbb{N}$, $0 \leq \gamma_{2n} \leq \gamma_{2n+1} \leq \gamma_1$. Therefore, $(\gamma_n)_{n\in\mathbb{N}}$ converges and we denote $\gamma^\star$ its limit. We have $\gamma^\star = \beta^2 - 1 + \alpha^2/(\gamma^\star + 1)$. Hence, $\gamma^\star$ is a root of $X^2 + (2 - \beta^2)X + 1 - \alpha^2 - \beta^2$. We get that $\gamma^\star = \gamma_0^\star$ or $\gamma^\star = \gamma_1^\star$ with

$$\gamma_0^\star = \beta^2/2 - 1 - (1/2)(\beta^4 + 4\alpha^2)^{1/2}, \qquad \gamma_1^\star = \beta^2/2 - 1 + (1/2)(\beta^4 + 4\alpha^2)^{1/2},$$

$\gamma_0^\star, \gamma_1^\star$ are non-decreasing function of $\beta$. We get that for any $\beta \geq 0$ such that $\beta^2 \geq 1 - \alpha^2$, $\gamma_0^\star \leq 0$. In addition, we have $\gamma_1^\star = 0$ for $\beta^2 = 1 - \alpha^2$, hence for any $\beta \geq 0$ such that $\beta^2 \geq 1 - \alpha^2$, $\gamma_1^\star \geq 0$. Since $\gamma^\star \geq 0$ we have

$$\gamma^\star = -1 + \beta^2/2 + (1/2)(\beta^4 + 4\alpha^2)^{1/2}. \tag{S72}$$

For any $n \in \mathbb{N}$, denote $\xi_n = \gamma_n - \gamma^\star$ and $\tau = \gamma^\star + 1$. Let $\varepsilon > 0$. Since $\lim_{n\to+\infty} \xi_n = 0$, there exists $n_0 \in \mathbb{N}$ such that $|\xi_n|/\tau \leq \varepsilon$. Using (S70), we obtain that for any $n \in \mathbb{N}$

$$|\xi_{n+1}| = \alpha^2|1/(\gamma_n + 1) - \tau^{-1}| = (\alpha^2/\tau)|1 - (\xi_n/\tau + 1)^{-1}| \leq (\alpha/\tau)^2|\xi_n|/(1 - \varepsilon).$$

Hence, we get that for any $\varepsilon \in (0, 1)$, there exists $C_\varepsilon > 0$ such that for any $n \in \mathbb{N}$

$$|\xi_n| \leq C_\varepsilon \kappa^n, \qquad \kappa = (\alpha/(\tau(1-\varepsilon)^{1/2}))^2.$$

Note that $\tau > \alpha$ using (S72) and $\kappa \in (0, 1)$ if $\varepsilon < 1 - \alpha/\tau$.

For any $n \in \mathbb{N}$ and $x, y \in \mathbb{R}^d$ we have

$$\Phi_n(x, y) = a_{n+1}(x)b_{n+1}(y) = \exp[-\gamma_{2n+1} \|x\|^2 - \gamma_{2n+2} \|y\|^2]/(C_{2n+1}C_{2n+2})$$

$$= \exp[-\gamma_{2n+1} \|x\|^2 - \gamma_{2n+2} \|y\|^2]/(\tilde{C}\tilde{C}_{\gamma_{2n+1}}),$$

with $\tilde{C} = C_0 C_2/C$. Therefore we obtain that for any $x, y \in \mathbb{R}^d$, $\Phi^\star(x, y) = \lim_{n\to+\infty} \Phi_n(x, y)$ exists and we have

$$\Phi^\star(x, y) = \exp[-\gamma^\star \|x\|^2 - \gamma^\star \|y\|^2]/(\tilde{C}\tilde{C}_{\gamma^\star}).$$

Using this result we get that for any $x, y \in \mathbb{R}^d$

$$(\mathrm{d}\pi^{2n}/\mathrm{d}\pi^\star)(x, y) = \exp[-\xi_{2n+1} \|x\|^2 - \xi_{2n+2} \|y\|^2]C_{\gamma^\star}/C_{\gamma_{2n+1}}$$

$$= \exp[-\xi_{2n+1} \|x\|^2 - \xi_{2n+2} \|y\|^2] \{(1 + \gamma_{2n+1})/(1 + \gamma^\star)\}^{-d/2}$$

$$= \exp[-\xi_{2n+1} \|x\|^2 - \xi_{2n+2} \|y\|^2] \{1 + \xi_{2n+1}/(1 + \gamma^\star)\}^{-d/2}.$$

Therefore we have for any $x, y \in \mathbb{R}^d$

$$\log\left((\mathrm{d}\pi^{2n}/\mathrm{d}\pi^\star)(x, y)\right) \leq |\xi_{2n+1}| \|x\|^2 + |\xi_{2n+2}| \|y\|^2 + (d/2) |\log(1 + \xi_{2n+1}/(1 + \gamma^\star))|$$

$$\leq |\xi_{2n+1}| \|x\|^2 + |\xi_{2n+2}| \|y\|^2 + (d/2) |\xi_{2n+1}|.$$

Therefore we obtain that for any $n \in \mathbb{N}$

$$\mathrm{KL}(\pi^\star|\pi^n) \leq (d/2)(\beta^{-2} |\xi_{2n+1}| + \beta^{-2} |\xi_{2n+2}| + |\xi_{2n+1}|).$$

A similar inequality holds for $\mathrm{KL}(\pi^\star|\pi^n)$. Therefore we get that for any $\varepsilon \in (0, 1 - \alpha/\tau)$ there exists $C_\varepsilon \geq 0$ such that for any $n \in \mathbb{N}$ we have

$$\mathrm{KL}(\pi^\star|\pi^n) \leq C_\varepsilon \kappa_\varepsilon^{2n},$$

with

$$\kappa_\varepsilon = \alpha/(\tau(1-\varepsilon)^{1/2}) = (2\alpha)/((\beta^2 + (\beta^4 + 4\alpha^2)^{1/2})(1-\varepsilon)^{1/2})$$

$$\leq \rho/((1 + (1 + \rho^2)^{1/2})(1 - \varepsilon)^{1/2}).$$

Let $\varepsilon < 1 - (1 + \rho)/(1 + (1 + \rho^2)^{1/2})$. Then we get that $\kappa_\varepsilon \leq \kappa$ which concludes the first part of the proof.

(b) If $\beta^2 = 1 - \alpha^2$ then the IPF is stationary since the IPF leaves $\mu$ invariant.

(c) Finally we assume that $\beta^2 < 1 - \alpha^2$. Using (S71) we have that $\gamma_1 < \gamma_0$ and $\gamma_2 < \gamma_0$ since $\beta^2 < 1 - \alpha^2$. Therefore, we obtain that $(\gamma_{2n})_{n\in\mathbb{N}}$ is non-increasing, that $(\gamma_{2n+1})_{n\in\mathbb{N}}$ is non-decreasing and that for any $n \in \mathbb{N}$, $0 \geq \gamma_{2n} \geq \gamma_{2n+1} \geq \gamma_1$. Therefore, $(\gamma_n)_{n\in\mathbb{N}}$ converges and we denote $\gamma^\star$ its limit. We have $\gamma^\star = \beta^2 - 1 + \alpha^2/(\gamma^\star + 1)$. Hence, $\gamma^\star$ is a root of $X^2 + (2 - \beta^2)X + 1 - \alpha^2 - \beta^2$. We recall that the two roots of this polynomial are given by

$$\gamma_0^\star = \beta^2/2 - 1 - (1/2)(\beta^4 + 4\alpha^2)^{1/2}, \qquad \gamma_1^\star = \beta^2/2 - 1 + (1/2)(\beta^4 + 4\alpha^2)^{1/2}.$$

We have

$$\gamma_1 - \gamma_0^\star = \beta^2 + \alpha^2 - 1 - \beta^2/2 + 1 - (1/2)(\beta^4 + 4\alpha^2)^{1/2}$$
$$= (1/2)(\beta^2 + 2\alpha^2 - (\beta^4 + 4\alpha^2)^{1/2}) \geq 0.$$

Since $\gamma_3 > \gamma_1$ we get that for any $n \in \mathbb{N}$ with $n \geq 3$, $\gamma_n \geq \gamma_3 > \gamma_0^\star$. Therefore $\gamma^\star > \gamma_0^\star$ and then $\gamma^\star = \gamma_1^\star$. The rest of the proof is similar to the case where $\beta^2 > 1 - \alpha^2$.

$\square$

## S7.2 Convergence to ground-truth

In this section, we provide an analytic form for the Schrödinger bridge in a Gaussian context. Let $\nu_0$ be the $d$ dimensional Gaussian distribution with mean $-a$ (with $a \in \mathbb{R}^d$) and covariance matrix $\mathbf{I} \in \mathbb{R}^{d\times d}$. Similarly, let $\nu_1$ be the one-dimensional Gaussian distribution with mean $a$ and covariance matrix $\mathbf{I}$. We consider the reference distribution $\pi^0$ such that $\pi_0^0 = \nu_0$ and for any $x, y \in \mathbb{R}^d$

$$(\mathrm{d}\pi_{1|0}^0/\mathrm{d}\lambda)(x, y) = (2\pi)^{-d/2}\exp[-\|x - y\|^2/2],$$

where $\lambda$ denotes the Lebesgue measure on $\mathbb{R}$. Note that $\pi_{1|0}^0$ can be obtained by running a $d$-dimensional Brownian motion up to time 1. We consider the following Schrödinger bridge problem

$$\pi^\star = \arg\min\{\mathrm{KL}(\pi|\pi^0) \; : \; \pi \in \mathscr{P}(\mathbb{R}^{2d}), \; \pi_0 = \nu_0, \pi_1 = \nu_1\}. \tag{S73}$$

Before giving the analytic solution of the SB problem we consider the following algebraic lemma.

**Lemma S39.** *Let $A \in \mathbb{R}^{d\times d}$ and*

$$M = \begin{pmatrix} \mathbf{I} & A \\ A^\top & \mathbf{I} \end{pmatrix}, \qquad M^S = \begin{pmatrix} \mathbf{I} & (A + A^\top)/2 \\ (A + A^\top)/2 & \mathbf{I} \end{pmatrix},$$

*such that $M$ is symmetric and positive semi-definite. Then $\det(M) \leq \det(M^S)$.*

*Proof.* Let $M^{\mathrm{up}} = M$ and $M^{\mathrm{down}} = \begin{pmatrix} \mathbf{I} & A^\top \\ A & \mathbf{I} \end{pmatrix}$. Since $M^{\mathrm{up}}$ is symmetric and real-valued, $M^{\mathrm{up}}$ is diagonalizable. Let $x, y \in \mathbb{R}^d$ and $\theta \geq 0$ such that $M^{\mathrm{up}}X = \theta X$ with $X = (x, y)$. Let $Y = (y, x)$. We have $M^{\mathrm{down}}Y = \theta Y$. Hence $M^{\mathrm{down}}$ is symmetric, positive semi-definite and $\det(M^{\mathrm{up}}) = \det(M^{\mathrm{down}})$. Hence using that $M \mapsto \log(\det(M))$ is concave on the space of symmetric positive semi-definite matrices we get that $\det(M^{\mathrm{up}}) \leq \det((M^{\mathrm{up}} + M^{\mathrm{down}})/2) = \det(M^S)$, which concludes the proof. $\square$

**Proposition S40.** *The solution to (S73) exists and $\pi^\star$ is a Gaussian distribution with mean $m \in \mathbb{R}^{2d}$ and covariance matrix $\Sigma \in \mathbb{R}^{2d\times 2d}$ where*

$$m = (-a, a), \qquad \Sigma = \begin{pmatrix} \mathbf{I} & \beta\mathbf{I} \\ \beta\mathbf{I} & \mathbf{I} \end{pmatrix},$$

*where $\beta = (-1 + \sqrt{5})/2$ and $\mathbf{I}$ is the $d$-dimensional identity matrix.*

*Proof.* The fact that $\pi^\star$ exists and is Gaussian is similar to Proposition S37. $\pi^\star$ has mean $m$ since $\pi_i^\star = \nu_i$ for $i \in \{0, 1\}$. Similarly, we have that $\Sigma_{00} = \Sigma_{11} = \mathbf{I}$ since $\pi_i^\star = \nu_i$ for $i \in \{0, 1\}$. We have that $\pi^0$ admits a density $p^0$ with respect to the Lebesgue measure such that for any $x, y \in \mathbb{R}$ we have

$$p^0(x, y) \propto \exp[-(1/2)\{2\|x\|^2 + \|y\|^2 + 2\langle a, x\rangle - 2\langle x, y\rangle + \|a\|^2\}].$$

Hence $\pi^0$ is a Gaussian distribution with mean $m^0$ and covariance matrix $\Sigma^0$ where

$$m^0 = (-a, -a), \qquad \Sigma^0 = \begin{pmatrix} \mathbf{I} & \mathbf{I} \\ \mathbf{I} & 2\mathbf{I} \end{pmatrix}.$$

The Kullback–Leibler divergence between a Gaussian distribution $\pi$, with mean $\tilde{m}$ and covariance matrix $\tilde{\Sigma}$, and $\pi^0$, with mean $m^0$ and covariance $\Sigma^0$ is given by

$$\mathrm{KL}(\pi|\pi^0) = (1/2)\{\log(\det(\Sigma^0)/\det(\tilde{\Sigma})) - d + \mathrm{Tr}((\Sigma^0)^{-1}\tilde{\Sigma}) + (\tilde{m} - m^0)^\top (\Sigma^0)^{-1}(\tilde{m} - m^0)\}.$$

Assume that $\tilde{m} = (-a, a)$ and $\tilde{\Sigma} = \begin{pmatrix} \mathbf{I} & S \\ S^\top & \mathbf{I} \end{pmatrix}$ with $S \in \mathbb{R}^{d \times d}$ such that $\tilde{\Sigma}$ is positive semi-definite . Then we have

$$\mathrm{KL}(\pi|\pi^0) = (1/2)\{-\log(\det(\tilde{\Sigma})) - 2\,\mathrm{Tr}(S) + C\},$$

where $C \geq 0$ is a constant which does not depend on $\Sigma$. In what follows, let $\tilde{\Sigma}' = \begin{pmatrix} \mathbf{I} & (S + S^\top)/2 \\ (S + S^\top)/2 & \mathbf{I} \end{pmatrix}$ and denote $\pi$ the distribution with mean $\tilde{m}$ and covariance matrix $\tilde{\Sigma}'$. Using Lemma S39 we have

$$\mathrm{KL}(\pi'|\pi^0) = (1/2)\{-\log(\det(\tilde{\Sigma}')) - 2\,\mathrm{Tr}(S) + C\}$$
$$\leq (1/2)\{-\log(\det(\tilde{\Sigma})) - 2\,\mathrm{Tr}(S) + C\} = \mathrm{KL}(\pi|\pi^0).$$

Hence, we can assume that $S = S^\top$ and therefore (since $S$ is real-valued), $S$ is diagonalizable. Let $\{\lambda_i\}_{i=1}^d$ the eigenvalues of $S$. Using Schur complements (Petersen et al., 2008, Section 9.1.2) we have

$$\det(\tilde{\Sigma}) = \det(\mathbf{I} - S^2) = \det(\mathbf{I} - S)\det(\mathbf{I} + S) = \prod_{i=1}^d (1 - \lambda_i^2).$$

Therefore we have that for any $\lambda \in (0, 1)$

$$\mathrm{KL}(\pi|\pi^0) = (1/2)\sum_{i=1}^d f(\beta_i) + C, \qquad f(\lambda) = -\log(1 - \lambda^2) - 2\lambda.$$

Hence we get that $\Sigma_{0,1} = \beta \mathbf{I}$ with $\beta = \arg\min_I f$, where $I = (-1, 0) \cup (0, 1)$. We have that $f'(\beta) = 0$ if and only if $\beta = (-1 + \sqrt{5})/2$ or $\beta = -(1 + \sqrt{5})/2$. We conclude the proof using that $\beta \in I$. $\qquad \square$

## S8  Continuous-time Schrödinger bridges

In this section, we prove Proposition 6 in Section S8.1 and draw a link between the potential approach to Schrödinger bridges and DSB in continuous time in Section S8.2.

### S8.1  Proof of Proposition 6

We recall the continuous Schrödinger problem is given by

$$\Pi^\star = \arg\min\{\mathrm{KL}(\Pi|\mathbb{P}) \,:\, \Pi \in \mathscr{P}(\mathcal{C}), \ \Pi_0 = p_{\mathrm{data}}, \ \Pi_T = p_{\mathrm{prior}}\}, \quad T = \sum_{k=0}^{N-1} \gamma_{k+1}. \quad \text{(S74)}$$

In this section, we prove Proposition 6. We start with the following property which can be found in (Léonard, 2014b, Proposition 2.3, Proposition 2.10) and establishes basic properties of dynamic continuous Schrödinger bridges.

**Proposition S41.** *The solution to (S74) exists if and only if the solution to the static Schrödinger bridge exists. In addition, if the solution exists and $\mathbb{P}$ is Markov then the Schrödinger bridge is Markov.*

We now turn to the proof of Proposition 6. First we highlight that $(\Pi^n)_{n \in \mathbb{N}}$ is well-defined since its static counterpart $(\pi_n)_{n \in \mathbb{N}}$ is well-defined using Proposition S26. We only prove that for any $n \in \mathbb{N}$, $(\Pi^{2n+1})^R$ is the path measure associated with the process $(\mathbf{Y}_t^{2n+1})_{t \in [0,T]}$ such that $\mathbf{Y}_0^{2n+1}$ has distribution $p_{\mathrm{prior}}$ and satisfies

$$\mathrm{d}\mathbf{Y}_t^{2n+1} = b_{T-t}^n(\mathbf{X}_t^{2n+1})\mathrm{d}t + \sqrt{2}\mathrm{d}\mathbf{B}_t.$$

The proof for $\Pi^{2n+2}$ is similar. Let $n \in \mathbb{N}$ and assume that $\Pi^{2n}$ is the path measure associated with the process $(\mathbf{X}_t^{2n})_{t \in [0,T]}$ such that $\mathbf{X}_0^{2n}$ has distribution $p_{\text{data}}$ and satisfies

$$\mathrm{d}\mathbf{X}_t^{2n} = f_t^n(\mathbf{X}_t^{2n})\mathrm{d}t + \sqrt{2}\mathrm{d}\mathbf{B}_t.$$

We have that

$$\Pi^{2n+1} = \arg\min\left\{\text{KL}(\Pi|\Pi^{2n}) \, : \, \Pi \in \mathscr{P}(\mathcal{C}), \, \Pi_T = p_{\text{prior}}\right\}.$$

Let $\phi = \text{proj}_T$ such that for any $\omega \in \mathcal{C}$, $\text{proj}_T(\omega) = \omega_T$. Using Proposition S18 we get that for any $\Pi \in \mathscr{P}(\mathcal{C})$ we have

$$\text{KL}(\Pi|\Pi^{2n}) = \text{KL}(\Pi_T|\Pi_T^{2n}) + \int_{\mathbb{R}^d} \text{KL}(\text{K}(x,\cdot)|\text{K}^{2n}(x,\cdot))\mathrm{d}\Pi_T(x),$$

where K and $\text{K}^{2n}$ are the disintegrations of $\Pi$ and $\Pi^{2n}$ with respect to $\phi$. Therefore, we get that $\Pi^{2n+1} = p_{\text{prior}}\text{K}^{2n}$. Since $\text{KL}(\Pi^{2n}|\mathbb{Q}) < +\infty$ and $\Pi^{2n}$ is Markov, Using (Cattiaux et al., 2021, Theorem 4.9) we get that $(\Pi^{2n})^R = \Pi_T\text{K}^{2n}$ satisfies the martingale problem associated with the diffusion

$$\mathrm{d}\mathbf{Y}_t^{2n} = \left\{-f_{T-t}^n(\mathbf{Y}_t^{2n}) + 2\nabla \log p_{T-t}^n(\mathbf{Y}_t^{2n})\right\}\mathrm{d}t + \sqrt{2}\mathrm{d}\mathbf{B}_t. \tag{S75}$$

Since $\Pi^{2n+1} = p_{\text{prior}}\text{K}^{2n}$ we get that $\Pi^{2n+1}$ also satisfies the martingale problem associated with (S75) and is Markov which concludes the proof by recursion.

## S8.2 IPF in continuous time and potentials

First, we recall that the IPF $(\Pi^n)_{n \in \mathbb{N}}$ with $\Pi^0 = \mathbb{P}$ associated with (6) and for any $n \in \mathbb{N}$

$$\Pi^{2n+1} = \arg\min\left\{\text{KL}(\Pi|\Pi^{2n}) \, : \, \Pi \in \mathscr{P}(\mathcal{C}), \, \Pi_T = p_{\text{prior}}\right\},$$
$$\Pi^{2n+2} = \arg\min\left\{\text{KL}(\Pi|\Pi^{2n+1}) \, : \, \Pi \in \mathscr{P}(\mathcal{C}), \, \Pi_0 = p_{\text{data}}\right\}.$$

In this section, we draw a link between our time-reversal approach and the potential approach in continuous time. More precisely, we explicit an identity between the two in Proposition S42.

**Proposition S42.** *Assume* **A**1 *and that there exist* $\mathbb{M} \in \mathscr{P}(\mathcal{C})$, $U \in \text{C}^1(\mathbb{R}^d, \mathbb{R})$, $C \geq 0$ *such that for any* $n \in \mathbb{N}$, $x \in \mathbb{R}^d$, $\text{KL}(\Pi^n|\mathbb{M}) < +\infty$, $\langle x, \nabla U(x)\rangle \geq -C(1 + \|x\|^2)$ *and* $\mathbb{M}$ *is associated with*

$$\mathrm{d}\mathbf{X}_t = -\nabla U(\mathbf{X}_t)\mathrm{d}t + \sqrt{2}\mathrm{d}\mathbf{B}_t,$$

*with* $\mathbf{X}_0$ *distributed according to the invariant distribution of* (14). *For any* $n \in \mathbb{N}$, *let* $\{\varphi_t^{n,\star}, \varphi_t^{n,\circ}\}_{t=0}^T$ *such that for any* $t \in [0,T]$, $\varphi_T^{n,\star} : \mathbb{R}^d \to \mathbb{R}$, $\varphi_0^{n,\circ} : \mathbb{R}^d \to \mathbb{R}$, *for any* $x_0, x_T \in \mathbb{R}^d$

$$\varphi_T^{\star,n}(x_T) = p_{\text{prior}}(x_T)/p_T^n(x_T), \qquad \varphi_0^{\circ,n}(x_0) = p_{\text{data}}(x_0)/p_0^{n+1}(x_0),$$

*and for any* $t \in (0,T)$ *and* $x_t \in \mathbb{R}^d$

$$\varphi_t^{\star,n}(x_t) = \int \varphi_T^{\star,n}(x_T)p_{T|t}^n(x_T|x_t)\mathrm{d}x_T, \qquad \varphi_t^{\circ,n+1}(x) = \int \varphi_0^{\circ,n+1}(x_0)q_{0|t}^n(x_0|x_t)\mathrm{d}x_0.$$

*We have for any* $n \in \mathbb{N}$, $t \in [0,T]$ *and* $x_t \in \mathbb{R}^d$

$$q_t^n(x_t) = p_t^n(x_t)\varphi_t^{\star,n}(x_t), \qquad p_t^{n+1}(x_t) = q_t^n(x_t)\varphi_t^{\circ,n}(x_t). \tag{S76}$$

*In particular, for any* $n \in \mathbb{N}$ *we have*

(a) $(\Pi^{2n+1})^R$ *is associated with* $\mathrm{d}\mathbf{Y}_t^{2n+1} = b_{T-t}^n(\mathbf{Y}_t^{2n+1})\mathrm{d}t + \sqrt{2}\mathrm{d}\mathbf{B}_t$ *with* $\mathbf{Y}_0^{2n+1} \sim p_{\text{prior}}$;

(b) $\Pi^{2n+2}$ *is associated with* $\mathrm{d}\mathbf{X}_t^{2n+2} = f_t^{n+1}(\mathbf{X}_t^{2n+2})\mathrm{d}t + \sqrt{2}\mathrm{d}\mathbf{B}_t$ *with* $\mathbf{X}_0^{2n+2} \sim p_{\text{data}}$;

*with for any* $x \in \mathbb{R}^d$ *and* $t \in (0,T)$

$$f_t^n(x) = f(x) + 2\sum_{k=1}^n \nabla \log \varphi_t^{\star,n}(x), \quad b_t^n(x) = -f(x) + \nabla \log p_t^0(x) + 2\sum_{k=1}^n \nabla \log \varphi_t^{\circ,n}(x). \tag{S77}$$

*Proof.* We only prove that (S76) holds. Then (S77) is a direct consequence of (S76) and Proposition 6. Let $n \in \mathbb{N}$. Similarly to the proof of Proposition S20, there exists $\varphi_T^{\star,n} : \mathbb{R}^d \to \mathbb{R}_+$ such that for any $\{\omega_t\}_{t=0}^T \in \mathcal{C}$ we have

$$(\mathrm{d}\Pi^{2n+1}/\mathrm{d}\Pi^{2n})(\{\omega_t\}_{t=0}^T) = \varphi_T^{\star,n}(\omega_T). \tag{S78}$$

Note that as in Proposition 6, that for any $s, t \in [0, T]$, $\Pi_{s,t}^{2n+1}$ admits a positive density w.r.t the Lebesgue measure denoted $q_{s,t}^n$ and $\Pi_{s,t}^{2n}$ admits a positive density w.r.t the Lebesgue measure denoted $p_{s,t}^n$. Combining this result and (S78), we get that for any $t \in [0, T]$ and $x_t, x_T \in \mathbb{R}^d$ we have

$$q_{t,T}^n(x_t, x_T) = p_{t,T}^n(x_t, x_T)\varphi_T^{\star,n}(x_T).$$

We have that for any $t \in [0, T]$

$$q_t(x_t) = p_t^n(x_t) \int \varphi_T^{\star,n}(x_T) p_{T|t}^n(x_T|x_t) \mathrm{d}x_T = p_t^n(x_t)\varphi_t^{\star,n}(x_t).$$

The proof for that for any $n \in \mathbb{N}$, $t \in [0, T]$ and $x_t \in \mathbb{R}^d$, $p_t^{n+1}(x_t) = q_t^n(x_t)\varphi_t^{\circ,n}(x_t)$, is similar. □

The link between the two formulations is explicit in (S76). Then, (S77) is a straightforward consequence of (S76) and should be compared with Section S8.1. Another proof of Proposition S42 make use of a generalization of (S76) to joint densities and use the fact that for any $n \in \mathbb{N}$, $\Pi^{n+1}$ is a Doob $h$-transform of $\Pi^n$ (see (Rogers and Williams, 2000, Paragraph 39.1) for a definition). Note that this relationship between the potential and the density of the half-bridge is not new. In particular, a similar version of this equation can be found in Bernton et al. (2019). In Finlay et al. (2020), the authors establish a similar relationship in the case of the full Schrödinger bridge.

## S8.3 Likelihood computation for Schrödinger bridges

We provide here details on the likelihood computation of generative models obtained with Schrödinger bridges. Under the conditions of (Léonard, 2011, Theorem 4.12), we define $(\mathbf{X}_t^\star)_{t \in [0,T]}$ the diffusion associated with $\Pi^\star$, see (S74) as well as its time reversal, $(\mathbf{Y}_t^\star)_{t \in [0,T]}$. There exist $f^\star, b^\star : [0, T] \times \mathbb{R}^d \to \mathbb{R}^d$ such that $(\mathbf{X}_t^\star)_{t \in [0,T]}$ and $(\mathbf{Y}_t^\star)_{t \in [0,T]}$ are weak solutions to the following SDEs

$$\mathrm{d}\mathbf{X}_t^\star = f_t^\star(\mathbf{X}_t^\star)\mathrm{d}t + \sqrt{2}\mathrm{d}\mathbf{B}_t, \qquad \mathrm{d}\mathbf{Y}_t^\star = b_{T-t}^\star(\mathbf{Y}_t^\star)\mathrm{d}t + \sqrt{2}\mathrm{d}\mathbf{B}_t.$$

We assume that for any $t \in [0, T]$ there exists $p_t^\star : \mathbb{R}^d \to \mathbb{R}_+$ such that for any $x \in \mathbb{R}^d$, $(\mathrm{d}\Pi_t^\star/\mathrm{d}\lambda)(x) = p_t^\star(x)$. In addition, we assume that $p^\star \in \mathrm{C}^\infty([0, T] \times \mathbb{R}^d, \mathbb{R}_+)$. In this case, we have that $\Pi^\star$ is also associated with the process $(\tilde{\mathbf{X}}_t^\star)_{t \in [0,T]}$ associated with the ODE

$$\mathrm{d}\tilde{\mathbf{X}}_t^\star = \{f_t^\star(\tilde{\mathbf{X}}_t^\star) - 2\nabla \log p_t^\star(\tilde{\mathbf{X}}_t^\star)\}\mathrm{d}t,$$

and $\tilde{\mathbf{X}}_T^\star$ has distribution $p_{\text{prior}}$; see e.g. (Song et al., 2021, Section A). Since $(\mathbf{Y}_t^\star)_{t \in [0,T]}$ is the time-reversal of $(\mathbf{X}_t^\star)_{t \in [0,T]}$ we have that for any $t \in [0, T]$ and $x \in \mathbb{R}^d$

$$b_t^\star(x) = -f_t^\star(x) + 2\nabla \log p_t^\star(x).$$

Therefore, we get that $(\tilde{\mathbf{X}}_t^\star)_{t \in [0,T]}$ is associated with the ODE

$$\mathrm{d}\tilde{\mathbf{X}}_t^\star = -b_t^\star(\tilde{\mathbf{X}}_t^\star)\mathrm{d}t. \tag{S79}$$

Using this result we can compute the log-likelihood of the model using the instantaneous change of variable formula (Chen et al., 2018), see also (Song et al., 2021, Appendix D.2)

$$\log p_{\text{data}}(\tilde{\mathbf{X}}_0^\star) = \log p_{\text{prior}}(\tilde{\mathbf{X}}_T^\star) - \int_0^T \mathrm{div}(b_t^\star)(\tilde{\mathbf{X}}_t^\star)\mathrm{d}t . \tag{S80}$$

As in Song et al. (2021), we can use the Skilling–Hutchinson trace estimator to compute the divergence operator Skilling (1989); Hutchinson (1989). In practice, we discretize the dynamics of $(\tilde{\mathbf{X}}_t^\star)_{t \in [0,T]}$ and use the network $B_{\beta^n}$ obtained with the last iterate of Algorithm 1 and solve the ODE backward in time, recalling that $\tilde{\mathbf{X}}_T^\star$ has distribution $p_{\text{prior}}$. Similarly, we can define

$$\mathrm{d}\tilde{\mathbf{Y}}_t = \{b_{T-t}^\star(\tilde{\mathbf{Y}}_t^\star) - 2\nabla \log p_{T-t}^\star(\tilde{\mathbf{Y}}_t^\star)\}\mathrm{d}t,$$

and $\mathbf{Y}_0^\star$ has distribution $p_{\text{prior}}$. Similarly to (S79), we get that $(\tilde{\mathbf{Y}}_t^\star)_{t \in [0,T]}$ is associated with the ODE

$$\mathrm{d}\tilde{\mathbf{Y}}_t = -f_{T-t}^\star(\tilde{\mathbf{Y}}_t^\star)\mathrm{d}t.$$

Similarly to (S81), we have

$$\log p_{\text{data}}(\tilde{\mathbf{Y}}_T^\star) = \log p_{\text{prior}}(\tilde{\mathbf{Y}}_0^\star) + \int_0^T \mathrm{div}(f_{T-t}^\star)(\tilde{\mathbf{Y}}_t^\star)\mathrm{d}t . \tag{S81}$$

In practice, we discretize the dynamics of $(\tilde{\mathbf{Y}}_t^\star)_{t \in [0,T]}$ and use the network $F_{\alpha^n}$ obtained with the last iterate of Algorithm 1 and solve the ODE forward in time, recalling that $\tilde{\mathbf{Y}}_0^\star$ has distribution $p_{\text{prior}}$. Note that in this case, we solve the ODE forward in time contrary to Durkan and Song (2021).

## S9 Training Techniques

In this section we present some practical guidelines for the implementation of DSB, based on Algorithm 1. We emphasize that, contrarily to previous approaches Song et al. (2021); Song and Ermon (2020); Ho et al. (2020); Dhariwal and Nichol (2021), we do not weight the loss functions as we do not notice any improvement. Let $I \subset \{0, N-1\} \times \{1, M\}$. We define the generalized losses $\hat{\ell}_{n,I}^b$ and $\hat{\ell}_{n,I}^f$ given by

$$\hat{\ell}_{n,I}^b(\beta) = M^{-1} \sum_{(k,j) \in I} \|B_\beta(k+1, X_{k+1}^j) - (X_{k+1}^j + F_k^n(X_{k+1}^j) - F_k^n(X_k^j))\|^2, \quad \text{(S82)}$$

$$\hat{\ell}_{n+1,I}^f(\alpha) = M^{-1} \sum_{(k,j) \in I} \|F_\alpha(k, X_k^j) - (X_k^j + B_{k+1}^n(X_{k+1}^j) - B_{k+1}^n(X_k^j))\|^2. \quad \text{(S83)}$$

We first describe three techniques to compute these losses, then further methods to improve performance.

**Technique 1.** *Simulated Trajectory*

The losses (S82) and (S83) may be computed by simulating diffusion trajectories as described in Algorithm 1. For each sample $j \in \{1, \dots, M\}$ the skeleton of points in the sampled trajectory, $\{X_k^j\}_k$, will be correlated hence only a single uniformly sampled time-step per sample is used to compute the loss per gradient step. In addition, after the initial DSB iteration, simulating the diffusion trajectory involves computationally heavy neural network operations per diffusion step.

**Technique 2.** *Closed Form Sampling*

Since $f_\alpha^0(x) = -\alpha x$, with fixed $\alpha$, it is not necessary to compute full trajectories for the first IPF iteration and one may sample points along the trajectory in closed-form by sampling from a Gaussian distribution with appropriate mean and covariance. This technique also improves the computational speed of the first DSB iteration.

**Technique 3.** *Cached Trajectory*

After the initial DSB iterations it is not possible perform closed form sampling as per Technique 2. Simulating the full diffusion trajectory is both wasteful and expensive as described in Technique 1. In order to obtain a speed-up we consider a cached-version of Algorithm 1 given by Algorithm 2 which entails storing and then resampling diffusion trajectories. Resampled trajectories are then used to compute losses (S82) and (S83). The cache may be refreshed at a certain frequency by once again simulating the diffusion. One may tune the cache-size and refresh frequency to available memory. This modification allows for significant speed-up as the trajectories are not simulated at each training iteration.

**Technique 4.** *Tune Gaussian Prior mean/ variance*

The convergence of the IPF is affected by the mean and covariance matrix of the target Gaussian. In Section S10.1 we investigate possible choices for these values. In practice we recommend to choose the variance of the Gaussian prior $p_{\text{prior}}$ to be slightly larger than the one of the target dataset and to choose the mean of $p_{\text{prior}}$ to be equal to the one of the target dataset. This remark is in accordance with (Song and Ermon, 2020, Technique 1).

**Technique 5.** *Network Refinement / Fine Tuning*

Training large networks from scratch, per DSB iteration, is very expensive. However, from (S49)-(S50),

$$b_{k+1}^n(x) = b_{k+1}^{n-1}(x) + 2\nabla \log p_{k+1}^n(x) - 2\nabla \log q_k^{n-1}(x),$$
$$f_k^n(x) = f_k^{n-1}(x) + 2\nabla \log q_k^{n-1}(x) - 2\nabla \log p_{k+1}^{n-1}(x).$$

One may therefore initialize networks at DBS iteration $n$ from $n-1$ in order to reduce training time. In future work, we plan to investigate more sophisticated warm-start approaches through meta-learning.

**Technique 6.** *Exponential Moving Average*

Similar to (Song and Ermon, 2020, Technique 5), we found taking the exponential moving average of network parameters across training iterations, with rate 0.999, improved performance.

**Algorithm 2** Cached Diffusion Schrödinger Bridge

1: **for** $n \in \{0, \ldots, L\}$ **do**
2:    **while** not converged **do**
3:       Sample and store $\{X_k^j\}_{k,j=0}^{N,M}$ where $X_0^j \sim p_{\text{data}}$ and
        $X_{k+1}^j = X_k^j + \gamma_{k+1} f_{\alpha^n}(k, X_k^j) + \sqrt{2\gamma_{k+1}} Z_{k+1}^j$
4:       **while** not refreshed **do**
5:          Sample $I$ (uniform in $\{0, N-1\} \times \{1, M\}$)
6:          Compute $\hat{\ell}_{n,I}^b(\beta^n)$ using (S82)
7:          $\beta^n = \text{Gradient Step}(\hat{\ell}_{n,I}^b(\beta^n))$
8:       **end while**
9:    **end while**
10:   **while** not converged **do**
11:      Sample $\{X_k^j\}_{k,j=0}^{N,M}$, where $X_N^j \sim p_{\text{prior}}$, and
        $X_k^j = X_{k+1}^j + \gamma_k b_{\beta^n}(k, X_k^j) + \sqrt{2\gamma_{k+1}} Z_k^j$
12:      **while** not refreshed **do**
13:        Sample $I$ (uniform in $\{0, N-1\} \times \{1, M\}$)
14:        Compute $\hat{\ell}_{n+1,I}^f(\alpha^{n+1})$ using (S83)
15:        $\alpha^{n+1} = \text{Gradient Step}(\hat{\ell}_{n+1,I}^f(\alpha^{n+1}))$
16:      **end while**
17:   **end while**
18: **end for**
19: **Output:** $(\alpha^{L+1}, \beta^L)$

## S10    Additional Experimental Results and Details

We provide additional examples for the two-dimensional setting in Section S10.1. We then turn to higher dimensional generative modeling in Section S10.2. Finally, we detail our dataset interpolation experiments in Section S10.3. Code is available here: .

### S10.1   Two-dimensional experiments

In the case of two-dimensional distributions we use a simple architecture for the networks $f_\alpha$ and $b_\beta$, see Figure S2. We use the variational formulation Section S5.2.2 because our network architecture does not have a residual structure. To optimize our networks we use ADAM Kingma and Ba (2014) with momentum $0.9$ and learning rate $10^{-4}$.

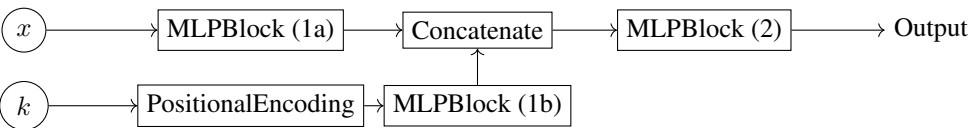

Figure S2: Architecture of the networks used in the two-dimensional setting. Each MLP Block is a Multilayer perceptron network. The "PositionalEncoding" block applies the sine transform described in Vaswani et al. (2017). MLPBlock (1a) has shape $(2, 16, 32)$, MLPBlock (1b) has shape $(1, 16, 32)$ and MLPBlock has shape $(64, 128, 128, 2)$. The total number of parameters is 26498.

In all two-dimensional experiments we fix $\gamma_k = 10^{-2}$ and use a batch size of $512$. The mean and variance of $p_{\text{prior}}$ are matched to those of $p_{\text{data}}$. The cache contains $10^4$ samples and is refreshed every $10^3$ iterations. We train each DSB step for $10^4$ iterations. All two-dimensional experiments are run on Intel(R) Core(TM) i7-10850H CPU @ 2.70GHz CPUs.

In Figure S3 we present additional two-dimensional experiments.

We found that the variance of $p_{\text{prior}}$ has an impact on the convergence speed of DSB, see Figure S4 for an illustration. This remark is in accordance with (Song and Ermon, 2020, Technique 1). In practice

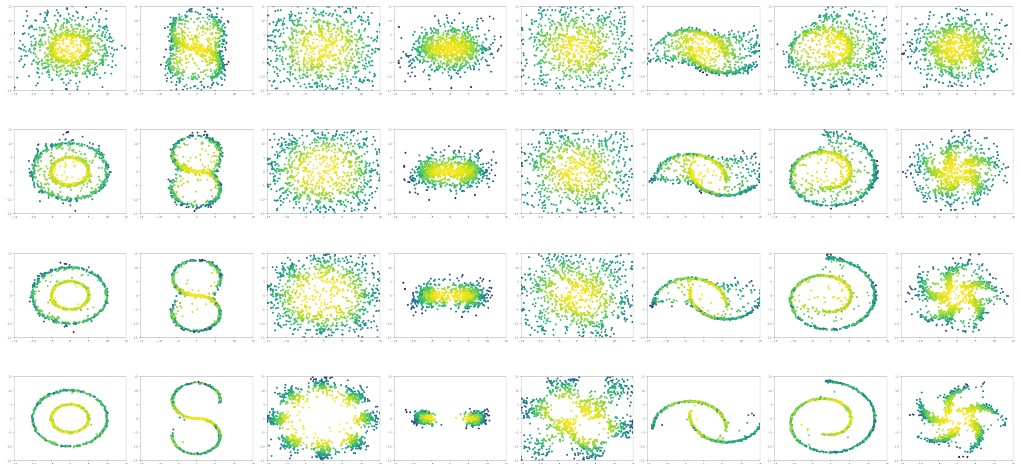

Figure S3: The first row corresponds to iteration 1 of DSB, the second to iteration 3 of DSB, the third to iteration 5 of DSB and the last to iteration 20 of DSB.

we recommend to set the variance to be larger than the variance of the target dataset, see Technique 4 in Section S9.

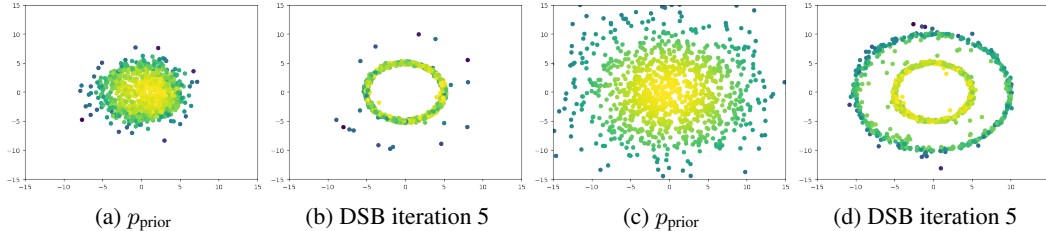

(a) $p_{\text{prior}}$         (b) DSB iteration 5         (c) $p_{\text{prior}}$         (d) DSB iteration 5

Figure S4: Effect of the variance of $p_{\text{prior}}$ on the convergence of DSB. If $p_{\text{prior}}$ has a small variance $\sigma^2$ (here $\sigma^2 = 5$ in (a) and (b)) then DSB converges more slowly. If $\sigma^2 \approx \sigma^2_{\text{data}}$, where $\sigma^2_{\text{data}}$ is the variance of $p_{\text{data}}$ then we observe more diversity in the samples obtained using DSB even for few iterations.

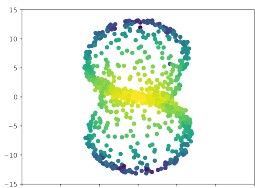

Figure S5: Failure of DSB for low $N$. DSB iteration 3 with $N = 2$ and $30,000$ training steps per DSB iteration. The results deteriorate significantly after 5 iterations of the algorithm.

Finally, since DSB does not require the number of Langevin iterations $N$ to be large, one may question why not use $N = 1$ in order to derive a feed-forward generative model. In practice this choice of $N$ is not desirable for two reasons. (a) Firstly, since $p_N$ is not a good approximation of $p_{\text{prior}}$, theoretical results such as (Léger, 2020, Corollary 1) indicates that more IPF iterations are needed. (b) Second, in our experiments we observe that in order to obtain similar results to $N = 10$ with $N = 1$ we need to substantially increase the size of the networks, even for a large number of IPF iterations, see Figure S5.

## S10.2 Generative Modeling

**Implementation details**  We use a reduced version of the U-net architecture from Nichol and Dhariwal (2021) for $F_\alpha$ and $B_\beta$, where we set the number of channels to 64 rather than 128 for computational resource purposes. We tried the architecture of Song and Ermon (2020), however we observed worse results in our framework. Although we observed improvement using the corrector scheme of Song et al. (2021), this improvement was similar to augmenting the number of steps in the Langevin scheme. We therefore chose to avoid using such techniques altogether because of the increase in computing time when sampling, often by doubling the number of passes through the network.

We chose the sequence $\{\gamma_k\}_{k=0}^N$ to be invariant by time reversal, *i.e.* for any $k \in \{0, \ldots, N\}$, $\gamma_k = \gamma_{N-k}$. In practice, we assume that $N$ is even and let $\gamma_k = \gamma_0 + (2k/N)(\bar{\gamma} - \gamma_0)$ for $k \in \{0, \ldots, N/2\}$ with $\gamma_0 = 10^{-5}$ and $\bar{\gamma} = 10^{-1}$. The rest of the sequence is obtained by symmetry.

In the case of the MNIST dataset (dimension $d = 28 \times 28 = 784$) we set the batch size to 128, the number of samples in the cache to $5 \times 10^4$ with 10 time-points sampled from each trajectory for each sample of $p_{\text{data}}$. We end up with an effective cache of size $5 \times 10^5$. The cache is refreshed each $10^3$ iterations and the networks are trained for $5 \times 10^3$ iterations. Again we use the ADAM optimizer with momentum 0.9 and learning rate $10^{-4}$. $p_{\text{prior}}$ is a Gaussian density with zero mean and identity covariance matrix. We have presented results for varying number of diffusion steps, $N$.

In the case of the CelebA dataset (dimension $d = 32 \times 32 \times 3 = 3072$) we set the batch size to 256, number of steps $N = 50$, the number of samples in the cache to 250 with 1 time-point sampled from each trajectory for each sample of $p_{\text{data}}$. The cache is refreshed each $10^2$ iterations and the networks are trained for $5 \times 10^3$ iterations. Again we use the ADAM optimizer with momentum 0.9 and learning rate $10^{-4}$. $p_{\text{prior}}$ is a Gaussian density with zero mean and identity covariance matrix.

Our results on MNIST and CelebA are computed using up to 4 NVIDIA Tesla V100 from the Google Cloud Platform.

**Additional examples**  In this section we present additional examples for our high-dimensional generative modeling experiments. In Figure S6 we perform interpolation in the latent space. More precisely we let $X_N^0$ and $X_N^1$ be two samples from $p_{\text{prior}}$. We then compute $X_N^\lambda = (1-\lambda)X_N^0 + \lambda X_N^1$ for different values of $\lambda \in [0, 1]$. For each value of $\lambda \in [0, 1]$ we associate $X_0^\lambda$ which corresponds to the output sample obtained using the generative model given by DSB with final condition $X_N^\lambda$. Note that in order to obtain a deterministic embedding we fix the Gaussian random variables used in the sampling. One could also have used the deterministic embedding used by Song et al. (2021), *i.e.* a neural ordinary differential equation that admits the same marginals as the diffusion thus enabling exact likelihood computation, see Section S8.3 for details.

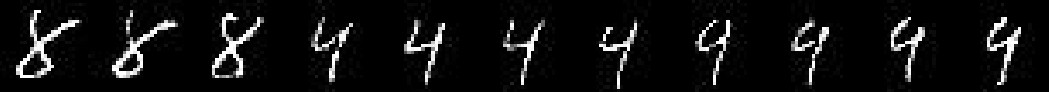

Figure S6: Interpolation in the latent space for MNIST.

In Figure S7 we present high quality samples for MNIST. In order to obtain these high quality samples we consider our baseline MNIST configuration but instead of choosing $N = 10$ time steps we consider $N = 30$. In addition, we train the networks for $15 \times 10^3$ iterations instead of $5 \times 10^3$. The number of samples in the cache is $M = 500$

In Figure S8 we present a temperature scaling exploration of the embedding obtained for CelebA. Similarly to the interpolation experiment we fix the Gaussian random variables in order to obtain a deterministic mapping from the latent space to the image space.

In Figure S9 we explore the latent space of our embedding of CelebA. To do so, we obtain samples using a Ornstein-Ulhenbeck process targeting $p_{\text{prior}}$. We refer to our project page project webpage for an animated version of this latent space exploration.

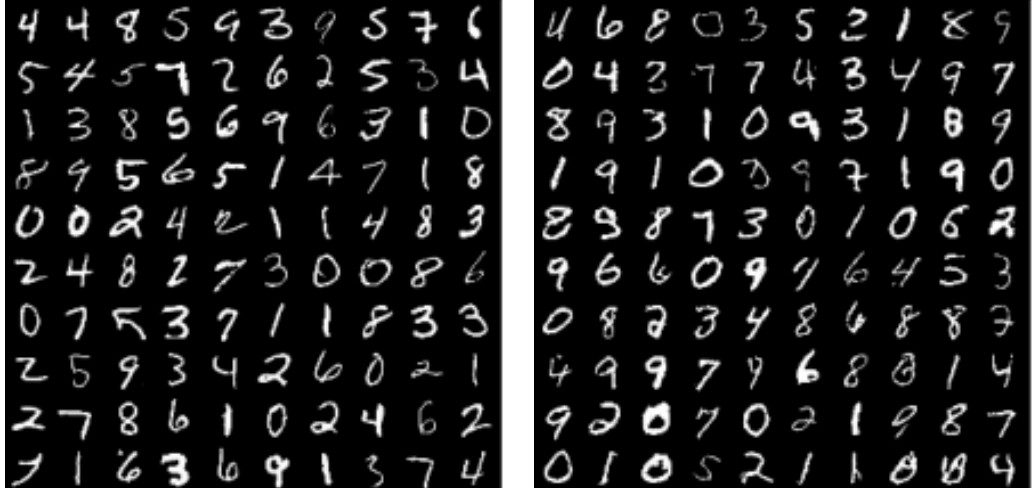

Figure S7: MNIST samples: original dataset (left) and generated MNIST samples (right) after 12 DSB iterations

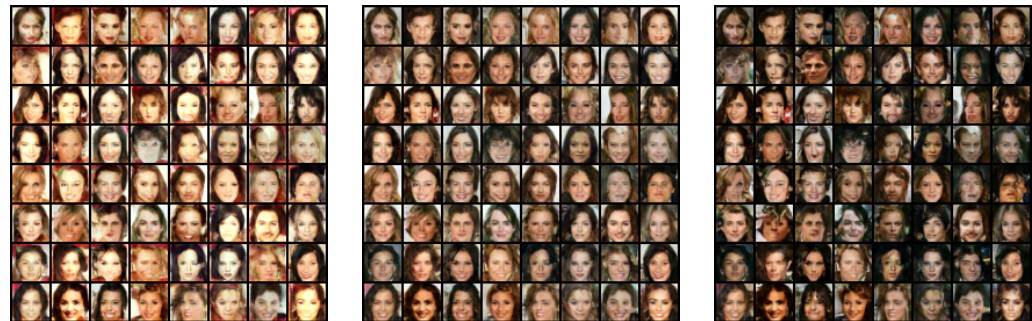

Figure S8: Temperature scaling in the latent space.

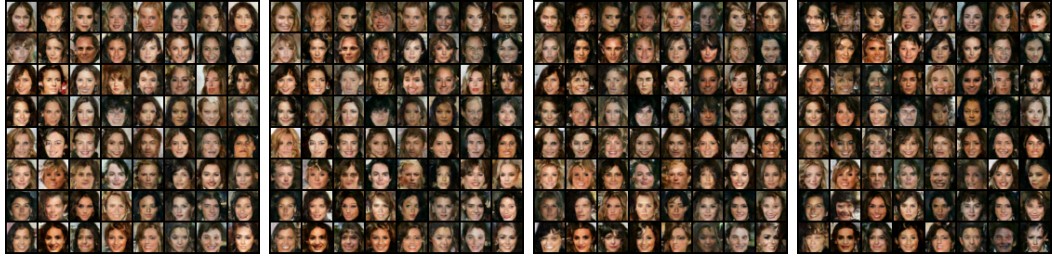

Figure S9: Exploration of the latent space. Samples are generated using a Ornstein-Ulhenbeck process targeting $p_{\text{prior}}$ to obtain the initial condition then using the generative model given by DSB. From left to right to right: samples at time $t = 0, 1.3, 3.6, 8.6$.

## S10.3 Dataset interpolation

For the dataset interpolation task we keep the same parameters and architecture as before except that the number of Langevin steps is increased to 50 steps in the two-dimensional examples and to 30 steps in the EMNIST/MNIST interpolation task. We also change the reference dynamics which is chosen to be the one obtained with the DSB where $p_{\mathrm{prior}}$ is a Gaussian. This choice allows us to speed up the training of DSB in this setting. Animated plots are available at project webpage.

**EMNIST/MNIST**    In order to perform translation between the dataset of handwritten letters (EMNIST) and handwritten digits (MNIST) we reduce EMNIST to 5 letters so that it contains as many classes as MNIST (we distinguish upper-case and lower-case letters), see Cohen et al. (2017) for the original dataset.



Figure S10: Iteration 10 of the IPF with $T = 1.5$ (30 diffusions steps). From left to right: $t = 0, 0.4, 1.25, 1.5$.

**Two dimensional examples**    We present interpolation for a number of classical two-dimensional datasets.

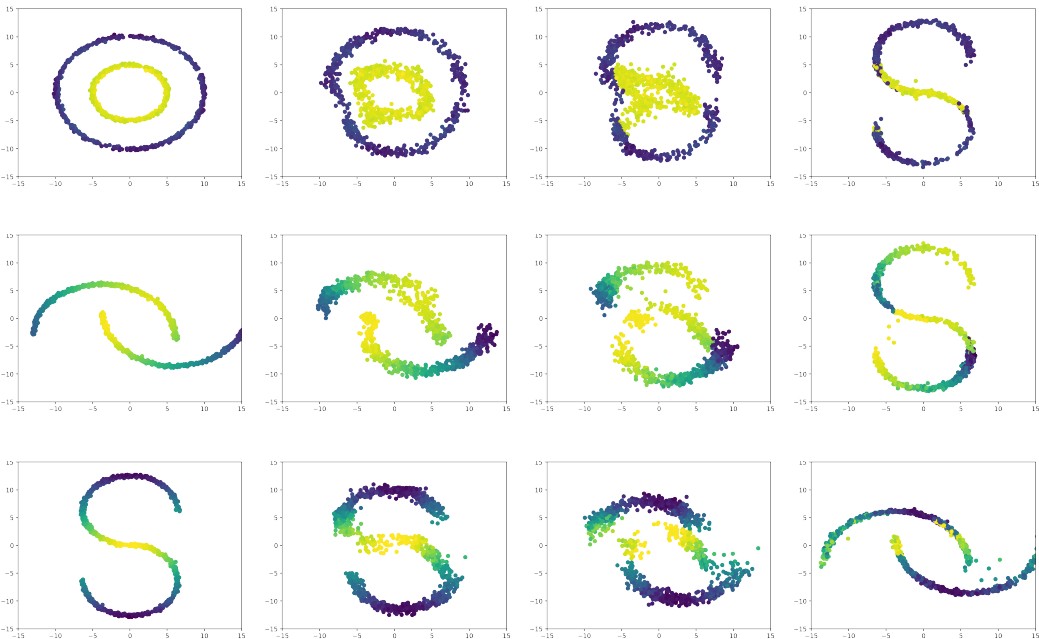

Figure S11: Dataset interpolation (DSB iteration 9). From left to right: $t = 0, 0.15, 0.30, 0.5$.