# OpenReview forum: "Diffusion Schrödinger Bridge with Applications to Score-Based Generative Modeling"
_NeurIPS.cc/2021/Conference — NeurIPS 2021 Spotlight_

### Official Review · Reviewer_TxZC · 2021-07-13

**Rating:** 8
**Confidence:** 3

**Summary:**

The paper introduces a new method to iteratively refine a diffusion-based generative model, that is motivated by the so-called Schrödinger-bridge. The method amounts to iteratively train a forward and a backward model, in a manner that is simpler than previous work.

**Ethical Concerns:**

nothing particular

**Limitations And Societal Impact:**

I don't think there are any particular societal impact for this work that would require special treatment.

Regarding the limitations, it is true that it could be interesting to have their opinion on what are actually the main weaknesses of their method to know which parts could benefit from further research.

**Main Review:**

This paper is definitely a great read, although it is a bit hard to follow at some crucial parts. The contribution is very interesting and will definitely prove very inspiring to many researchers, I recommend acceptance.

Here are my comments on the fly:


* Figure 1 is not really informative and should be extended to better represent what happens at each step
* I guess approximation (4) is classical ? It would be better to write "The following approximation is usually made" instead of using "we", that would make it clearer it's not a contribution
* is the other formulation for \theta^\star that uses all the \log p_{k|k-1}(X_k|X_{k-1}) instead of p_{k|0} equivalent and also principled theoretically, or is it some kind of approximation ?
* I think that your short discussion after th.1 is very interesting, but it's hard for me to understand why the second term for the bound is related to the increase of \alpha. Could you please be a bit more precise on that point ?

* I don't think your notations on the "static Schrödinger bridge problem" are easy to follow. What is \mu_{0,N} exactly ? Is it the restriction of \mu on 0 and N ? I guess that's the case, but the notation is a bit confusing. Maybe writing something like \mu_{0:N}=\mu_{0,N}\mu_{1:N-1\mid0,N} or something in this vein would be a bit clearer

* As it is worded, I don't understand whether the result from Mikami holds only in the case of the gaussian transition probability P_{k+1\mid k}.

* From the point at which you introduced the SB problem, I think you should provide some wordy explanation regarding what you have and what you want to achieve. If I understand correctly, the objective is to progressively refine the transportation through iteratively apply this static SB problem ? There could be a figure about this giving the big picture, or something of the like. I must tell that the paper starts to be a bit hard to follow here.

* The proposition 2 appears as very nice. However, I would once again gladly use a support figure to make the ideas clearer, that could very advantageously replace your current figure 1 that is not informative at all.

* Proposition 3 seems like the central part of the paper, that holds the core contribution. However, I don't manage to understand where equations (12)-(13) come from, they are not sufficiently explained in my opinion, especially considering the space you take to explain other less important things. Please take the time to better explain the reason for these.

* regarding figure 1 and proposition 3: it would be way more effective to have a figure where your actual networks appear (the B(k,x) and F(k,x)).

* Is it important to train \beta^0 using previous SGM methods, or would you propose an alternative easier way ?

* Reading the text, it looks like we'll have to store all the N networks, but actually it looks like only the last one may be stored, so that we actually discard the other ones on the way, right ?







**Time Spent Reviewing:**

3

---

> ### Author Response · Authors · 2021-08-09
> **Response to Reviewer TxZC**
>
> Thank you for your review and constructive feedback. We are glad that you appreciate the paper.
>
> ---------------------------
>
> - **“Figure 1 is not really informative and should be extended to better represent what happens at each step”**
> - **“regarding figure 1 and proposition 3: it would be way more effective to have a figure where your actual networks appear (the B(k,x) and F(k,x)).”**
> - **“The proposition 2 appears as very nice. However, I would once again gladly use a support figure to make the ideas clearer, that could very advantageously replace your current figure 1 that is not informative at all.”**
>
> We will incorporate your feedback on Figure 1 by adding the SDEs equations with network approximations from Proposition 3, and also explaining it in more detail in relation to Proposition 2. It may be clearer with a single image rather than a distribution of 2D data points, as was done in similar diagrams in previous works such as Figure 1 of Song 2021 [1].
>
> [1] Song, Yang, et al. "Score-Based Generative Modeling through Stochastic Differential Equations." ICLR. 2020.
>
> ---------------------------
>
> **“From the point at which you introduced the SB problem, I think you should provide some wordy explanation regarding what you have and what you want to achieve. If I understand correctly, the objective is to progressively refine the transportation through iteratively apply this static SB problem ? There could be a figure about this giving the big picture, or something of the like.”**
>
> We will strive to provide more high-level insights on our methodology to help the reader. We believe the new and improved version of Figure 1 will help achieve this goal.
>
> ---------------------------
>
> **“Proposition 3 seems like the central part of the paper, that holds the core contribution. However, I don't manage to understand where equations (12)-(13) come from, they are not sufficiently explained in my opinion, especially considering the space you take to explain other less important things. Please take the time to better explain the reason for these.”**
>
> We will provide a sketch of the proof of Proposition 3 in the main text. The detailed proof can be found in Section S5.1. In particular, as we are approximating the drift/mean of the diffusions rather than the score, the objective loss is slightly different to the score-matching objective.
>
> ---------------------------
>
> **“is the other formulation for \theta^\star that uses all the \log p_{k|k-1}(X_k|X_{k-1}) instead of p_{k|0} equivalent and also principled theoretically, or is it some kind of approximation ?” (line 102-104)**
>
> Thanks for pointing this out. We will add a discussion on the differences between these two methods.
> Both approaches are theoretically principled. However, they are not equivalent and yield different score approximations as neural networks represent only a subset of $L^2$ functions.
>
> ---------------------------
>
> **“I think that your short discussion after th.1 is very interesting, but it's hard for me to understand why the second term for the bound is related to the increase of $\alpha$. Could you please be a bit more precise on that point ?”**
>
> We will clarify this discussion in the revised version of our paper. The second term in Theorem 1 corresponds to the discretization error of the forward dynamics. This discretization error term depends on the Lipschitz constant of the drift of the forward dynamics which is related to $\alpha$. The explicit dependence of this term w.r.t $\alpha$ is given in the appendix (see Equations S24 and S32).
>
> ---------------------------
>
> **“As it is worded, I don't understand whether the result from Mikami holds only in the case of the gaussian transition probability $P_{k+1\mid k}$.”**
>
> Mikami’s approximation only holds in the case of a Brownian forward dynamics, i.e. a Gaussian transition probability $p_{k+1|k}$ with mean $x_k$. We conjecture that the results of Mikami can be extended to general Ornstein-Ulhenbeck processes yielding Wasserstein distances for other ground costs of the form $c(x,y) = \| x - \eta y \|^2$ with $\eta > 0$.
>
> ---------------------------
>
> **“Is it important to train \beta^0 using previous SGM methods, or would you propose an alternative easier way ?”**
>
> The first reverse diffusion problem is exactly the same problem addressed by existing SGM methods. We do not have an alternative, easier way to solve the first iteration of the IPF.
>
> ---------------------------
> **“Reading the text, it looks like we'll have to store all the N networks, but actually it looks like only the last one may be stored, so that we actually discard the other ones on the way, right ?”**
>
> You are correct, we discard all networks except the last ones. We will clarify that only the final networks are required, in particular the final backward network for the generative model, and the final forward and backward pair for optimal transport purposes.
>
> ---------------------------
>
> **“I don't think your notations on the "static Schrödinger bridge problem" are easy to follow. What is \mu_{0,N} exactly ? Is it the restriction of \mu on 0 and N ? I guess that's the case, but the notation is a bit confusing. Maybe writing something like \mu_{0:N}=\mu_{0,N}\mu_{1:N-1\mid0,N} or something in this vein would be a bit clearer”**
>
> Thank you for pointing this out. We will aim to make the notation clearer in our revision.
>
> ---------------------------
>
> **“I guess approximation (4) is classical ? It would be better to write "The following approximation is usually made" instead of using "we", that would make it clearer it's not a contribution”**
>
> You are correct. This is a known approximation and we will make this point clear as you suggested.

---

### Official Review · Reviewer_hQmL · 2021-07-15

**Rating:** 6
**Confidence:** 5

**Summary:**

The authors present a generative model based on the Iterative Proportional Fitting of the Schrödinger Bridge problem. They provide convergence analysis of their proposed algorithm, and show the performance of their algorithm with image generation and datasets interpolation.

**Ethical Concerns:**

No.

**Main Review:**

# Novelty:
This paper introduces a novel score-based generative model based on the Iterative Proportional Fitting of the Schrödinger Bridge problem. The proposed algorithm can be used as optimal transport between any distributions, which is also new, compared to existing score-based models.

# Major concern:
The originality claimed in the paper is overestimated, and related works are not properly cited. In line 314, the authors said they have provided the first convergence result for SGM methods. As far as I know, the convergence of SGM has been studied by Block et al. 2020 and Wang et al. 2021. Also in line 283, the authors claim that their method DSB “is the first practical algorithm for approximating the solution to the SB problem in high dimension”. They should clearly discuss the connection and difference between their proposal and the method in Wang et al. 2021 which also utilizes SB for generative modeling.

Overall, the paper is well-written and theoretically sound, and the authors show the image generation and interpolation results on image datasets and the reproducibility is good. This work has worthy contribution to the SGM methods.

Reference:
Block A, Mroueh Y, Rakhlin A. Generative modeling with denoising auto-encoders and Langevin sampling[J]. arXiv preprint arXiv:2002.00107, 2020.
Wang G, Jiao Y, Xu Q, et al. Deep Generative Learning via Schr\"{o} dinger Bridge[J]. In ICML 2021, arXiv preprint arXiv:2106.10410, 2021.



**Time Spent Reviewing:**

16

---

> ### Author Response · Authors · 2021-08-09
> **Response to Reviewer hQmL**
>
>
> Thank you for your comments, we appreciate your acknowledgements of the paper’s merits.
>
> ---------------------------
>
> **“The originality claimed in the paper is overestimated, and related works are not properly cited. In line 314, the authors said they have provided the first convergence result for SGM methods. As far as I know, the convergence of SGM has been studied by Block et al. 2020.”**
>
> Thank you for pointing out the Block et al. 2020 reference; we were not aware of it at the time of writing. It has been added to our paper and we will position our results appropriately.
>
> Please note that our results are significantly different. Block et al., 2020 analyzed a time-homogeneous process based on score-matching that approximates a Langevin diffusion. However, all state-of-the-art score-based generative models require time-inhomogeneous scores (Song and Ermon, 2019 [1]; Ho et al., 2020 [2]; Song et al., 2021 [3], Dhariwal [4] etc.) and are based on finite-time time-inhomogeneous reverse diffusions. This is what is analyzed in our Theorem 1.
>
> [1] Song, Yang, and Stefano Ermon. "Generative Modeling by Estimating Gradients of the Data Distribution." NeurIPS 2019. \
> [2] Ho, Jonathan, Ajay Jain, and Pieter Abbeel. "Denoising Diffusion Probabilistic Models." NeurIPS. 2020. \
> [3] Song, Yang, et al. "Score-Based Generative Modeling through Stochastic Differential Equations." ICLR. 2020. \
> [4] Dhariwal, Prafulla, and Alex Nichol. "Diffusion models beat gans on image synthesis." arXiv preprint arXiv:2105.05233 (2021).
>
> ---------------------------
>
> **“Also in line 283, the authors claim that their method DSB “is the first practical algorithm for approximating the solution to the SB problem in high dimension”. They should clearly discuss the connection and difference between their proposal and the method in Wang et al. 2021 which also utilizes SB for generative modeling."**
>
> We will include a discussion of the interesting paper by Wang et al. (ICML 2021). This was not possible at the time of submission of our paper as the paper of Wang et al. was not publicly available then. Specifically, the NeurIPS paper submission deadline was May 28th 2021; the paper by Wang et al., 2021 was first posted on arXiv on June 19th 2021 and the ICML camera-ready deadline was June 10th 2021. Hence the ICML proceedings appeared only after.
>
> While the method of Wang et al., 2021 also relies on SB ideas, our methodology is significantly different. In particular, the method of Wang et al. does not attempt to solve the SB problem between data and prior distributions but focuses on the task of generative modelling. It relies on a SB problem from a Dirac mass at the origin to a smoothed version of the data distribution, followed by another SB problem which corresponds to a denoising stage from this smoothed distribution (adding Gaussian noise) to the data distribution of interest. Their algorithm makes use of score-matching techniques (as existing state-of-the-art score generative models) and also requires another neural network to perform density ratio estimation.
>
> Our approach addresses the SB problem between a continuous prior and the data distribution directly, and hence may be viewed as an entropy-regularized approximation of optimal transport. In addition, our framework offers the additional flexibility of employing any prior distribution to perform tasks such as dataset interpolation (see Figure 6) and can be used more generally to solve a variety of optimal transport tasks in addition to generative modeling.

---

> > ### Comment · Reviewer_hQmL · 2021-08-23
> > **Thanks for the authors reply.**
> >
> > Thanks for the authors reply.  My concerns are eliminated.

---

### Official Review · Reviewer_QQ6H · 2021-07-15

**Rating:** 8
**Confidence:** 4

**Summary:**

Leveraging recent advance in score-based generative modeling with SDEs, this paper proposes to solve high dimensional Schrodinger bridge problems by iteratively training score-based models. The method is the first to effectively solve Schrodinger bridge problems in high dimensional spaces, with demonstrations on distributions of real-world images, and can be viewed as a generalization to the previous SDE approach for score-based generative modeling.

**Limitations And Societal Impact:**

No obvious concerns.

**Main Review:**

Score-based generative models/ diffusion models have attracted increasing attention for theorists and practitioners in machine learning. This work shows that the basic ideas of them can be generalized to solve high dimensional Schrodinger bridge problems, all while enabling new capabilities and applications. Here are several important advantages of this submission:

1. Clear writing. Although fully understand everything in this submission definitely requires some background on stochastic analysis, authors have done a good job on conveying the intuition and basic concepts, which is not an easy task. The discussion on discrete cases makes it much easier for readers to intuitively comprehend the continuous case (Schrodinger bridges are defined as a continuous case) without much background.

2. Solid theoretical analysis. Theorem 1 bounds the deviation of distributions with imperfect score estimations, and provides one of the first proofs of the convergence of SGMs. Section 3.4 proves the convergence of iterative proportional fitting in non-compact settings. Both are very solid contributions.

3. The proposed framework generalizes score-based generative modeling with SDEs. The prior distribution is no longer required to be tractable, as long as samples can be obtained. The training procedure of prior work can be viewed as one step of iterative proportional fitting. As a result, the proposed method can be leveraged to fine-tune existing score-based models for better matching the prior, and allows additional applications like dataset interpolation.

Below are some drawbacks and questions for the authors:

1. Shorter time intervals doesn't necessarily mean smaller number of iterations for sampling. The time scales of SDEs can be rescaled arbitrarily, so we can have arbitrarily small time horizon without having any practical difference. From this perspective, I'm not convinced that Schrodinger bridges can speed up sampling of score-based models in the general case.

2. In section 3.3, authors claim that using score matching to approximate $\\{ \nabla \log p_{k+1}^i (x) \\}_{i=0}^n$ is prohibitively costly. However, if I understand it correctly, the mean-matching method proposed in Proposition 3 is **exactly** the same as using denoising score matching to estimate the score functions individually.

3. Experiments are relatively weak as results are not competitive with previous score-based models on SDEs, and image resolutions are small.

**Time Spent Reviewing:**

3

---

> ### Author Response · Authors · 2021-08-09
> **Response to Reviewer QQ6H**
>
> Thank you for your comments and thoughtful questions, we are glad you enjoyed the paper.
>
> ---------------------------
>
> **“Shorter time intervals doesn't necessarily mean smaller number of iterations for sampling. The time scales of SDEs can be rescaled arbitrarily, so we can have arbitrarily small time horizon without having any practical difference. From this perspective, I'm not convinced that Schrodinger bridges can speed up sampling of score-based models in the general case.”**
>
> We agree with the reviewer that one could rescale time and step-size. However, depending on the diffusion, a large step-size and hence a small number of diffusion steps may result in a large discretization error. We will be more careful in our writing in order to clarify our intended meaning.
>
> The key point we will clarify is that we do not require the initial forward SDE to converge to a distribution that is close to the prior distribution, in contrast to existing score-based methods. The IPF procedure iteratively improves the forward and backward dynamics until we obtain a pair of forward and backward SDEs that can reach the given data and prior distributions in any pre-specified number of steps (or equivalently “diffusion time”).
>
> ---------------------------
>
> **“In section 3.3, authors claim that using score matching to approximate
>  is prohibitively costly. However, if I understand it correctly, the mean-matching method proposed in Proposition 3 is exactly the same as using denoising score matching to estimate the score functions individually.”**
>
>
> We will clarify our writing in view of this comment. Our drift/mean matching procedure is indeed similar to score-matching in that it involves simulating a forward SDE and learning a single network for the backward SDE. However, we do not estimate the score at each IPF iteration but the overall drift, for the following reasons.
>
> As we iteratively update the forward and backward SDEs for N IPF iterations, we would require 2N neural networks if we were to approximate each score and hence approximate the drift of the final backward diffusion (see S5.2.1 and S5.2.3 for a discussion). We will revise and make this remark explicit in the main document.
>
> Score networks are typically very large, often with >10^8 parameters. Storing many of these requires significant memory resources. Similarly, evaluating this many large networks at each training iteration to learn the next network requires significant compute cost.
>
> Instead we approximate the overall drift/mean of each diffusion which requires only a single neural network (see S5.2.2).
>
> ---------------------------
>
> **“Experiments are relatively weak as results are not competitive with previous score-based models on SDEs, and image resolutions are small.”**
>
>
> We agree with the reviewer and this will be a focus going forward.
>
> Due to limited shared resources, we used significantly less compute when compared to current state-of-the-art results.
>
> With sufficient computational resources for hyperparameter tuning, we believe our proposed method can obtain competitive results using very few diffusion steps.

---

> > ### Comment · Reviewer_QQ6H · 2021-08-23
> > **Thanks**
> >
> > Thank you for the clarification. I think this is a great paper!

---

### Official Review · Reviewer_dJt4 · 2021-07-19

**Rating:** 8
**Confidence:** 4

**Summary:**

The paper proposes a novel training method to solve Schrödinger bridge (SB) problems by performing (approximate) iterative proportional fittings (IPFs).
- SB problems is to find a diffusion process defined in a finite time range $t \\in [0, T]$ for given marginals at the two ends in time $t = 0, t=T$. Specifically, the paper focuses on cases when diffusion processes have state-independent and time-homogeneous diffusion terms. Note that SB problems can have variant conditions; for example, one gives a reference diffusion instead of the two marginals.
- Classical IPFs solve SB problems when a reference diffusion is provided instead of the marginals at two time ends. IPFs iteratively solve optimization problems. With the given initial distribution at one end, each optimization problem finds a diffusion in one time direction by minimizing the reverse KL between it and a target. Here, the target is the previous step's solution (a diffusion runs reversed in time). Each IPF step alternates the time direction by setting initial distributions correspondingly. The first target is set to the reference distribution of the SB problem. For some limited cases, it has been proven that IPFs solve SB problems by repeating the optimizations.

In order to motivate the proposed method, the paper first shows that for discrete-time IPFs, the optimal solution $q_{k|k+1}^{n}$ of $n$-th IPF iteration (of one time direction) is the posterior $p_{k|k+1}^{n}$ of the previous step's solution $p_{k+1|k}^{n}$: the result also holds for reversed direction. Instead of minimizing the reverse KL on the entire path in time, the authors emphasize that one can minimize per-time-step loss between a current estimate and the previous optimal.

The paper then shows that there are two properties for discrete-time IPFs, when diffusion terms are state-independent and time-homogeneous diffusions. First, the optimal solution $q_{k|k+1}^{n}$ per each time step must follows the well-known reserve diffusion formula, which includes the target's $k$+$1$-th marginal score function $\nabla \log p_{k+1}^{n}$ (Song et al, 2021). Next, instead of using score functions, the optimal $q_{k|k+1}^{n}$'s mean can be written as the conditional expected value of $X_{k+1} + F_{k}^n(X_k) - F_{k}^n(X_{k+1})$ under $p_{k+1|k}^{n}$, where $F_k^n(\cdot)$ is the mean function of $p_{k+1|k}^{n}$. Intuitively, this can be understood as representing marginal score as conditional expectation of finite difference.

Finally, acknowledging these two properties, the paper proposes a novel (approximate) IPF method named Diffusion Schrödinger Bridge (DSB). Assume we have marginals at the two ends in time $t = 0, t=T$. Define two discrete-time Markov chain models: one runs in one time direction, and the other runs in reverse-time, combined with their corresponding initial distributions. For each $n$-th DSB step, by fixing one chain, e.g. $p_{k+1|k}^{n}$, the other chain $q_{k|k+1}^{n}$ learns their optimal mean by minimizing mean-squared loss, whose target is the conditional expected value of $X_{k+1} + F_{k}^n(X_k) - F_{k}^n(X_{k+1})$ under $p_{k+1|k}^{n}$. By alternating time directions of these mean matchings, both Markov chains learn to map from one distribution at one end to another. While the proposed method indirectly solves one IPF step, the authors demonstrate that the proposed methods converge in practice. The authors also emphasize that the proposed method works as long as sampling from the marginal distributions at two time-ends is available.

Furthermore, the paper improves theoretical analysis in previous studies on IPFs to solve SB problems and score matching. In particular, the paper shows the convergence of both the discrete-time and continuous-time IPFs under more general assumptions. Another example includes the convergence of learning continuous-time diffusion-based generative models via score matchings for state-independent and time-homogeneous diffusion.

In the experiments, the paper demonstrates that the proposed DSBs can be used in generative modeling and modeling interpolations between two distributions.

**Limitations And Societal Impact:**

(Limitations)
I strongly believe that the paper is excellent. However, I'm concerned that the limitations of the proposed method are understated. Improving such discussion will enormously improve the quality of the paper.

(Societal Impact)
N/A

**Main Review:**

(Originality & Significance)
In my understanding, the paper's contributions are clear, and I also consider that the results are essential for several reasons:
First, it introduces a novel learning model to attack SB problems, learning diffusions between two distributions, which is previously known to be challenging for high-dimensional settings.

Second, the proposed method can be applied to various application cases, including generative models and learning random processes that transform one distribution to another. The proposed method can also be understood as one type of generalization of now-popular diffusion-based generative models. Moreover, the proposed method can be extended to more general cases beyond state-independent and time-homogeneous diffusions.

Third, the paper provides thorough theoretical discussions to motivate the proposed method.

Finally, the paper demonstrates the effectiveness of the proposed method by a series of experiments.


(Quality & Clarity)
In general, the paper has a well-organized structure to motivate the proposed method and provide thorough discussions on relevant topics. However, the following aspects of the paper can be improved.

First, discussions on the limitations of the proposed method can be improved. I found that the main text only emphasizes the pros of the proposed method without discussing any of its critical drawbacks. For example, the paper only states that the proposed method improves generation qualities for a given number of discretization steps compared to previous diffusion-based models. However, the main text doesn't correctly mention that the improvement is achieved by increasing the training time. Moreover, I can only find many non-trivial implementation tricks of DSB at the very end of the supplementary materials. Without properly mentioning such characteristics of the proposed method, the main text sounds misleading to me.

Second, the connection between Theorem 1 and the proposed method can be improved. I initially thought that Theorem 1 is introduced to prove the convergence of the approximate IPS (= DSB) (maybe I'm wrong). However, I couldn't find if Theorem 1 is used to discuss the convergence of the proposed method.

(Typos)
- Lines 609-610, $p_{k|k+1}$ → $p^n_{k|k+1}$.

**Time Spent Reviewing:**

>12hrs

---

> ### Author Response · Authors · 2021-08-09
> **Response to Reviewer dJt4**
>
> Thank you for your positive comments and feedback.
>
> ---------------------------
>
> **“I'm concerned that the limitations of the proposed method are understated. Improving such discussion will enormously improve the quality of the paper.”** \
>    **“I found that the main text only emphasizes the pros of the proposed method without discussing any of its critical drawbacks.”**
>
> We will clarify the limitations and point out areas for future work. In particular:
>   - Due to its iterative nature, the implementation of our proposed method is not as straightforward when compared to existing score-based methods as one needs to assess convergence of training a neural network at each IPF step.
>   - It is not clear how to adapt some of the standard training techniques used to improve performance in regular score matching to DSB, e.g. exponential moving average (EMA) of weights across IPF iterations.
>   - It is not clear how to assess the trade off between number of diffusion steps and number of IPF iterations.
> ---------------------------
>
> **“the paper only states that the proposed method improves generation qualities for a given number of discretization steps compared to previous diffusion-based models. However, the main text doesn't correctly mention that the improvement is achieved by increasing the training time”**
>
> It is not clear to us whether the training time will be longer or shorter than existing methods in general.
> - Our approach requires fewer diffusion steps per IPF iteration, which makes the training of each IPF iteration faster than existing approaches.
> - It may be possible to tune the number of diffusion steps and the number of required IPF iterations to improve training time compared to existing score methods. We noticed such improvements when training networks for MNIST but not with CelebA. We note that such comparisons are application specific and highly dependent on tuning.
> - There are many training techniques for previous diffusion methods which have been developed over time. We have deployed some of these techniques but we believe there is still a lot more scope for technical innovation to improve the training time of our approach.
>
> ---------------------------
> **“Moreover, I can only find many non-trivial implementation tricks of DSB at the very end of the supplementary materials. Without properly mentioning such characteristics of the proposed method, the main text sounds misleading to me.”**
>
> We will revise and mention these techniques in the main text. The originality of our contribution is not in the set of techniques used to scale the methods since many of these techniques are common in SGM papers.
>
> - The implementation techniques we introduce in the supplementary materials improve training efficiency and performance for limited computing resources. They are supplementary to the main method, and are not necessarily needed. For instance, none of these techniques were used for the 2D examples.
> - Note that many of the techniques we describe are not novel and are commonly used in existing work such as [1,2,3] e.g. exponential moving average (EMA) of weights
>
> ---------------------------
>
> **“the connection between Theorem 1 and the proposed method can be improved. I initially thought that Theorem 1 is introduced to prove the convergence of the approximate IPS (= DSB) (maybe I'm wrong). However, I couldn't find if Theorem 1 is used to discuss the convergence of the proposed method.”**
>
> In the revised version of our paper, we will clarify that Theorem 1 is not used to prove the convergence of DSB but to provide a theoretical justification for existing score matching approaches (Song et al., 2021) [4].
>
> Theorem 1 highlights that existing score matching approaches require a large number of diffusion steps in order to be performant. This is because the forward diffusion needs many steps to converge to a distribution that is close to the prior distribution.
>
> In contrast, DSB circumvents the need for using a large number of diffusion steps. Hence the role of Theorem 1 is to provide a theoretical justification for score matching approaches, emphasize their limitation and motivate our contribution.
>
> ---------------------------
>
> [1] Song, Yang, and Stefano Ermon. "Improved techniques for training score-based generative models." NeurIPS (2020). \
> [2] Jolicoeur-Martineau, Alexia, et al. "Adversarial score matching and improved sampling for image generation." ICLR. 2020. \
> [3] Nichol, Alex, and Prafulla Dhariwal. "Improved denoising diffusion probabilistic models." arXiv preprint arXiv:2102.09672 (2021). \
> [4] Song, Yang, et al. "Score-Based Generative Modeling through Stochastic Differential Equations." International Conference on Learning Representations. 2020.

---

### Decision · Program_Chairs · 2021-09-27

**Decision:**

Accept (Spotlight)

**Comment:**

All reviewers recommended accepting this paper, and even if there were a number of concerns, the consensus among the reviewers was clear. This was an enjoyable paper to read and I recommend that you address the issues raised by the reviewers in the camera-ready, and especially pay attention to the comments regarding improving clarity and facilitating understanding.